# DeepACSON automated segmentation of white matter in 3D electron microscopy

Ali Abdollahzadeh [1], Ilya Belevich [2], Eija Jokitalo [2], Alejandra Sierra [1,3]✉ & Jussi Tohka [1,3]

Tracing the entirety of ultrastructures in large three-dimensional electron microscopy (3D-EM) images of the brain tissue requires automated segmentation techniques. Current segmentation techniques use deep convolutional neural networks (DCNNs) and rely on high-contrast cellular membranes and high-resolution EM volumes. On the other hand, segmenting low-resolution, large EM volumes requires methods to account for severe membrane discontinuities inescapable. Therefore, we developed DeepACSON, which performs DCNN-based semantic segmentation and shape-decomposition-based instance segmentation. DeepACSON instance segmentation uses the tubularity of myelinated axons and decomposes under-segmented myelinated axons into their constituent axons. We applied DeepACSON to ten EM volumes of rats after sham-operation or traumatic brain injury, segmenting hundreds of thousands of long-span myelinated axons, thousands of cell nuclei, and millions of mitochondria with excellent evaluation scores. DeepACSON quantified the morphology and spatial aspects of white matter ultrastructures, capturing nanoscopic morphological alterations five months after the injury.

[1] A. I. Virtanen Institute for Molecular Sciences, University of Eastern Finland, Kuopio, Finland. [2] Electron Microscopy Unit, Institute of Biotechnology, University of Helsinki, Helsinki, Finland. [3] These authors contributed equally: Alejandra Sierra, Jussi Tohka. ✉email: alejandra.sierralopez@uef.fi

Recent advances in automated serial-sectioning electron microscopy (EM) techniques enable acquiring 3-dimensional (3D) image datasets of brain ultrastructures, from hundreds of micrometers of tissue with a voxel-size less than ten nanometers[1–3]. Quantitative analysis of key morphological parameters of ultrastructures or mapping neuronal connections from the acquired EM datasets requires annotating individual components. Manual annotation of ultrastructures in even a small 3D-EM dataset is tedious, consuming thousands of hours of experts' time. Berning et al.[4] reported that manual annotation of 215 neurites in $5 \times 10^8$ voxels required 1500 h, and we estimated that the manual segmentation of $15 \times 15 \times 15\ \mu m^3$ of white matter ultrastructures requires 2400 h[5]. Semi-automated segmentation methods based on machine learning approaches[4,6] have improved the rate of segmentation, but still require a considerable amount of manual interaction because of the manually extracted skeletons of neuronal processes, proofreading, or correction of errors.

State-of-the-art automated segmentation techniques use deep convolutional neural networks (DCNNs) to trace ultrastructures in EM volumes[7–11]. A DCNN-based segmentation technique generally comprises two steps: semantic and instance segmentation. Semantic segmentation assigns each voxel a probability value of belonging to a tissue type, and instance segmentation turns the probability maps into individual object instances. DCNNs are typically used for semantic segmentation, whereas other more traditional image analysis techniques are used for instance segmentation. Moreover, the segmentation techniques generally favor a bottom-up design, i.e., over-segmentation and subsequent merge. Examples are DeepEM3D[8], applying 3D watershed transform on DCNN probability maps of the neuronal membrane, or U-Net MALA[9], where an iterative region agglomeration is applied on predicted affinities between voxels from a 3D U-Net[12]. More recently, Januszewski et al.[10] suggested flood-filling networks (FFNs), a single-object tracking technique, and Meirovitch et al.[11] introduced cross-classification clustering, a multi-object tracking technique, merging the semantic- and instance segmentation in recurrent neural networks. These techniques have a bottom-up design, where recurrent networks maintain the prediction for the object shape and learn to reconstruct neuronal processes with more plausible shapes.

Although these automated EM segmentation techniques have yielded accurate reconstructions of neuronal processes, they have been applied to very high-resolution EM images, exploring synaptic connectivity. Imaging large tissue volumes at synaptic resolutions generates massive datasets. For example, imaging 1 mm³ tissue volume at $4 \times 4 \times 40\ nm^3$ generates a dataset of 1500 tera-voxels in size, demanding fully automated image acquisition techniques and microscopes, which run for several months continuously[2,3]. We can make acquiring large tissue volumes a plausible task by reducing the image resolution: imaging 1 mm³ tissue at $50 \times 50 \times 50\ nm^3$ generates a dataset of eight tera-voxels during few days. However, imaging at low-resolution can limit the visualization of the cellular membranes, for example, at nodes of Ranvier, where no distinctive image feature differentiates the intra- and extra-axonal space of a myelinated axon. Distinctive image features are required for a segmentation technique with a bottom-up design that is subjected to greedy optimization, making the locally optimal choice at each stage while searching for a global optimum. Therefore, the mentioned automated techniques[7–11] cannot be used to segment low-resolution images. For example, techniques such as DeepEM3D[8] and its cloud-based implementation[7] essentially rely on a precise semantic segmentation and apply no mechanism to correct potential topological errors during instance segmentation. Therefore, semantic segmentation errors propagate into instance segmentation as either over- or under-segmentation. Techniques such as FFN[10] and its multi-object tracking counterpart[11], where networks learn the shape of a neural process, would experience over-segmentation errors. Merging FFN super-voxels does not necessarily generate a correct segmentation of an axon as the segmentation leaks to the extra-axonal space at nodes of Ranvier.

Our goal is to segment low-resolution images in a large field-of-view using the information learned from the high-resolution images acquired in a small field-of-view. To achieve this goal, we developed a pipeline called DeepACSON, a Deep learning-based AutomatiC Segmentation of axONs, to account for severe membrane discontinuities inescapable with low-resolution imaging of tens of thousands of myelinated axons. The proposed pipeline utilizes an innovative combination of the existing deep learning-based methods for semantic segmentation and a devised shape decomposition technique for instance segmentation that uses the information about the geometry of myelinated axons and cell nuclei. Applying DeepACSON, we were able to segment low-resolution large field-of-view datasets of white matter automatically. The instance segmentation of DeepACSON approaches the segmentation problem from a top-down perspective, i.e., under-segmentation and subsequent split, using the tubularity of the shape of myelinated axons and the sphericality of the shape of cell nuclei. We applied DeepACSON on low-resolution, large field-of-view 3D-EM datasets acquired using serial block-face scanning EM[13] (SBEM). The SBEM volumes were obtained from the corpus callosum and cingulum, in the same dataset, of five rats after sham-operation ($n = 2$) or traumatic brain injury (TBI) ($n = 3$). The images were acquired ipsi- and contralaterally, thus for five rats, we had ten samples. Each sample was simultaneously imaged at two resolutions in two fields-of-view: high-resolution images, $15 \times 15 \times 50\ nm^3$, were acquired in a small field-of-view, $15 \times 15 \times 15\ \mu m^3$, and low-resolution images, $50 \times 50 \times 50\ nm^3$, were acquired in a large field-of-view, $200 \times 100 \times 65\ \mu m^3$. The low-resolution images covered a field-of-view 400 times bigger than the high-resolution images. We segmented the high-resolution datasets using our earlier automated ACSON pipeline[5]. We down-sampled the high-resolution images to build a training set for DCNNs for the semantic segmentation of the low-resolution (large field-of-view) images, eliminating the need for manually annotated training sets. Using the DeepACSON pipeline, we segmented the low-resolution datasets, which sum up to $1.09 \times 10^7\ \mu m^3$ of white matter tissue, into myelin, myelinated axons, mitochondria, and cell nuclei. The segmentation resulted in about 288,000 myelinated axons spanning 9 m, 2600 cell nuclei, and millions of mitochondria with excellent evaluation scores. Our segmentation of white matter enabled quantifying axonal morphology, such as axonal diameter, eccentricity, and tortuosity and the spatial organization of ultrastructures, such as the spatial distribution of mitochondria and the density of cells and myelinated axons. Our analysis indicated that changes in the axonal diameter, tortuosity, and the density of myelinated axons persisted five months after the injury in TBI rats. Our findings can yield a better understanding of TBI, hence its mechanisms. We have made our SBEM datasets, their segmentation, and the DeepACSON software freely available for download and immediate use by the scientific community.

## Results

Figure 1 illustrates DeepACSON, where DCNNs perform semantic segmentation, and 3D shape-analysis techniques perform instance segmentation. DeepACSON segmented and analyzed ten low-resolution, large field-of-view SBEM datasets of white matter.

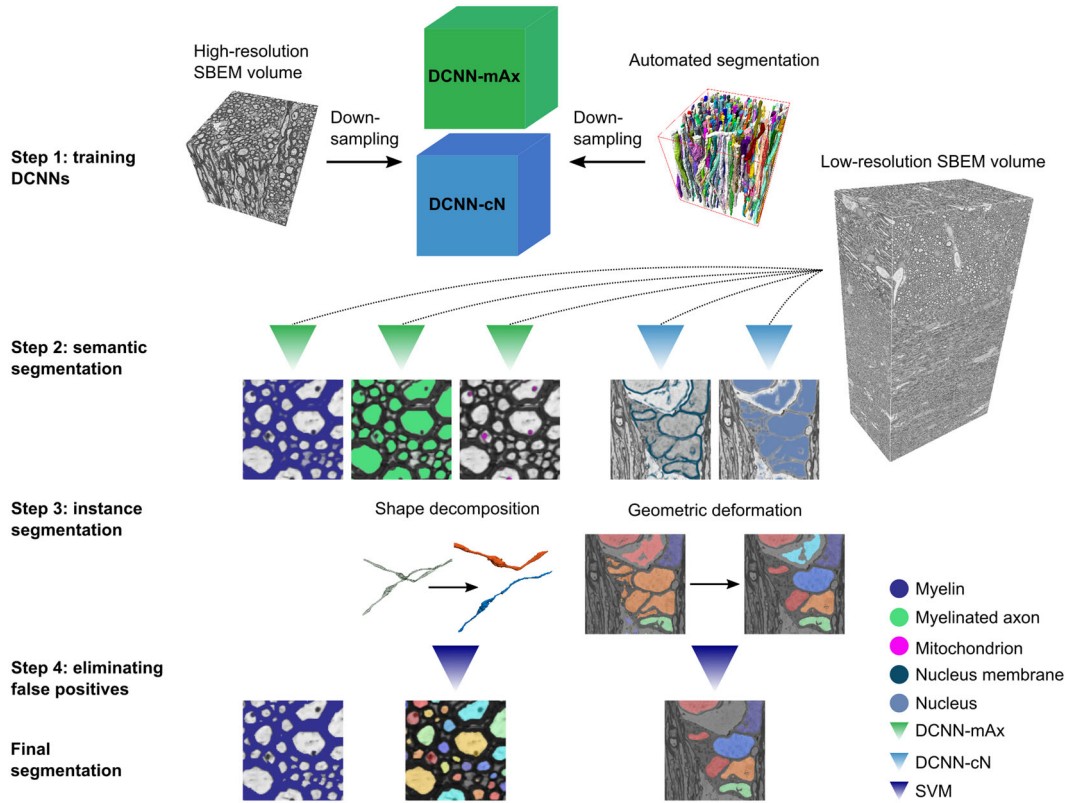

**Fig. 1 DeepACSON pipeline.** Step 1: We used the ACSON segmentation of the high-resolution (small field-of-view) SBEM images down-sampled to the resolution of the low-resolution (large field-of-view) images to train DeepACSON. We trained two DCNNs denoted as DCNN-mAx and DCNN-cN. Step 2: DCNN-mAx returned the probability maps of myelin, myelinated axons, and mitochondria. DCNN-cN returned the probability maps of cell nuclei and the membrane of cell nuclei. Step 3: The segmentation of myelin was finalized by thresholding the myelin probability map. We performed the initial segmentation of myelinated axons by the binarization and connected component analysis. The geometry of the segmented components was subsequently rectified using our newly developed cylindrical shape decomposition (CSD) technique[14]. We performed the segmentation of cell nuclei in a geometric deformable model (GDM) framework by applying elastic deformations to the initial segmentation of cell nuclei. Step 4: The segmentation of myelinated axons and cell nuclei was finalized by eliminating non-axonal and non-nucleus structures using support vector machines (SVMs).

**Datasets**. The samples from rats' white matter were prepared for SBEM imaging by a single person following an exact procedure. We simultaneously acquired SBEM images of the white matter at the low- and high-resolution (Fig. 2a). The low-resolution datasets were acquired from big tissue volumes of $200 \times 100 \times 65\ \mu m^3$ with a voxel size of $50 \times 50 \times 50\ nm^3$. Two-thirds of the low-resolution images correspond to the corpus callosum and one-third to the cingulum (Supplementary Table S1). The high-resolution datasets were acquired from small tissue volumes of $15 \times 15 \times 15\ \mu m^3$ and imaged with a voxel size of $15 \times 15 \times 50\ nm^3$ from the corpus callosum. All the images were acquired from the ipsi- and contralateral hemispheres of sham-operated and TBI animals. Figure 2a, b shows the contralateral corpus callosum and cingulum of a sham-operated rat in the low- and high-resolution. Ultrastructural components such as myelin, myelinated axons, mitochondria, and cell nuclei were resolved in both settings, however, unmyelinated axons (Fig. 2b fuchsia panel, asterisks) and axonal membrane (Fig. 2b cyan panel, arrowheads), were only resolved in the high-resolution images. Figure 2b (purple panel) shows a cell nucleus from a low-resolution image volume, whose membrane is partially resolved.

**Semantic segmentation of white matter ultrastructures**. To provide a human-annotation-free training set for DCNNs, we segmented the high-resolution SBEM datasets using ACSON[5]. We down-sampled the high-resolution datasets and their corresponding segmentation to the resolution of low-resolution datasets

(Fig. 1, step 1). Figure 2c shows a 3D rendering of myelinated axons of a high-resolution SBEM dataset segmented by ACSON. We trained two DCNNs denoted as DCNN-mAx and DCNN-cN: DCNN-mAx for semantic segmentation of myelin, myelinated axons, and mitochondria and DCNN-cN for semantic segmentation of cell nuclei and the membranes of cell nuclei (Fig. 1, step 2). Figure 3a, b present the semantic segmentation of myelinated axons, mitochondria, and myelin returned from DCNN-mAx. Figure 4a, b shows the semantic segmentation of cell nuclei and the membranes of cell nuclei returned from DCNN-cN.

**Instance segmentation of myelin, myelinated axons, and mitochondria**. We segmented myelin by binarizing its semantic segmentation returned from DCNN-mAx (Fig. 1 step 3 and Fig. 3b). We applied connected component analysis on semantic segmentation of myelinated axons (mitochondria were included), achieving an initial segmentation. Because the axonal membrane was not fully resolved, e.g., at nodes of Ranvier (Fig. 3c), this initial segmentation was prone to under-segmentation, requiring a further shape analysis step (Fig. 1, step 3). For that, we devised cylindrical shape decomposition[14] (CSD) algorithm (Supplementary Fig. S1) to decomposes under-segmented myelinated axons (Fig. 3c, $c_1$ and Supplementary Fig. S2). To finalize the segmentation of myelinated axons, we excluded non-axonal instances from the segmentation using a support vector machine (SVM) with a quadratic kernel (Fig. 1, step 4). We segmented mitochondria, applying the connected component analysis on

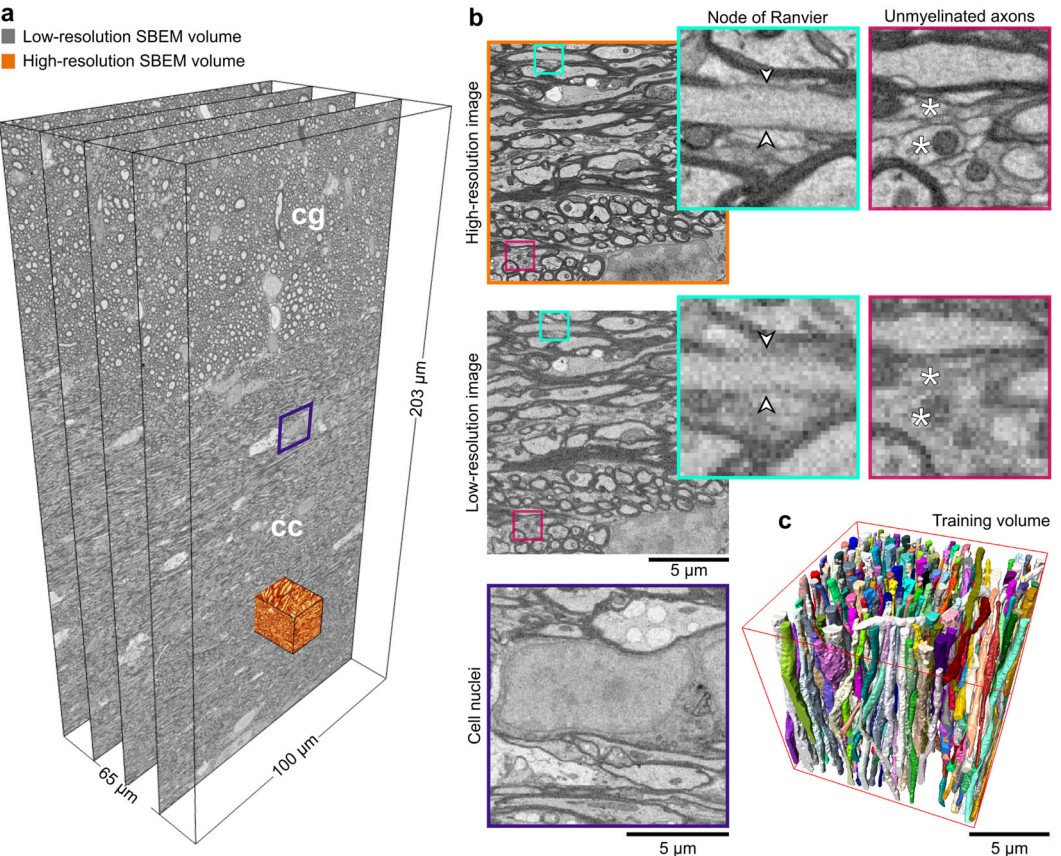

**Fig. 2 Low- and high-resolution SBEM imaging of the contralateral corpus callosum and cingulum of a sham rat. a** We acquired SBEM images of the white matter, corpus callosum (cc) and cingulum (cg), simultaneously at the high- and low-resolution. The field-of-view of the low-resolution dataset is $204.80 \times 102.20 \times 65.30$ μm³ equivalent to $4096 \times 2044 \times 1306$ voxels in x, y, and z directions, respectively, which is approximately 400 times larger than the field-of-view of the high-resolution datasets. **b** Images of the low- and high-resolution datasets acquired from the same location (the orange-rendered volume in **a**). The visualization of the high- and low-resolution images shows that myelin, myelinated axons, mitochondria, and cell nuclei were resolved in both settings. In contrast, the axonal membrane at nodes of Ranvier (cyan panel, arrowheads) and unmyelinated axons (fuchsia panel, asterisks) was only resolved in the high-resolution images. The purple panel shows a cell nucleus from the low-resolution dataset (**a**), where the membrane was resolved, but not continuously. **c** A 3D rendering of myelinated axons in the high-resolution SBEM dataset (contralateral sham #25) segmented by the automated ACSON pipeline.

their semantic segmentation and excluded mitochondria that were not within myelinated axons. Figure 3d shows the 3D rendering of myelinated axons in the contralateral corpus callosum and cingulum of a sham-operated rat. Figure 3e shows the 3D rendering of myelinated axons randomly sampled at two locations from the corpus callosum and one location from the cingulum.

**Instance segmentation of cell nuclei**. The membranes of cell nuclei were severely discontinuous at $50 \times 50 \times 50$ nm³ voxel size; hence their semantic segmentation was discontinuous. We segmented cell nuclei in a geometric deformable model (GDM) framework, where the initial segmentation of a cell nucleus was rectified for its topological errors (see Fig. 1, step 3 and Fig. 4c, d). Non-nucleus instances were excluded from the segmentation using an SVM with a quadratic kernel (Fig. 1, step 4). Figure 4e shows a 3D rendering of cell nuclei of a sham-operated rat dataset.

**White matter morphology analysis**. We quantified several morphological and volumetric aspects of the segmented ultrastructures to demonstrate the applications of the white matter segmentation (Fig. 5, Supplementary Data 1–3). For every myelinated axon, we automatically extracted cross-sections along its

axonal skeleton with a plane perpendicular to the skeleton. The cross-sections were quantified by the equivalent diameter and the eccentricity of the fitted ellipse (Fig. 5a, b). We measured the tortuosity of myelinated axons as $\tau = \frac{l_{Gd}}{l_{Ed}}$, where $l_{Gd}$ is the geodesic distance, and $l_{Ed}$ is the Euclidean distance between the two endpoints of the axonal skeleton. To quantify the spatial distribution of mitochondria, we measured the inter-mitochondrial distances along each myelinated axon. We applied two definitions to quantify the inter-mitochondrial distances: we projected the centroids of mitochondria on the axonal skeleton and measured the geodesic distance between the consecutive projected centroids, and we projected the entirety of mitochondria on the axonal skeleton and measured the shortest geodesic distance between two consecutive mitochondria as shown in Supplementary Fig. S3.

The analysis indicated that the diameter of myelinated axons varies substantially along the axonal skeleton. Moreover, the distribution of diameters was bimodal, which can partially be related to the location of mitochondria[15] (Fig. 5a, b, d). The cross-sections of myelinated axons were elliptic rather than circular (Fig. 5d). Myelinated axons were almost straight as the mean of $\tau$ in the corpus callosum was 1.068 and in the cingulum was 1.072, but there was a big variation in the tortuosity as the standard deviation of $\tau$ was 0.731 and 0.396 in the corpus callosum and

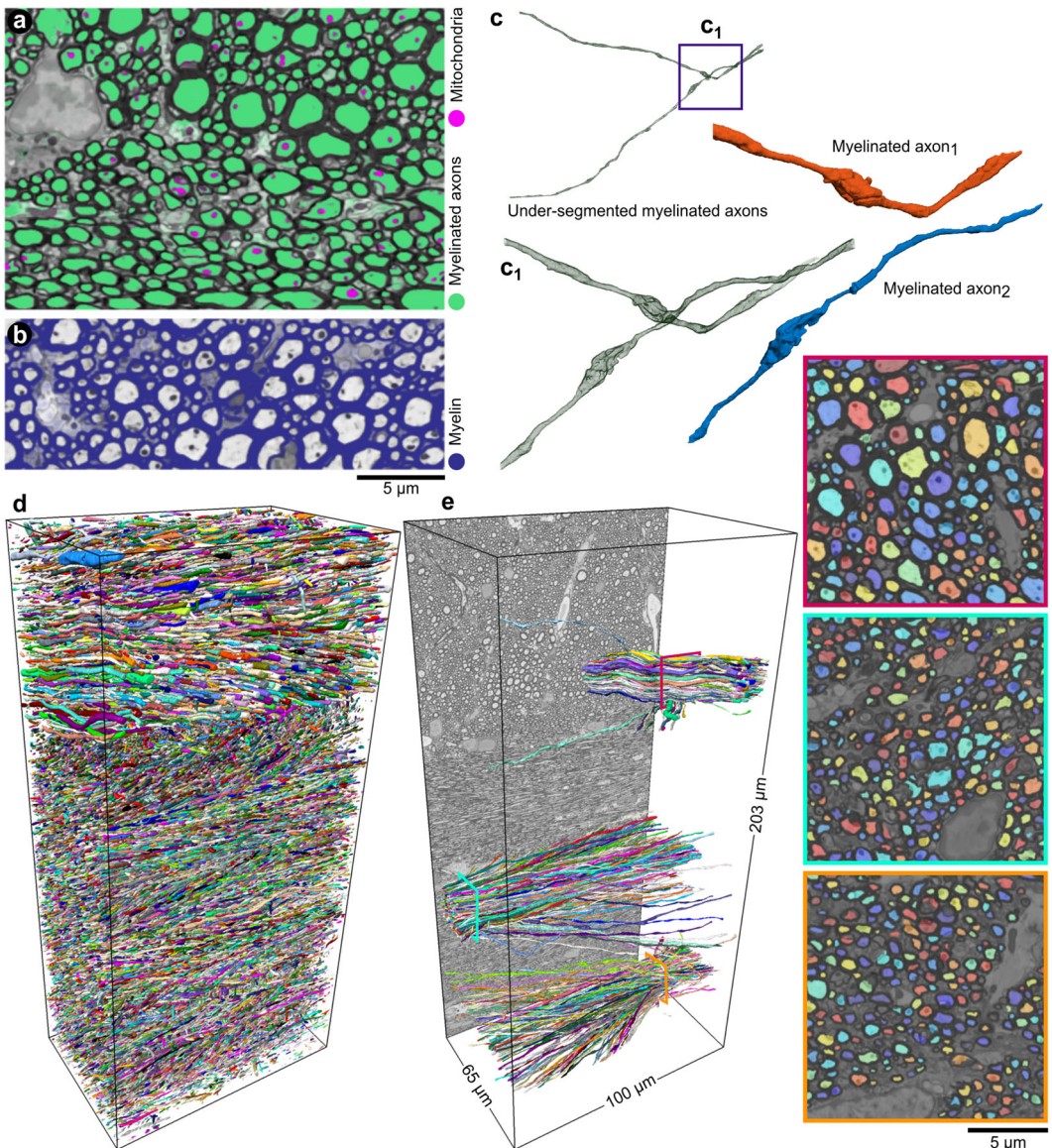

**Fig. 3 DeepACSON segmentation of myelin, myelinated axons, and mitochondria. a, b** The probability maps of myelinated axons, mitochondria, and myelin returned from DCNN-mAx, overlaid on their corresponding BM4D filtered images. **c** The CSD algorithm decomposed myelinated axons with erroneous merges. **d** 3D rendering of DeepACSON final segmentation of myelinated axons (at one-third of the original resolution) in the contralateral corpus callosum and cingulum of sham #25 low-resolution dataset. **e** 3D rendering of myelinated axons sampled at the corpus callosum and cingulum illustrates the variance of the axonal diameter among myelinated axons and the orientation dispersion in these bundles.

cingulum, respectively (see Fig. 5a, d). The distribution of the inter-mitochondrial distance along a myelinated axon was bimodal because mitochondria were either accumulated or appeared distant from each other (Fig. 5a, c, d).

For the statistical hypothesis testing between the sham-operated and TBI animals, myelinated axons were represented by the median of equivalent diameters, the median of the eccentricities, tortuosity, and the mean of inter-mitochondrial distances (Fig. 5d). We subjected the measurements to the nested (hierarchical) 1-way analysis of variance (ANOVA) separately for each hemisphere[16] and set the alpha-threshold defining the statistical significance as 0.05 for all analyses. The equivalent diameter was significantly smaller in the ipsi- ($F = 16.27$, $p = 0.027$) and contralateral cingulum ($F = 29.28$, $p = 0.011$) and in the ipsilateral corpus callosum ($F = 15.75$, $p = 0.047$) of TBI rats as compared to sham-operated rats. Also, the tortuosity in the ipsilateral cingulum of TBI rats was significantly greater than

sham-operated rats ($F = 25.23$, $p = 0.018$). The equivalent diameter of the contralateral corpus callosum of TBI rats was slightly smaller than sham-operated rats, but not significantly ($F = 5.78$, $p = 0.095$). In the ipsilateral cingulum, the inter-mitochondrial distance was slightly smaller in TBI rats, but not significantly ($F = 6.27$, $p = 0.086$). We did not observe a difference in the tortuosity of the contralateral cingulum ($F = 3.20$, $p = 0.134$), ipsilateral corpus callosum ($F = 1.30$, $p = 0.292$), and ipsilateral corpus callosum ($F = 0.03$, $p = 0.878$) when comparing sham-operated and TBI rats. We did not observe a difference between sham-operated and TBI rats in the eccentricity (contralateral cingulum: $F = 1.57$, $p = 0.299$; ipsilateral cingulum: $3.33$, $p = 0.165$; contralateral corpus callosum: $F = 0.67$, $p = 0.471$; ipsilateral corpus callosum: $F = 0.10$, $p = 0.780$). We did not find a difference between sham-operated and TBI rats regarding the inter-mitochondrial distances (distance between centroids) of the contralateral cingulum ($F = 0.33$, $p = 0.603$),

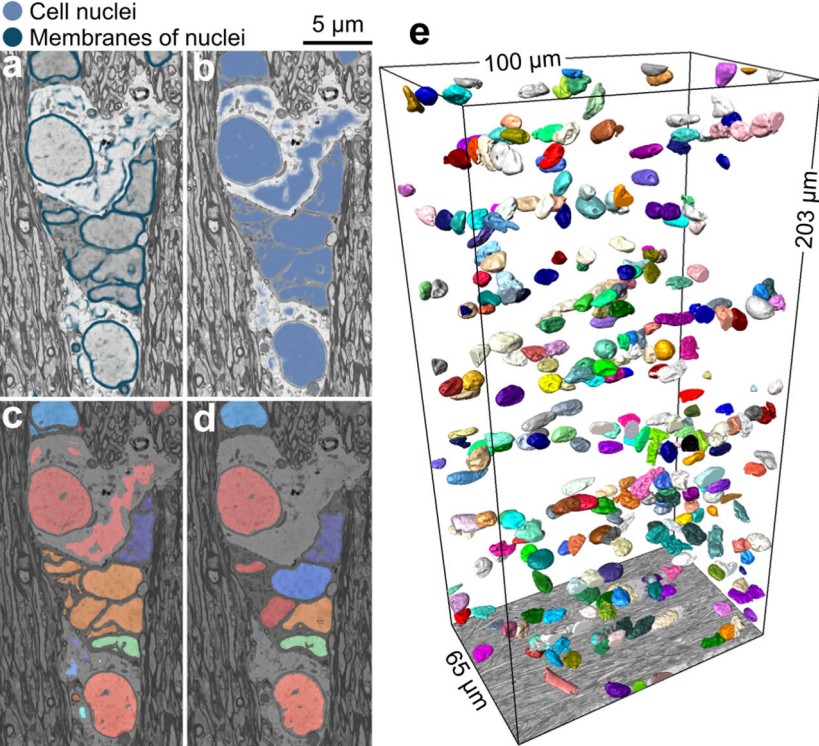

**Fig. 4 DeepACSON segmentation of cell nuclei. a, b** The probability maps of cell nuclei and their membrane were returned from DCNN-cN and overlaid on their corresponding BM4D filtered images. **c** The initial segmentation of cell nuclei contained topological errors as the membrane of cell nuclei exhibited discontinuity. **d** We rectified the segmentation of cell nuclei in a GDM framework and excluded non-nucleus instances by an SVM with a quadratic kernel. **e** 3D rendering of cell nuclei in the contralateral corpus callosum and cingulum of sham #25 dataset.

contralateral corpus callosum ($F = 0.07$, $p = 0.812$), ipsilateral cingulum ($F = 6.26$, $p = 0.086$), and ipsilateral corpus callosum ($F = 1.04$, $p = 0.414$) (Fig. 5d), nor for the inter-mitochondrial distances when measuring the shortest distance between consecutive mitochondria in the contralateral cingulum ($F = 0.28$, $p = 0.630$), contralateral corpus callosum ($F = 0.05$, $p = 0.830$), ipsilateral cingulum ($F = 7.10$, $p = 0.073$), and ipsilateral corpus callosum ($F = 0.43$, $p = 0.577$) (Supplementary Fig. S3). Defining the inter-mitochondrial distance as the distance between centroids of mitochondria was highly correlated with defining the inter-mitochondrial distance as the shortest distance between consecutive mitochondria; the Pearson correlation coefficient was 0.99. We also quantified the volumetric aspects of the ultrastructures (Supplementary Table S2). We could not directly compare the volume of the myelin and myelinated axons among datasets because the volume of the extra-axonal space varies among datasets. Therefore, we calculated the density of myelinated axons as the ratio of the volume of myelinated axons to the myelin volume plus the volume of myelinated axons (Fig. 5e). We observed that the density of myelinated axons was significantly smaller in the ipsilateral cingulum ($F = 13.03$, $p = 0.037$) of TBI compared to sham-operated rats (Fig. 5e). We did not observe a significant difference in the density of myelinated axons in the contralateral cingulum ($F = 4.29$, $p = 0.130$), ipsi- ($F = 3.42$, $p = 0.162$) and contralateral corpus callosum ($F = 2.13$, $p = 0.282$) of TBI rats (Fig. 5e). We also calculated the density of cells defined as the number of cell nuclei over the volume of the corresponding dataset. We did not observe a significant difference in the density of cells comparing TBI and sham-operated rats (Fig. 5f): (contralateral cingulum: $F = 0.16$, $p = 0.717$; ipsilateral cingulum: $F = 1.79$, $p = 0.273$; contralateral corpus callosum: $F = 0.48$, $p = 0.540$; ipsilateral corpus callosum: $F = 1.02$, $p = 0.419$). Figure 5g contains examples of 3D-rendered

myelinated axons in the cingulum, demonstrating the high variability of axonal diameters. Figure 5h shows the sparsity of myelinated axons due to the injury in the cingulum and corpus callosum of TBI rats.

**Evaluations.** We used two test sets to evaluate the DeepACON pipeline: 1) a test set that comprised six high-resolution SBEM volumes down-sampled to the resolution of low-resolution images. We applied this test set to compare DeepACSON against state-of-the-art automated segmentation methods and perform an ablation study on DeepACSON. Labels for this test set was provided automatically using ACSON[5] and proofread by A.S. In this test set, each SBEM volume included approximately 300 axons, thus, we evaluated DeepACSON on approximately $6 \times 300 = 1800$ myelinated axons; 2) a test set, which comprised 50 patches of size $300 \times 300$ voxels only for the expert evaluations. We randomly sampled every low-resolution dataset for five non-overlapping windows of size $300 \times 300$ voxels (10 datasets, 50 samples). Each patch, on average, included approximately 130 axonal cross-sections and 30 mitochondria. Therefore, the expert has evaluated approximately 6500 axonal cross-sections and 1500 mitochondria in total. The expert had no access to the dataset ID nor the sampling location. The expert evaluated the sampled images of the final segmentation by counting the number of true-positives (TP), false-positives (FP), and false-negatives (FN).

We compared DeepACSON with DeepEM2D[8] and DeepEM3D[8], which rely on a precise semantic segmentation, and FFN[10], which accounts for the shape of neural processes during instance segmentation. We trained DeepEM2D/3D[8] using the same training set as DeepACSON to segment myelinated axons. To train FFN[10], we used the same training set as DeepACSON but preserving the label of each myelinated axon. We first trained FFN to segment myelin, myelinated axons, and mitochondria, but

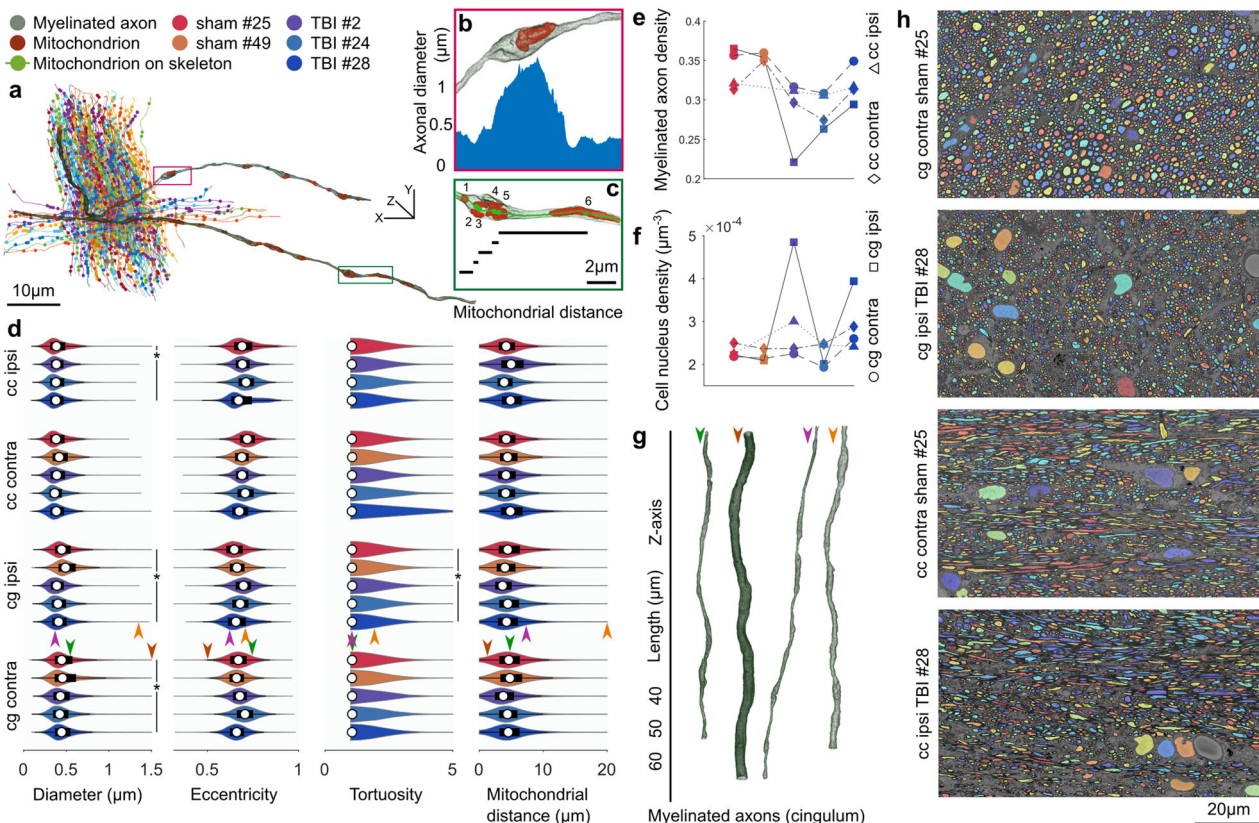

**Fig. 5 White matter morphology analysis. a** A bundle of myelinated axons was sampled from the cingulum of the sham #25 dataset. Myelinated axons are represented by their curve skeletons. The centroids of mitochondria were projected on the axonal skeletons, shown as filled-circles. **b** A small section of a myelinated axon from **a** represents how the axonal diameter can vary substantially along its length. The increased axonal diameter can be related to the accumulation of mitochondria. The plot shows the axonal diameter along the magnified section of the myelinated axon. **c** A small section of a myelinated axon from **a** shows the measure of inter-mitochondrial distance. Five mitochondria are accumulated with distances less than 1 μm, and one mitochondrion is distant from others with over 5 μm distance. **d** DeepACSON quantified the axonal diameter, eccentricity, and tortuosity of about 288 000 myelinated axons and the inter-mitochondrial distance of about 1 800 000 mitochondria. On each bean plot, the central mark indicates the median, and the left and right edges of the box indicate the 25th and 75th percentiles, respectively. The whiskers extend to the most extreme data points not considered outliers. The colors correspond with the animal ID. **e** The comparison of the density of myelinated axons, as the ratio of the volume of myelinated axons to the myelin volume plus the volume of myelinated axons. The color of the indicators corresponds with the animal ID. **f** The comparison of the density of cells, as the number of cell nuclei over the dataset volume. The color of the indicators corresponds with the animal ID. DeepACSON segmented about 2 600 cell nuclei in the ten large field-of-view datasets. **g** 3D rendering of myelinated axons from the cingulum visualizes the normative and outliers of the axonal diameter distribution. Each myelinated axon is given an arrowhead to mark its measurements in panel **d**. **h** Representative images of the cingulum and corpus callosum in sham-operated and TBI rats visualize the smaller density of myelinated axons caused by the injury.

the FFN network generated very poor results. Therefore, we excluded myelin from segmentation and included mitochondria in the intra-axonal space of myelinated axons. We trained DeepACSON and DeepEM2D/3D for one day and FFN for one week on a single NVIDIA Tesla V100-32 GB graphics processing unit (GPU). As shown in Fig. 6a–c, we quantitatively evaluated the segmentation on a test set comprising six SBEM volumes (Supplementary Data 4). We compared these techniques on the segmentation of myelinated axons using three metrics: the variation of information (VOI, split and merge contribution, lower value is better), Wallace indices (split and merge contribution, higher value is better), and adapted Rand error (ARE, lower value is better), as defined in Materials and Methods. DeepACSON outperformed DeepEM2D/3D and FFN as it generated the smallest VOI and ARE and the biggest Wallace measures.

We evaluated the DeepACSON pipeline to understand the behavior of its main components better (Supplementary Data 4). We replaced the original fully convolutional network[17] (FCN) with a U-Net[18] and omitted the block-matching and 4D

filtering[19] (BM4D) and resolution adjustment steps from the pipeline. We considered this high-level ablation study more informative than the traditional ablation study of the details of the standard FCN architecture. In Fig. 6 d–f, we denoted the standard DeepACSON design as DeepACSON-A, which used a light-weight FCN[17] for semantic segmentation. The standard DeepACSON was trained using down-sampled and BM4D filtered volumes. We replaced the FCN design of the standard DeepACSON with a U-Net[18] with residual modules, denoted as DeepACSON-B in Fig. 6 d–f. In this figure, we also show the effect of omitting BM4D denoising as a pre-processing step (DeepACSON-C) and down-sampling the high-resolution images to generate the training set (DeepACSON-D). In addition, we demonstrated the choice of thresholds at which the probability maps were binarized. Evaluations were run over the six SBEM volumes segmented automatically using ACSON[5]. The comparisons showed that the standard DeepACSON performed better than a deeper network, which was prone to over-fitting. Denoising the train/test datasets as a pre-processing step improved our results as did adjusting the resolution between

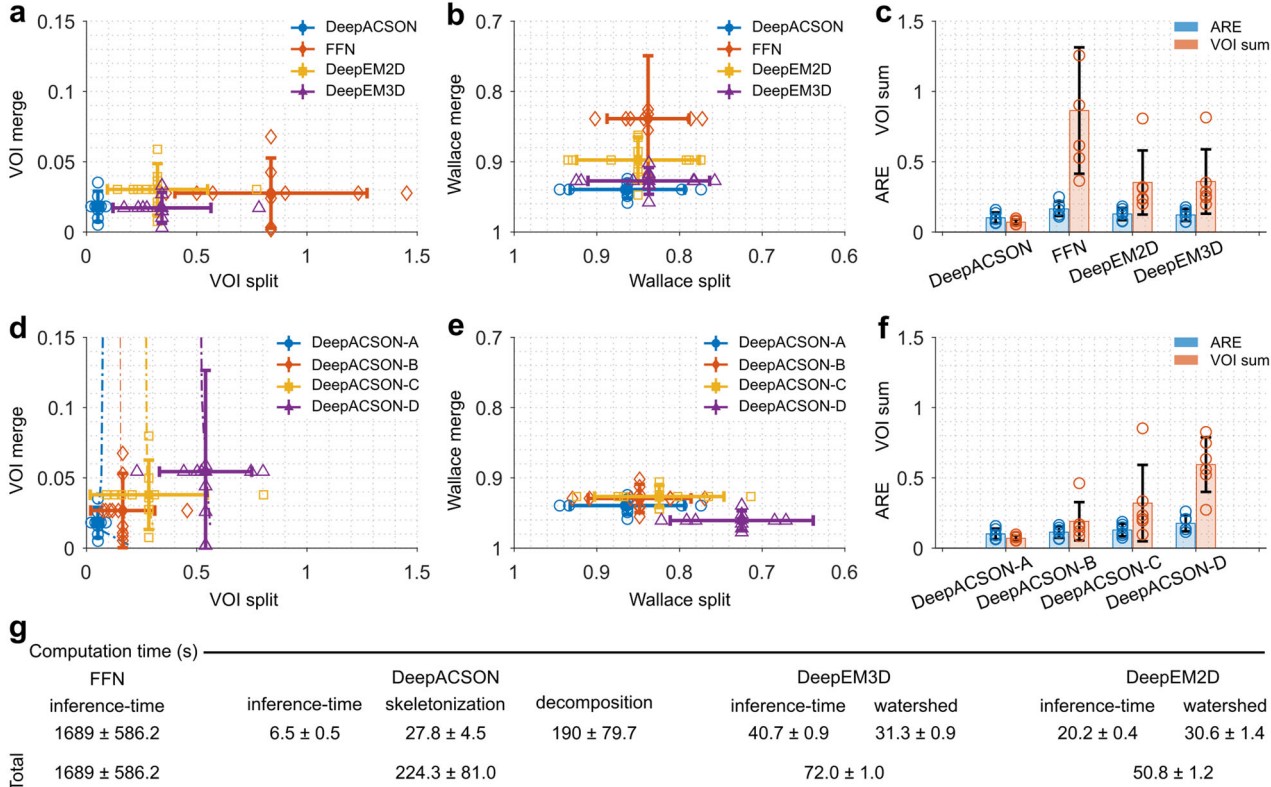

**Fig. 6 DeepACSON quantitative evaluations.** Comparison of DeepACSON against state-of-the-art segmentation methods, DeepEM2D, DeepEM3D, and FFN, using **a** variation of information (VOI, split and merge contribution, lower value is better), **b** Wallace indices (split and merge contribution, higher value is better), and **c** adapted Rand error (ARE, lower value is better) and the sum of VOI split and VOI merge (VOI sum, lower value is better). DeepACSON outperformed other techniques as it produced the smallest VOI split, VOI merge, VOI sum, and ARE, and the biggest Wallace split and merge values. Comparison of the design parameters of DeepACSON: standard DeepACSON (DeepACSON-A), a U-Net with residual modules (DeepACSON-B), the effect of BM4D denoising (DeepACSON-C), and adjusting the resolution between the training and test sets (DeepACSON-D) over **d** VOI (split and merge contribution) **e** Wallace indices (split and merge contribution), and **f** ARE and VOI sum. The filled-circles and error bars show the mean and standard deviation of the evaluations, respectively. The dash-dotted lines show the choice of binarization threshold. The comparisons were run over the best threshold, i.e., smallest VOI merge and VOI split. **g** Comparison of the computation time of DeepACSON against DeepEM2D/3D and FFN (mean ± standard deviation). All comparisons were run over six test SBEM datasets of size $290 \times 290 \times 285$ voxel$^3$, automatically segmented using the ACSON pipeline[5].

the training and test sets. We binarized the probability maps at thresholds that generated the smallest VOI split/merge values.

In addition, we evaluated the semantic segmentation of the standard DeepACSON on an ultrastructural level, i.e., myelin and myelinated axons (including mitochondria), on the six SBEM volumes. For this evaluation, we reported precision (positive predictive value), recall (sensitivity), and F1 scores (harmonic mean of precision and recall) in Supplementary Fig. S4. The average F1 score was $0.886 \pm 0.049$ for myelin and $0.861 \pm 0.037$ for myelinated axons.

We evaluated the performance of the two SVMs by the leave-one-group-out (LOGO) cross-validation approach: the classifier was trained excluding the data from one group of animals (sham-operated or TBI) and evaluated against the excluded data (Supplementary Fig. S4). F1 scores were 0.988 (sham datasets) and 0.955 (TBI) for eliminating non-axonal structures. For eliminating non-nucleus structures, F1 scores were 0.969 (sham datasets) and 0.981 (TBI). The LOGO cross-validation showed that the performance of the SVMs was equally robust in all datasets, regardless of the condition (sham-operated or TBI).

Finally, an expert (A.S.) evaluated the DeepACSON segmentation of myelinated axons and mitochondria at an object-level using GUI-based visualization software, called gACSON[20] (Supplementary Fig. S5). We developed gACSON to facilitate the visualization and validation procedures for the expert. gACSON enables the expert to manually click on the segmented image overlaid on the original EM image and express if a segmentation component was a TP, FP, or FN, as shown in Supplementary Fig. S5. The expert's evaluation is an added qualitative measure over the entire pipeline, which resulted in the following scores: myelinated axons (precision: $0.965 \pm 0.027$, recall: $0.877 \pm 0.061$, and F1 score: $0.918 \pm 0.038$) and mitochondria (precision: $0.856 \pm 0.100$, recall: $0.804 \pm 0.091$, and F1 score: $0.823 \pm 0.067$).

**Computation time.** DeepACSON required approximately five days to segment a low-resolution SBEM dataset of $4000 \times 2000 \times 1200$ voxels into its final segmentation (Supplementary Table S3). Approximately 40% of the DeepACSON computation time was spent on BM4D denoising. We run BM4D filtering on non-overlapping patches of the SBEM volumes to enable parallel processing. BM4D is computationally expensive for denoising large EM volumes; however, as shown in Fig. 6d–f, the application of BM4D improved the segmentation results. The number of floating point operations required by BM4D is $O(N)$ with large constants, where $N$ is the number of voxels[21]. Approximately 30% of the DeepACSON computation time was spent on the CSD

algorithm. In more detail, the time complexity of the sub-voxel precise skeletonization is $O(n \, N_\Omega \log N_\Omega)$, where $n$ is the number of skeleton branches, and $N_\Omega$ is the number of voxels of a discrete object. The $N_\Omega \log N_\Omega$ factor is from the fast marching algorithm[22]. The time complexity to determine a critical point is $O(N_p)$, where $N_p$ is the number of inquiry points to check for the cross-sectional changes in a decomposition interval. Therefore, the overall time complexity of the CSD algorithm is $O(n \, N_\Omega \log N_\Omega) + O(N_p)$. The inference time of the FCN corresponded to approximately 10 % of the DeepACSON computation time. For the general analysis of the time complexity of FCNs, we refer to[23].

We also compared the computation time of DeepACSON, DeepEM2D, DeepEM3D, and FFN techniques (see Fig. 6g and Supplementary Data 4 for results). The techniques were compared over the six test datasets on a computer with an NVIDIA Tesla V100-32 GB GPU, 2 × Intel Xeon E5 2630 CPU 2.4 GHz, and 512 GB RAM. DeepEM2D and DeepEM3D had the shortest computation time (about 1 minute) as the segmentation essentially relies on an Inception-ResNet-v2 network[24] and watershed segmentation. DeepACSON required about 4 minutes (using 15 CPU cores) to segment the test datasets. FFN required the longest computation time for an end-to-end segmentation (about 28 minutes).

## Discussion
We developed DeepACSON, using an innovative combination of DCNN-based semantic segmentation and shape-decomposition-based instance segmentation. We applied DeepACSON on low-resolution, large field-of-view SBEM volumes of white matter to trace long myelinated axons. DeepACSON segmented hundreds of thousands of myelinated axons, thousands of cell nuclei, and millions of mitochondria in ten SBEM datasets. We provided DeepACSON with human-annotation-free training sets, using the segmentation of high-resolution SBEM volumes by our earlier automated pipeline, ACSON[5]. DeepACSON evaluations demonstrated excellent scores in the segmentation of white matter ultrastructures, outperforming state-of-the-art methods. DeepACSON quantified the morphology of myelinated axons, the spatial distribution of mitochondria, and the density of myelinated axons and cells in the white matter of sham-operated and TBI rats. Our findings indicated that changes in the axonal diameter, tortuosity, and the density of myelinated axons due to TBI were persistent even five months after the injury.

The top-down design of DeepACSON instance segmentation allows for including the tubularity of the shape of myelinated axons and the sphericity of the shape of cell nuclei that make it different from the bottom-up design of the current automated neurite segmentation techniques[8–11,25–27]. A related study is MaskExtend[28] that proposed a 3D watershed transform, subsequent merge, and the detection of X-shape objects to find under-segmentation errors. MaskExtend makes a restrictive assumption about the geometry of under-segmented objects to be X-shape. Unlike MaskExtend, we allow for more general shape for under-segmented objects as X-shape objects are a special case in our CSD algorithm. Also, CSD reconstructs an under-segmented object on its own by using generalized cylinders as opposed to MaskExtend that requires accessing the watershed segments and merging them at a high merge-threshold[28]. Falk et al.[29] also addressed under-segmentation errors in cells by inserting an artificial one-pixel wide background ridge between touching instances in the training set. This approach is similar to our technique for segmenting cell nuclei, where we inserted an artificial membranes around cell nuclei using morphological operations and assigned high-weights while training.

We compared our cylindrical shape decomposition approach to other state-of-the-art techniques[30–32], showing that CSD outperforms these methods in decomposition of voxel-based objects[14]. CSD evaluates the preliminary segmentation of myelinated axons in-parallel on high-performance computing (HPC) servers, and if required, decomposes and reconstructs an under-segmented myelinated axon. The parallelizability of CSD makes DeepACSON highly scalable. For example, we analyzed hundreds of thousands of myelinated axons traversing large field-of-view datasets on different CPU cores of different HPC servers, reducing the computation time of the segmentation. We also remark that the CSD algorithm evaluates the cylindricity of an object using the object skeleton. In cases where the surface protrusion of the object is very irregular, the skeletonization may over-estimate the number of skeleton branches. Therefore, CSD may over-segment the surface protrusion, yielding false-positives. We eliminated false-positives after the CSD algorithm using support vector machines.

We trained the FCN of DeepACSON with four SBEM volumes to segment myelinated axons and six volumes to segment cell nuclei. These volumes included about 300 axons, one or two cell nuclei, and approximately $300 \times 300 \times 300$ voxels that is sufficient to train a semantic segmentation network according to our experiments. Training a network for semantic segmentation does not necessarily need many annotated training images[29]. To avoid overfitting, DeepACSON utilized a ten-layers FCN[17] in its original design. We compared this design to a deeper network (U-net with ResNet encoder), demonstrating that the deeper network can experience overfitting and produce worse VOI, ARE, and Wallace indices than the original design.

The validation of DeepACSON demonstrated an equally robust segmentation performance in datasets of both sham-operated and TBI rats. We remark that abnormalities in the shape of myelinated axons, such as myelin delamination, are more frequent in TBI datasets[5]. Furthermore, the expert evaluation showed that the recall of DeepACSON was lower than its precision: we had a bigger number of false-negatives, i.e., myelinated axons that were not segmented, than false-positives. We speculate that many false-negatives were caused by over-segmentation of very thin myelinated axons into small components and the subsequent removal of small components before the shape decomposition step. For example, an axon with a diameter smaller than 0.2 μm was resolved with less than 13 cross-sectional voxels at $50 \times 50 \times 50$ nm$^3$ resolution, making it prone to over-segmentation. The segmentation of myelinated axons at low-resolutions can be difficult for an expert as well: Mikula et al.[33] reported that an expert made an error in every 80 nodes of Ranvier in white matter, when the axonal diameter was greater than 0.5 μm, and the resolution was $40 \times 40 \times 40$ nm$^3$.

The simultaneous high- and low-resolution imaging enables the use of high-resolution images as the training data to segment low-resolution datasets. The high-resolution images can be segmented using automated methods, hence providing a human-annotation-free training set; we used ACSON to segment the high-resolution SBEM datasets[5]. ACSON is fully automated, requires no training data, and segments one of our high-resolution datasets (containing about 300 myelinated axons) in approximately 24 hours. Also, high-resolution imaging resolves the membrane of tissue ultrastructures in several voxels wide resulting in a more accurate segmentation of ultrastructures. Accurate segmentation of ultrastructures yields an accurate training set, reducing the label-noise, i.e., less mismatch between labels and the underlying ultrastructures in the training set.

Acquiring large field-of-view image datasets at high-resolutions for several animals and experimental conditions is time-consuming and costly. Despite the improvement in the image

acquisition rate[2,3,34], microscopes should continuously run for several months to acquire a cubic millimeter at a resolution of $4 \times 4 \times 40$ nm[33]. As a comparison, imaging at $50 \times 50 \times 50$ nm$^3$ resolution leads to approximately 200-fold reduction of the imaging time and image size. Low-resolution imaging allows for axonal tracking[33], morphometry of myelinated axons and spatial distribution analyses of myelinated axons, cell nuclei and mitochondria. Also, low-resolution imaging and the corresponding segmentation provide a base for neuronal tissue modeling and histological validations of magnetic resonance imaging (MRI).

Large field-of-view imaging enables quantifying parameters whose measurement in a small field-of-view is not reliable because the measurement in the small field-of-view may reflect a very local characteristic of the underlying ultrastructure. Particular examples of such parameters are the tortuosity of myelinated axons, inter-mitochondrial distance, and cell density. For those parameters that we could measure in both low-resolution (large field-of-view) and high-resolution (small field-of-view) datasets, notably the axonal diameter and eccentricity, we compared DeepACSON measurements in the low-resolution datasets with ACSON measurements in the high-resolution datasets[5]. This comparison indicated that the quantification in the low-resolution datasets was consistent with the high-resolution datasets[5]. For example, we measured an average equivalent diameter for myelinated axons in the contralateral corpus callosum of sham #25 (sham #49) equal to 0.41 μm (0.44 μm) and 0.44 μm (0.50 μm) in the low- and high-resolution imaging, respectively. Additionally, we measured an average eccentricity for myelinated axons in the contralateral corpus callosum of sham #25 (sham #49) equal to 0.71 (0.69) and 0.72 (0.71) in the low- and high-resolution imaging, respectively. This consistency in the morphology analysis of low- and high-resolution datasets suggests that we can capture the morphology of myelinated axons, even with the coarser resolution of the large field-of-view datasets.

We quantified morphological changes in myelinated axons in white matter five months after a severe TBI. The white matter pathology after TBI is extremely complex. The initial response of an axon to a brain injury can be either degeneration or regeneration[35,36]. Moreover, morphological alterations in axons can persist for years after injury in humans[37,38] and up to one year in rats[39]. We found that TBI significantly reduced the axonal diameter of myelinated axons in the ipsilateral corpus callosum, ipsilateral cingulum, and contralateral cingulum. We further measured that TBI significantly reduced the density of myelinated axons, reflecting the degeneration of axons after injury[38]. We also found that TBI increased the tortuosity of myelinated axons in the ipsilateral cingulum. We speculate that prolonged damage in microtubules might underlie the increase in axonal tortuosity[40].

Ultrastructural tissue modeling is an active research field aiming to bridge the gap between macroscopic measurements and cellular- and sub-cellular tissue levels. Currently, tissue models are based on simplistic representations and assumptions of the ultrastructural features, which typically assume axons to be perfect cylinders, or neglect the variation of the axonal diameter along axons[41,42]. The segmentation of brain tissue ultrastructures in 3D-EM datasets can substitute simplistic biophysical models by more realistic models. Such realistic 3D tissue models open the possibility to investigate underlying reasons for the diffusion MRI contrast and its macroscopic changes in brain diseases[43–46] or investigate conduction velocity in myelinated and unmyelinated axons in electrophysiology[47,48].

## Methods
**Animal model, tissue preparation and SBEM imaging**. We used five adult male Sprague-Dawley rats (10-weeks old, weight 320 and 380 g, Harlan Netherlands B. V., Horst, Netherlands). The animals were singly housed in a room (22 ± 1 °C, 50%

− 60% humidity) with 12 h light/dark cycle and free access to food and water. All animal procedures were approved by the Animal Care and Use Committee of the Provincial Government of Southern Finland and performed according to the guidelines set by the European Community Council Directive 86/609/EEC.

TBI was induced by lateral fluid percussion injury[49] in three rats (TBI #2, #24, #28). Rats were anesthetized with a single intraperitoneal injection. A craniectomy (5 mm in diameter) was performed between bregma and lambda on the left convexity (anterior edge 2.0 mm posterior to bregma; lateral edge adjacent to the left lateral ridge). Lateral fluid percussion injury was induced by a transient fluid pulse impact (21-23 ms) against the exposed intact dura using a fluid-percussion device. The impact pressure was adjusted to 3.2-3.4 atmosphere to induce a severe injury. The sham-operation of two rats (sham #25, #49) included all the surgical procedures except the impact. Five months after TBI or sham operation, the rats were transcardially perfused using 0.9% NaCl, followed by 4% paraformaldehyde. The brains were removed from the skull and post-fixed in 4% paraformaldehyde 1% glutaraldehyde overnight.

The brains were sectioned into 1-mm thick coronal sections using a vibrating blade microtome. Sections from -3.80 mm from bregma from each brain were selected and further dissected into smaller samples containing the areas of interest. We collected two samples from each brain: the ipsi- and contralateral of the cingulum and corpus callosum. The samples were osmium stained using an enhanced staining protocol[50]. After selecting the area within the samples, the blocks were further trimmed into a pyramidal shape with a $1 \times 1$ mm$^2$ base and an approximately $600 \times 600$ μm$^2$ top (face), which assured the stability of the block while being cut in the microscope. Silver paint was used to electrically ground the exposed block edges to the aluminum pins, except for the block face or the edges of the embedded tissue. The entire surface of the specimen was then sputtered with a thin layer of platinum coating to improve conductivity and reduce charging during the sectioning process. The details of the animal model and tissue preparation are described in the ACSON study[5].

The blocks were imaged in a scanning electron microscope (Quanta 250 Field Emission Gun; FEI Co., Hillsboro, OR, USA), equipped with the 3View system (Gatan Inc., Pleasanton, CA, USA) using a backscattered electron detector (Gatan Inc.). The face of the blocks was in the $x$-$y$ plane, and the cutting was in $z$ direction. All blocks were imaged using a beam voltage of 2.5 kV and a pressure of 0.15 Torr. We acquired the high- and low-resolution datasets consistently at one specific location in the white matter in both sham-operated and TBI animals and in both hemispheres. The images were collected with an unsigned 16-bits per voxel. Supplementary Table S1 shows the volume size of the low-resolution datasets.

**DeepACSON segmentation pipeline**. We devised DeepACSON: a DCNN-based segmentation pipeline to automatically annotate white matter ultrastructures in low-resolution 3D-EM datasets (Fig. 1). The DeepACSON pipeline annotated white matter such that myelin and every myelinated axon, mitochondrion, and cell nucleus carries a separate label. Figure 1 shows the four steps of the DeepACSON pipeline. Step 1: Generating a training set for semantic segmentation. DeepACSON requires labeled EM datasets as the training material to learn to perform semantic segmentation tasks. To automatically provide a training set for DeepACSON, we utilized our previously developed ACSON segmentation pipeline[5]. Step 2: Semantic segmentation of white matter ultrastructures. The volume of white matter ultrastructures was highly imbalanced, i.e., numbers of voxels representing different ultrastructures were not equal. Thus, we trained two separate DCNNs: DCNN-mAx to generate the probability maps of myelin, myelinated axons, and mitochondria and DCNN-cN to generate the probability maps of cell nuclei and the membrane of cell nuclei. Step 3: Instance segmentation of myelin, myelinated axons, mitochondria, and cell nuclei. Myelin was segmented by thresholding the myelin probability map. The segmentation of myelinated axons and cell nuclei required further, automatic geometrical and topological corrections after the binarization and connected component analysis. The CSD algorithm was applied to decompose an under-segmented myelinated axon into its axonal components. In a GDM framework, elastic deformations of the initial segmentation of cell nuclei segmented this ultrastructure. Step 4: Automatic elimination of false-positive instances. We eliminated non-axonal and non-nucleus instances, i.e., false-positives, from the segmentation of myelinated axons and cell nuclei by training two separate SVMs.

*Pre-processing: data alignment and denoising*. We used Microscopy Image Browser[51] (MIB; http://mib.helsinki.fi) to pre-process the collected images from the electron microscope. We found the bounding box to the collected images and run the voxel-size calibration. We aligned the images by measuring the translation between the consecutive SBEM images using the cross-correlation cost function (MIB, Drift Correction). We acquired a series of shift values in $x$ direction and a series of shift values in $y$ direction. From each series, the running average of the shift values (window size was 25 values) was subtracted to preserve the orientation of myelinated axons. We applied contrast normalization such that the mean and standard deviation of the histogram of each image matches to the mean and standard deviation of the whole image stack. The images were converted to the unsigned 8-bits format. The size of low-resolution SBEM datasets ranged from 10-15 GB, summing up to 85 GB (8-bits per voxel). To denoise SBEM datasets, we used BM4D[19], a non-local filtering algorithm. The algorithm was run in its low-

complexity mode to reduce the computation time. We processed the low-resolution SBEM datasets in non-overlapping patches with an approximate size of $1000 \times 1000 \times 300$ to account for RAM requirements and enabling parallel BM4D denoising on CPU clusters (see Supplementary Fig. S6).

*Step 1: training sets for DCNNs.* We used high-resolution SBEM datasets of the corpus callosum segmented by the automated ACSON pipeline to generate a training set for DeepACSON. The automated ACSON pipeline was designed for segmenting high-resolution small field-of-view SBEM datasets into myelin, myelinated and unmyelinated axons, mitochondria, and cells. The ACSON pipeline segments white matter based on a bounded volume growing algorithm in which seeds are defined automatically using regional maxima of the distance transform of myelin maps[5]. We generated two training sets: one for training DCNN-mAx and one for training DCNN-cN. The segmented high-resolution datasets were modified before using them for training as follows: the labels of unmyelinated axons and cells were set equal to the background label to generate the DCNN-mAx training set. The labels of all ultrastructures except for the cell nuclei were set equal to the background label to generate the DCNN-cN training set. We created an additional label for the membrane of cell nuclei by a morphological gradient with a flat $11 \times 11$ square structuring element applied to the segmented cell nuclei. The membrane of cell nuclei was 22 voxels or 0.33 μm wide. We over-emphasized the membrane of cell nuclei because the number of voxels representing it was very small compared to the nucleus and background labels. After modifying labels, we uniformly down-sampled the high-resolution datasets in both training sets by a window size of $3 \times 3$ in $x - y$ plane to conform to the resolution of the low-resolution datasets. The DCNN-mAx training set included datasets of the ipsi- and contralateral of sham-operated and TBI rats. It is important to include both intact and injured axons to enrich the training set. The DCNN-cN training set included only ten cell nuclei.

*Step 2: semantic segmentation of white matter ultrastructures; architecture and implementation of DCNNs.* We implemented the DCNNs and data augmentation using the ElektroNN2 library[6] based on the Python Theano framework[52]. ElektroNN was optimized for short model training times and fast inference on large datasets by eliminating the redundant computations of sliding window approaches[53]. The training set was augmented using randomized histogram distortions to become invariant against varying contrast and brightness gradients. In addition, half of the training set underwent image warping consisting of affine and random perspective transformations. We determined the optimal architectural parameters and hyperparameters of the DCNNs experimentally using the training set. We used an FCN architecture with the same architectural parameters for DCNN-mAx and DCNN-cN, as shown in Supplementary Fig. S7. We set the patch size equal to (185, 185, 23) voxels in $(x, y, z)$ directions. The receptive field was (45, 45, 15) voxels, with approximately 3 million trainable parameters. The DCNNs were trained on the cross-entropy loss with Softmax activation. We set the batch size equal to 1 because there was a sufficient number of prediction neurons for updating the gradient. For the optimization, we used Adam optimizer[54] and set its initial learning rate $\alpha = 5 \times 10^{-5}$, the exponential decay rate for the first moment $\beta_1 = 0.9$, the exponential decay rate for the second-moment $\beta_2 = 0.999$, and the weight decay $= 5 \times 10^{-6}$ (we use the same notation as in[54]). After every 1 000 training steps, the learning rate $\alpha$ was decayed by a factor of 0.995. For DCNN-mAx, we set the class weights to 0.2, 0.5, 1, and 1 for the background, myelin, myelinated axons, and mitochondria, respectively. The class weights for DCNN-cN were set to 0.1, 1, and 4 for the background, cell nuclei, and the membrane of cell nuclei, respectively.

For the semantic segmentation of myelinated axons, we also tested a U-Net[18] with a ResNet[55] in the encoding path (ResNet-34 with 21 million parameters[55], pre-trained on the ImageNet dataset[56] as implemented in PyTorch[57]). We selected the U-Net architecture because it is widely used for the semantic segmentation of biomedical image volumes, resulting in precise segmentation and not requiring many annotated training images[29]. Also, ResNet, which we used in the encoding path of U-Net, is the most widely used network for image feature extraction. The residual blocks of ResNet are easy to optimize and can gain accuracy from increased network depth[55]. Supplementary Fig. S8 shows this network architecture. In the encoding path of the U-Net, the height and width of the feature maps were halved, and the depth of the feature maps was doubled. In the decoding path of the U-Net, the height and width of the feature maps were doubled, and the depth of the feature maps were halved. The basic residual blocks of ResNet-34 were constructed on convolutional layers, including rectified linear unit (ReLU) as the activation function, and batch normalization[58]. The basic decoding blocks applied nearest-neighbor interpolation for up-sampling the feature maps to recover the spatial resolution of input images. Feature maps generated in the encoding path were concatenated to the corresponding feature maps in the decoding path by the skip connections. The model was trained using cross-entropy loss with Softmax normalization. We set the batch size equal to eight. We normalized the input images to have the same mean and standard deviation as the pre-trained network used for normalizing its input images. Also, half of the training set was augmented by geometrical transformations, such as horizontal and vertical flips, scaling, shifting, and rotation. For the optimization, we used Adam optimizer[54] and set its initial learning rate $\alpha = 1 \times 10^{-4}$, the exponential decay rate for the first moment $\beta_1 = 0.9$, the exponential decay rate for the second-moment $\beta_2 = 0.999$, and the weight decay $= 1 \times 10^{-5}$ (we use the same notation as in[54]). We set the class

weights to 0.3, 0.3, 0.3, and 1 for the background, myelin, myelinated axons, and mitochondria, respectively. We trained this U-Net architecture with the same datasets as FCN network in three scenarios: 1) the training set was BM4D denoised and down-sampled, 2) the training set was not BM4D denoised but down-sampled, 3) the training set was BM4D denoised but not down-sampled. In the first case, we evaluated the effect of using a deeper network than our FCN design, and in the second and third cases, we evaluated the effect of BM4D denoising and down-sampling the high-resolution datasets of the training set.

*Step 3: instance segmentation of white matter ultrastructures.*
Step 3.1: myelin, myelinated axons, and mitochondria segmentation. DCNN-mAx generated probability maps of myelin, myelinated axons, and mitochondria. We binarized the probability maps of myelin, myelinated axons, and mitochondria with a threshold of $\theta_{myelin} = 0.5$, $\theta_{mAxon} = 0.8$, and $\theta_{mitochondria} = 0.8$; the thresholds were assigned based on the VOI split/merge error values. The binarized volume of myelin required no further processing, and it was considered to be the final myelin segmentation. The binarized volume of mitochondria was dilated with a flat $3 \times 3 \times 3$ structuring element, and unified with the binarized volume of myelinated axons to form the intra-axonal space of myelinated axons which we denote by $I_{IAS}$. The initial segmentation of individual myelinated axons was obtained using connected component analysis on $I_{IAS}$. The segmented volumes underwent a morphological volume closing with a flat $3 \times 3 \times 3$ cubic structuring to close the holes. We excluded volumes whose size was smaller than 4 700 voxels (the volume of a circular cylinder with a base radius of 0.25 μm and height of 3 μm, resolved with an isotropic $50 \times 50 \times 50$ nm$^3$ voxel size).

In the low-resolution datasets, the axonal membrane was not resolved, e.g., at nodes of Ranvier. Thus, the initial segmentation of myelinated axons might exhibit under-segmentation errors (Supplementary Fig. S2). We developed a specific shape analysis algorithm, CSD, to decompose an object into its semantic-components. As shown in Supplementary Fig. S1a, b, we defined the object as the union of several tubular components, similar to an axon with under-segmentation errors. The decomposition of such an object requires first identifying its semantic components, and second, correcting the ambiguity at intersections of branches, i.e., decomposing the shape at intersecting branches. The mathematical details of the algorithm and its validation are described in a different manuscript[14], but we summarize the algorithm here:

Step 3.1.1: skeleton partitioning and critical points. We propose to decompose an object into its semantic-components using its curve skeleton and cross-sectional analysis. We determined the curve skeleton of the object $\Omega$ by applying the method from Hassouna & Farag[59]. This method also returns the collection of skeleton branches $\Gamma = \{\gamma_1, \ldots, \gamma_n\}$ (see Supplementary Fig. S1c). Based on $\Gamma$, CSD partitioned the curve skeleton of the object into $m \leq n$ maximal-length sub-skeletons denoted as $\psi_i, i = 1, \ldots, m$ (Supplementary Fig. S1d) via minimizing an orientation cost-function. Each sub-skeleton corresponds to a semantic-component. In other words, in this step, skeleton branches were merged if they were thought to belong to the same semantic component. To identify intersections of the semantic components, we analyzed the object cross-sections along intervals of $\psi$, called decomposition intervals. A decomposition interval occurs in the proximity of junction-points where skeleton branches connect, denoted as $j \in \psi$. The boundaries of a decomposition interval at $j$ were defined using the radius of the maximal inscribed ball at $j$, and two factors $\alpha_s = 10$ and $\alpha_e = 1.5$. At each decomposition interval, we search for a critical point. It is such a point that the object cross-section changes substantially; the normalized Hausdorff distance $H_\rho$[14] between a cross-sectional contour and the mean of visited cross-sectional contours exceeds $\theta_H = 0.85$ (Supplementary Fig. S1e). We cut the object at critical points to obtain object-parts. We assigned the same label to the object-parts along the same sub-skeleton to acquire a semantic-component (Supplementary Fig. S1f).

Step 3.1.2: object reconstruction. To reconstruct a semantic component at an intersection, we used generalized cylinders. We constructed a generalized cylinder $\Phi : \mathbb{R}^2 \to \mathbb{R}^3$ as a linear homotopy between two simple closed curves $C_{c_1}$ and $C_{c_2}$ elongated on a simple curve $\zeta \subset \mathbb{R}^3$. To define $C_{c_1}$ and $C_{c_2}$, let $x_{c_1}, x_{c_2} \in \psi$ be two critical points in the proximity of the joint $j$ on a sub-skeleton $\psi$. We cut the surface of the object $\partial\Omega$ at these two critical points. The cross-sectional curves of $\partial\Omega$ at $x_{c_1}$ and $x_{c_2}$ are $C_{c_1}$ and $C_{c_2}$. We defined $\zeta$ to be the spline interpolation of $\psi$ between $x_{c_1}$ and $x_{c_2}$. The acquired generalized cylinder was used to reconstruct the object at the intersection, which interior includes $j$ (Supplementary Fig. S1f).

Step 3.2: cell nuclei segmentation. DCNN-cN classifier was trained to generate a semantic segmentation of cell nuclei and the membrane of cell nuclei from the low-resolution datasets. SBEM imaging with $50 \times 50 \times 50$ nm$^3$ voxel size, however, might not fully resolve the membrane of cell nuclei. Thus, the binarization followed by the connected component analysis of DCNN-cN probability maps at sites where several cell nuclei appear adjacently would come with an under-segmentation error. Therefore, for the instance segmentation of cell nuclei, we devised an approach in a GDM framework based on active surface implemented with level sets[22,60]. Let the probability map of cell nuclei be denoted as $PR_n(x) : X \subset \mathbb{Z}^3 \to [0, 1]$, and the probability map of the membrane of cell nuclei as $PR_m(x) : X \subset \mathbb{Z}^3 \to [0, 1]$, where $x$ is a 3D coordinate belonging to the 3D image domain $X$. The position of the initial surface $\partial U$ was determined by binarizing the

curvilinear enhanced version of $PR_m$. We used Frangi filter[61] to enhance the curvilinear structures of $PR_m$, i.e., the membrane of cell nuclei, and suppress the background based on Hessian of $PR_m$ (see Supplementary Fig. S9). The Frangi filtered $PR_m$ is denoted as $\hat{PR}_m$. We subtracted $\hat{PR}_m$ from $PR_n$ and thresholded the result at $\theta_{nucleus} = 0.5$. This resulted in a binary image $I_N(x): X \to \{0, 1\}$, where $x \in X$ was a foreground voxel and $I_{(x)} = 1$ if $PR_n(x) - \hat{PR}_m(x) > \theta_{nucleus}$. The initial positions of deformable surfaces were determined using connected component analysis: $I_N$ was segmented to $n'$ individual cell nuclei denoted as $\hat{U}_i$, where $i = 1, \dots, n'$. At locations where several cell nuclei appear adjacently, the initial segmentation may include under-segmented cell nuclei. For that, the initial segmentation of cell nuclei required a further topological alteration. Therefore, the surface of each initially segmented cell nuclei underwent a primary deformation, where the surface was propagated for $-0.3\,\mu m$ using a 3D Euclidean distance transform. The inward propagation of the surfaces split the merged cell nuclei. Relabeling the split cell nuclei resulted in $U_i$, where $i = 1, \dots, n$ and $n \geq n'$. We used the volumes in which the number of voxels was greater than $1\,000$ as initial surfaces for a secondary deformation. The evolution of a surface started at $\partial U$ and stopped at the desired nucleus membrane because we defined the speed function equal to $PR_m$. Each initial surface $\partial U$ was deformed for 300 iterations, where the smoothing term of GDMs preserved the surface sphericity.

*Step 4: eliminating false positives*. The final step of the DeepACSON pipeline was to eliminate non-axonal and non-nucleus instances, i.e., false positives. We eliminated false positives from the segmentation of myelinated axons using an SVM with a quadratic kernel (as implemented in MATLAB, Statistics and Machine Learning Toolbox, *fitcsvm* function). For each component segmented as a myelinated axon, we computed several features (at one-third of the original size): volume, surface area, equivalent diameter, extent (the ratio of voxels in the component to voxels in the smallest bounding box of the component), solidity (the volume of the component divided by the volume of the convex hull of the component), and principal axis length (the length of the major axis of the ellipsoid that has the same normalized second central moment as the component). Features were standardized to the mean of zero and the standard deviation of one. To generate a training set, we randomly picked five datasets out of ten segmented large field-of-view datasets. From the selected datasets, we manually selected representative myelinated axons and non-axonal instances from both the cingulum and corpus callosum (1 097 myelinated axons and 189 non-axonal instances; less than 0.2% of the total number of components segmented as myelinated axons in all ten datasets). To determine the optimal SVM hyperparameters, the regularization parameter $C$ and kernel parameter $\sigma$, we selected the pair that minimized 5-fold cross-validation error on the training set using Bayesian optimization algorithm[62,63]. $C$ and $\sigma$ were constrained in the range $[10^{-6}, 10^6]$. The optimal parameters were $C = 1.12$ and $\sigma = 8.48$. The trained SVM was applied to eliminate non-axonal structures from all the datasets. We applied the same approach to eliminate false-positives from cell nuclei. For each component segmented as a cell nucleus, we calculated the same features (as described for myelinated axons) before and after the geometric deformation. To generate a training set, out of ten segmented large field-of-view datasets, we picked four random datasets. From the selected datasets, we manually selected representative cell nuclei and non-nucleus instances (263 cell nuclei and 113 non-nucleus instances; 7.3 % of the total number of components segmented as cell nuclei in all ten datasets) to train an SVM with a quadratic kernel. The optimal hyperparameters were found as described above, where $C$ was equal to 6.51, and $\sigma$ was equal to 4.17.

**Segmentation evaluation metrics**. We used precision $P$, recall $R$, and F1 score to compare an automated semantic segmentation to the ground truth, in such a way that there is a one-to-one match between voxels in the automated segmentation and ground truth. To define these measures, let $A$ and $B$ be the sets of voxels of a particular ultrastructure (myelin, myelinated axon, mitochondrion) in an automated segmentation and ground-truth, respectively. We defined $P = \frac{|A \cap B|}{|A|}$, and $R = \frac{|A \cap B|}{|B|}$, and F1 score as $F1 = 2 \times \frac{P \times R}{P + R}$. The maximum of the precision, recall, and F1 score is equal to one when the test segmentation perfectly matches the ground-truth.

Precision, recall, and F1 do not describe topological differences and are sensitive to small changes in the region boundary. Therefore, we evaluated automated segmentation with additional metrics that are less sensitive to small variations in boundary, but sensitive to topological differences. For that, we measured variance of information[64] (VOI, split and merge contributions) and Wallace indices[65]. We also computed adapted Rand error (ARE) as defined by the SNEMI3D contest (http://brainiac2.mit.edu/SNEMI3D/evaluation). We computed these metrics using the Gala library described by Nunez-Iglesias et al.[66]. The VOI metric is defined as the sum of the conditional entropies between two segmentations $VOI(A, B) = H(A|B) + H(B|A)$. The VOI metric is decomposed into VOI split $H(A|B)$ and VOI merge $H(B|A)$[66]. A lower VOI value indicates a better segmentation; for a perfect match between an automated segmentation and ground truth, we have VOI split = VOI merge = 0. The Wallace splitting index is defined as $\frac{a}{a+b}$ and the Wallace merging index is defined as $\frac{a}{a+c}$, where $a$ is the number of pairs of voxels in the input image that have the same label in $A$ and the same label in $B$, $b$ is the number

of pairs of voxels in the input image that have the same label in $A$ but a different label in $B$, $c$ is the number of pairs of voxels in the input image that have a different label in $A$ but the same label in $B$, and $d$ is the number of pairs of voxels in the input image that have different labels in $A$ and different labels in $B$. The Wallace indices take values between zero and one, where a higher value indicates a better segmentation. The Rand index is defined as $\frac{a+d}{a+b+c+d}$, where $a$, $b$, $c$, and $d$ are as in Wallace indices, and adapted Rand error is defined as one minus the maximal F-score of the Rand index[67] (excluding the zero component of the original labels). The adapted Rand error takes values between zero and one, where a lower value indicates a better segmentation. As Nunez-Iglesias et al.[66] argued, the VOI metric has several advantages over the Rand index and is a better metric for comparing EM segmentation results. For example, errors in the VOI scale linearly with the error size, whereas the Rand index scales quadratically, making VOI more directly comparable between volumes than the Rand index. Also, the Rand index has a limited useful range near one, and that range is different for each image. In contrast, VOI ranges between zero and $log(K)$, where $K$ is the number of objects in the image.

**Statistics and reproducibility**. For the statistical hypothesis testing between the sham-operated and TBI animals, we subjected the measurements to the nested (hierarchical) 1-way ANOVA separately for each hemisphere[16]. The nested ANOVA tests whether there exists a significant variation in means among groups, or among subgroups within groups. The myelinated axons were nested under the rats' ID, and the rats' ID was nested under the groups (sham-operated and TBI). The rats' IDs were treated as random effects, and the group was treated as a fixed effect. We set the alpha-threshold defining the statistical significance as 0.05 for all analyses. All sample sizes are specified within the manuscript or in figure legends.

**Reporting summary**. Further information on research design is available in the Nature Research Reporting Summary linked to this article.

## Data availability
The datasets generated during and/or analyzed[68] in the current study are publicly available at https://etsin.fairdata.fi/dataset/f8ccc23a-1f1a-4c98-86b7-b63652a809c3. The source data files underlying the graphs of the main figures are available through Supplementary Data 1-4: Fig. 5d: Supplementary Data 1-2; Fig. 5e-f: Supplementary Data 3; and Fig. 6: Supplementary Data 4. All other data that support the findings of the present study are available from the corresponding author upon request.

## Code availability
The source code of DeepACSON software[69] is publicly available at https://github.com/aAbdz/DeepACSON.

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

## Acknowledgements

This work was supported by the Academy of Finland (grant #316258 to J.T. and grant #323385 to A.S.) and Biocenter Finland and the University of Helsinki (I.B., E.J., and SBEM imaging). We would like to thank Maarit Pulkkinen for her help with the animal handling, and Mervi Lindman and Antti Salminen for help preparing the samples for SBEM. We would like to thank Andrea Behanova for developing gACSON, the segmentation evaluation software. The authors acknowledge CSC-IT Center for Science, Finland, for computational resources.

## Author contributions

A.A., J.T. and A.S. conceived the project and designed the study. A.A. implemented the methods and performed the experiments. A.A., J.T. and A.S. analyzed the data. I.B. and E.J. contributed to electron microscopy imaging. A.A., J.T. and A.S. wrote the manuscript. All authors commented on and approved the final manuscript.

## Competing interests

The authors declare that they have no conflict of interest.
