## [Peer Review File · Communications Biology]

Reviewers' comments:

Reviewer #1 (Remarks to the Author):

This paper presents a pipeline for automated segmentation of ultrastructures in 3D EM images of neuronal structures. The authors have proposed a top-bottom pipeline to perform the segmentation as compared to previously proposed methods that involve over-segmentation and region merging.

The proposed research lacks novelty as follows:

- The authors claim that the previously proposed methods (bottom-up design) require sophisticated training and rely on a considerable amount of manual interaction. However, the proposed method also involves sophisticated steps and the computational complexity of the proposed method has not been compared to the state-of-the-art methods.
- The proposed network architecture involves a limited number of parameters, as highlighted in the paper. Considering the bits of data used for training such a shallow network that does not even include residual blocks, the proposed network becomes highly prone to underfitting and hence weak inference results. This research lacks ablation studies.
- The authors have stressed out deep learning in the proposed study; however, the results primarily rely on post-processing and demonstrates less dependency on deep learning as the proposed network is vulnerable to underfitting.
- The evaluation metrics are limited, and the choice of thresholding values for binarising the images remains vague. Potential drawbacks of downsampling the high-resolution images also remain uninvestigated.
- The authors have split the training and test data by 0.4 and 0.6 ratios, respectively. However, the random sampling for testing purposes involves very few patches extracted randomly, and this could lead to highly biased inference results (please check the sample sizes).
- The authors have not compared the computational complexity of the proposed method to that of the state-of-the-art methods and have just provided the timing of the proposed pipeline.
- The authors have referenced some of the state-of-the-art methods in the discussion section. However, the comparisons made in that section seems to be subjective rather than objective as the discussion section lacks justification of the claims that the authors have made.

In summary, I think this paper provides very limited novelty to the field and needs much comprehensive analysis and justification of the claims made in the paper.

Reviewer #2 (Remarks to the Author):

A formatted version of response is also attached.

Summary

DeepACSON is a segmentation method that utilizes segmented data from a previous method, ACSON, to train a network for segmenting lower resolution data. By reducing the size of the

original data along with the segmented data from ACSON, DeepACSON can be trained with this “ground truth” data. The authors claim two major advantages of this technique:

1. Less human interaction, “providing human-annotation-free training sets.”
2. Being able to use lower resolution and thus larger field of view datasets.

Community Impact

This method can be useful to the neuronal community that uses EM data for their studies. The ability to take a single high-resolution sample to train from and then use low-resolution there after could allow for higher throughput. A much larger section of brain matter could be imaged quickly and more reliably at this lower resolution.

Originality

This method is very close to previous methods mentioned in the Introduction. However, DeepACSON differs enough to be called an improvement or at least a hybrid of prior work.

Conceivability

It would be of interest to the reader to have timing comparison of this method to some of the others. Five days to segment a single image seems to be a lot. But maybe, this needs to be in the context of volume per hour or something comparable.

This manuscript emphasizes imaging speed. However, it is not clear how much is gained without knowing if this method is just as fast computationally as others? In the discussion section it is mentioned that it takes 24 hours to train on the high resolution data using ACSON. It would be beneficial for the authors to comment on where most of the computational time is spent, clarify the timing from start to finish. Meaning, break out the training phase from the high resolution, training on low resolution, and running final segmentation.

It is mentioned that training sets were “augmented using randomized histogram distortions...” to make this method more robust to changes in contrast and brightness. How does this perform in practice with different sample preparation? Were there samples made by more than one person?

It is not clear why a new GUI was necessary to write for validation. How do the authors ensure the GUI is not biasing the results in any way?

The simplification of mitochondria to a centroid seems ill advised to measure distribution. Take for instance two large mitochondria almost touching. These centroids can be equi-distance to that of two small mitochondria. However, the two large mitochondria would seem more dense than the two small ones... I think minimum distance between mitochondria, or even the coverage of mitochondria projected on the skeleton would be a more informative metric. Alternatively, the centroid distance can be useful if the variance of mitochondria length is small enough. In which case, reporting this variance would be sufficient.

It is mentioned that the volumes of the cells could not be directly compared because of variation between samples. Is there a reason for this variation? Can it not be normalized out in some manner?

In the discussion section, it is not clear what the authors mean by “error-aware” in the second paragraph.

My largest concern is that the source code has not been made available to the reviewers.

Recommendation

Over all this paper is well written. The benefits of this method if adopted should increase throughput of neuron tracing in large field of views. I would recommend that, after minor changes relating to the comments above and the release of the code for review, this manuscript be accepted.

Reviewer #3 (Remarks to the Author):

- Introduction and motivation

The reason for doing the image segmentation on the low-resolution images (using learning) is not well or clearly motivated. The introduction has a list of relevant studies with some trivialised details, but lacks distillation and categorisation of the related work.

The introduction has not summarised so well about the existing works: what are the limitations and how this proposed new approach is motivated? Compared with existing techniques, the paper's unique contribution is vague, because why going for low-resolution images is not elaborated clearly, especially the paper mentioned that the high-resolution images are actually available. (only a few sentences about the rationale of using low and high-resolution images are stated much later in the discussion session).

The paper uses bottom-up and top-down design as the standard to classify different learning techniques, which is a debatable way of differentiating their own work from the previous works. In essence, the bottom-up approach uses the local features to detect or represent semantic information - this is the case no matter how CNN is used (referring to thousands of papers from the computer vision), because this is how neural network works, since NNs have no parameterized model to represent the Marco information, and this brings the following statement into a question: "This approach is made possible by the use of a-priori knowledge of the topology of myelinated axons and cell nuclei.". How the typological information is used as prior knowledge? The best it can do is the pre-training of networks given enough datasets, which is different from the "knowledge" in normal human's definition.

- methodology

"The simultaneous high- and low-resolution imaging enables to use the high-resolution images as the training data to segment low-resolution datasets."

There is a one-to-one correspondence between high res and low-resolution images, ie both of them record the same spatial-temporal information, and the low-resolution images *are* down-sampled from the very same higher resolution ones. Of course, using the high-resolution images as the ground truth for training the low-resolution datasets down-sampled from themselves, you can always get very good fitting results with minimal loss - using simply brute force supervised learning, for example.

- computational advantage

There's not yet information to compare the processing time for the low-resolution images vs the processing time of the higher resolution images *directly*. Note that most of the computational time is in (1) BM4D filtering and (2) CSD SVM code, which is not related with the CNNs being the selling point of the paper.

Computation time (The pipeline required about five days to segment one raw low-resolution SBEM dataset of size $4\,000 \times 2\,000 \times 1\,200$ voxels into its final segmentation. The details are presented in Supplementary Table S4)

Supplementary material gives the specification of CPU, GPU and time. There are several technical questions: 1 the pipeline seems not automatically because it's a mixture of MATLAB code and Python code; 2 the hurdle of the computational time is limited primarily by the MATLAB computing which is known to be slow, and the second is the CSD Python code (despite python is popular in machine learning but it is too slow compared to C++, and especially if the CSD Python code is not optimised). In short, if the whole codebase is all written in Python class and heavily optimised, the speed should be much faster.

Even regardless of all these technical aspects, when it comes to the central motivation of the paper, what is the indication of this computation time? What it supports? Because it seems to possess such low-resolution images is also very slow (5 days). The computational time usually can be resolved by parallel computing, change/optimize the programming code, use clusters instead of PCs. Therefore the justification of this approach is not clear.

- motivation of using low-resolution images (especially down-sampled from existing high resolution ones)

The author also clearly states that "We segmented the high-resolution datasets using our earlier automated ACSON pipeline⁴. We used the ACSON segmentation of the high-resolution datasets to train DeepACSON, eliminating the need for manually annotated training sets." and "We acquired SBEM images of the white matter simultaneously at low- and high-resolution".

So indeed, high-resolution datasets are available, which means that a lot of existing methods can be used directly to analyse as well, and therefore what is the motivation of using down-sampled low-resolution images?

Also, regarding "We used the ACSON segmentation of the high-resolution datasets to train DeepACSON", we know that the training of deep neural network always have artifacts, where the tests are different from training sets. So how the authors take that into account? (also see a separate comment regarding support vector machine later).

In the Essence, the key scientific fundamental is that a low-resolution image, no matter original or down-sampled, loses important features and information, therefore to acquire truthful high-resolution data at the first place is the reason for scientists to improve high-resolution imaging techniques - that is a whole purpose, because no matter humans or trained CNNs, all the extracted information and understanding are all *probabilistic* based on limited information because of lack of certainty.

- Datasets

In addition to the description, Please provide a table to clearly show what are the datasets resolved in both low and high resolution, which are only resolved only in high resolution. A table is much clearer to read.

"the remaining six datasets were reserved for testing", and "six unseen high-resolution SBEM datasets labeled by the automated ACSON pipeline"

It is a small number of datasets, which are not sufficient to have valid statistical conclusions. Is there a more indicated number, for example, the number of images?

- Evaluation

In that section "White matter 3D morphology analysis", there are descriptions of how these measurements are quantified, but there's no information about how this process of measurement is done. The question is that: is this procedure done manually, for automatically? This is important because a manual procedure provides limited data and therefore the statistical indication is compromised. Instead, an automated procedure or benchmark can process a large quantity of data, yielding much better statistics. (Computer vision algorithms of object recognition have a well-established image bank benchmark to quantify.)

Also, it is not convincing that the evaluation using 1 human expert is conclusive, and we don't know how much the human expert is biased due to other factors (conflict of interest, benefits), despite an expert has no access to the info.

- CSD SVM

There are improvements that can be made regarding the relationship between the artefacts from the CSD composition algorithm using support vector machine and the overall contribution of DeepACSON. There should be more discussion about ambiguities and artefacts processed from CSD the algorithm.

The trained SVM is manually designed and tuned (specifically adapted parameters), which do not generalise to corner cases. This makes step 4 "eliminating false positives" less robust for processing different new datasets.

- Discussion section

Many part of the discussion section actually is doing the job of the introduction section. Instead, the discussion should concentrate more on the conclusive findings and further implications.

- English Writing and language usage

Some examples of imprecise wording: "The purple panel shows a cell nucleus from the low-resolution dataset (a), which membrane is resolved, but not continuously."; "The segmentation of myelinated axons and cell nuclei was finalized by eliminating non-axonal and non-nucleus structures with support vector machines (SVMs)."; "We evaluated the performance of the two SVMs by a leave-one-group-out (LOGO) cross-validation, where the classifier was trained excluding the data from one group of animals (sham-operated or TBI) from training and evaluated against it (Supplementary Table S2).

Word usage: use "where" instead of "which"; use 'using' instead of 'with'; avoid using several clauses and put 'it' at the end of a long sentence (what exactly this "it" is pointing to?).

The paper overall needs significant re-writing and requires proofreading from native speakers and senior academics.

DeepACSON: Automated Segmentation of White Matter in 3D Electron Microscopy

Ali Abdollahzadeh, Ilya Belevich, Eija Jokitalo, Alejandra Sierra, and Jussi Tohka

Manuscript COMMSBIO-20-0173-T

Summary of the major changes applied to the manuscript in response to the reviewers' comments:

- We compared our technique, DeepACSON, to state-of-the-art, deep learning-based, automated techniques in the segmentation of three-dimensional electron microscopy (3D-EM) images: 1) DeepEM2D, 2) DeepEM3D, and 3) Flood filling network (FFN). We showed that DeepACSON outperforms these techniques in the segmentation of low-resolution EM datasets of white matter. We also compared the computation time of DeepACSON to these techniques. The computation time of DeepACSON is shorter than FFN but longer than DeepEM2D and DeepEM3D.
- We performed an ablation study on the DeepACSON pipeline to evaluate the effect of:
 - utilizing a deeper architecture (a U-Net with residual modules) on the semantic segmentation. We found that this deeper network is prone to overfitting, and for this reason, the segmentation was more accurate with the original architecture in terms of the evaluation metrics.
 - BM4D denoising, as a pre-processing step, on the semantic segmentation. We found that dropping the BM4D denoising step made segmentations less accurate.
- We clarified our motivation for low-resolution EM imaging of large tissue volumes, compared to high-resolution EM imaging of small tissue volumes, as a technique to increase the field-of-view at a reasonable usage of resources.

Reviewers' comments:

Reviewer #1:

Reviewer summary: This paper presents a pipeline for automated segmentation of ultrastructures in 3D EM images of neuronal structures. The authors have proposed a top-bottom pipeline to perform the segmentation as compared to previously proposed methods that involve over-segmentation and region merging.

Comment 1.1: The authors claim that the previously proposed methods (bottom-up design) require sophisticated training and rely on a considerable amount of manual interaction. However, the proposed method also involves sophisticated steps and the computational complexity of the proposed method has not been compared to the state-of-the-art methods.

Answer 1.1: We agree with the reviewer that DeepACSON required comparisons with existing state-of-the-art segmentation methods. DeepACSON is a fully automated segmentation technique, requiring no human interaction, and even provided with a human-annotation-free training set. As suggested, we compared DeepACSON against state-of-the-art automated segmentation techniques, DeepEM2D⁸, DeepEM3D⁸, and FFN¹⁰, and showed that our method outperforms these methods in terms of accuracy. Also, in the computation time section of the manuscript, we now included the computation-complexity of deepACSON for the BM4D¹⁵ denoising step, cylindrical shape decomposition algorithm¹⁴, and convolutional neural networks (CNNs) in generating semantic segmentations. We also compared the computation time of DeepACSON to DeepEM2D, DeepEM3D, and FFN. The computation time of DeepACSON is shorter than FFN but longer DeepEM2D and DeepEM3D. We added this information to the Results section on **page 11-13, line 217-235**, and also in the **panel g of the new Figure 6**.

Computation time

The pipeline required about five days to segment a raw, low-resolution SBEM dataset of 4000×2000×1200 voxels into its final segmentation. The details are presented in Supplementary Table S4. Approximately 40% of the DeepACSON computation time was spent during BM4D denoising¹⁵. We run BM4D filtering on non-overlapping patches of the SBEM volumes to enable parallel processing. We remark that BM4D filtering can be dropped from the pre-processing steps to improve the computation time, but as shown in Fig. 6d-f, dropping BM4D filtering made segmentations less accurate. Approximately 30% of the DeepACSON computation time was spent on the CSD algorithm. In more detail, the time complexity of the sub-voxel precise skeletonization is $O(n N_{\Omega} \log N_{\Omega})$, where n is the number of skeleton branches, and N_{Ω} is the number of voxels of a discrete object, i.e., a myelinated axon. The $N_{\Omega} \log N_{\Omega}$ factor is from the fast marching algorithm²¹. The time complexity to determine a critical point is $O(N_p)$, where N_p is the number of inquiry points to check for the cross-sectional changes in a decomposition interval. Therefore, the overall time complexity of the CSD algorithm is $O(n N_{\Omega} \log N_{\Omega}) + O(N_p)$. Approximately 10% of the DeepACSON computation time was spent on the semantic segmentation, with time complexity of $O(N)$, where N is the number of voxels in an SBEM dataset.

We also compared the computation time of DeepACSON, DeepEM2D, DeepEM3D, and FFN techniques (see Fig.6g for results). These techniques were compared over the six test datasets on a computer with an NVIDIA Tesla V100-32 GB GPU, 2 × Intel Xeon E5 2630 CPU 2.4 GHz, and 512 GB RAM. DeepEM2D and DeepEM3D had the shortest computation time (about 1 minute) as the segmentation mainly relies on an Inception-ResNet-v2 network²² and watershed segmentation. FFN required the longest computation time for an end-to-end segmentation (about 28 minutes). DeepACSON required about 4 minutes (using 15 CPU cores) to segment the test datasets, which was longer than DeepEM2D/3D and shorter than FFN.

Figure 6. DeepACSON evaluations. Comparison of DeepACSON against state-of-the-art segmentation, DeepEM2D, DeepEM3D, and FFN, by (a) variation of information (VOI, split and merge contribution, lower is better), (b) Wallace indices (split and merge contribution, higher is better), and (c) adapted Rand error (ARE, lower is better) and the sum of VOI split and VOI merge (VOI sum, lower is better). DeepACSON outperformed these techniques as it produced the smallest VOI split, VOI merge, VOI sum, and ARE and the biggest Wallace split and merge values. Comparison of the design parameters of DeepACSON: standard DeepACSON (DeepACSON-A), a U-Net with residual modules (DeepACSON-B), the effect of BM4D denoising (DeepACSON-C), and adjusting the resolution between the training and test sets (DeepACSON-D) over (d) VOI (split and merge contribution) (e) Wallace indices (split and merge contribution), and (f) ARE and VOI sum. The filled circles and error bars show the mean and standard deviation of the evaluations, respectively. The dash-dotted lines show the choice of binarization threshold. The comparisons were run over the best threshold, i.e., smallest VOI merge and VOI split. (g) Comparing the computation time of DeepACSON against DeepEM2D/3D and FFN (mean ± standard deviation). All comparisons were run over six test SBEM datasets of size 290×290×285 voxel³ automatically segmented using the ACSON⁵ pipeline.

References:

5. Abdollahzadeh, A., Belevich, I., Jokitalo, E., Tohka, J. & Sierra, A. Automated 3D Axonal Morphometry of White Matter. *Sci. Reports* **9**, 6084 (2019).
8. Zeng, T., Wu, B. & Ji, S. DeepEM3D: approaching human-level performance on 3D anisotropic EM image segmentation. *Bioinforma. (Oxford, England)* **33**, 2555–2562 (2017).
10. Januszewski, M. et al. High-precision automated reconstruction of neurons with flood-filling networks. *Nat. Methods* **15**, 605–610 (2018).
14. Abdollahzadeh, A., Sierra, A. & Tohka, J. Cylindrical shape decomposition for 3D segmentation of tubular objects. *arXiv:1911.00571v2 [cs.CV](2019)*.
15. Maggioni, M., Katkovnik, V., Egiazarian, K. & Foi, A. Nonlocal Transform-Domain Filter for Volumetric Data Denoising and Reconstruction. *IEEE Transactions on Image Process.* **22**, 119–133 (2013).
21. Sethian, J. A. A fast marching level set method for monotonically advancing fronts. *Proc. Natl. Acad. Sci.* **93**, 1591–1595 (1996).
22. Längkvist, M., Karlsson, L. & Loutfi, A. A review of unsupervised feature learning and deep learning for time-series modeling. *Pattern Recognit. Lett.* **42**, 11–24 (2014).

Comment 1.2: The proposed network architecture involves a limited number of parameters, as highlighted in the paper. Considering the bits of data used for training such a shallow network that does not even include residual blocks, the proposed network becomes highly prone to underfitting and hence weak inference results. This research lacks ablation studies.

Answer 1.2: We now performed an ablation study on the DeepACSON pipeline, as suggested by the reviewer. We utilized a deeper architecture, a U-Net with residual modules, for the semantic segmentation. We used a ResNet-34, pre-trained on the ImageNet dataset, as the encoder of the U-Net. We found that a deeper network is prone to overfitting, hence generating less accurate segmentation. We evaluated the effect of BM4D denoising, as a pre-processing step, on the semantic segmentation. We found that dropping the BM4D denoising step made segmentation less accurate. We also evaluated the effect of down-sampling of the training set to the resolution of the low-resolution images. We found that adjusting the resolution for training is essential, and dropping this step worsens the evaluation measures. We demonstrated the choice of binarization thresholds over the variation of information (split and merge errors) metric. We added this information in Results, Materials and Methods, and Supplementary Information as follows:

Please, see Fig. 6d-f (Answer 1.1) for the results of the ablation study.

Results, page 10-11, line 189-199:

We also evaluated the DeepACSON pipeline to understand its design parameters, such as denoising, resolution adjustment, and an alternative, deeper architecture. In Fig.6d-f, we denoted the standard DeepACSON design as DeepACSON-A, which used a light fully convolutional network¹⁸ (FCN) for the semantic segmentation. The

standard DeepACSON was trained using down-sampled and BM4D filtered volumes. We replaced the FCN design of the standard DeepACSON with a U-Net¹⁹ with residual modules, denoted as DeepACSON-B in Fig.6d-f. In this figure, we also show the effect of omitting BM4D¹⁵ denoising as a pre-processing step (DeepACSON-C) and down-sampling the high-resolution images to generate the training set (DeepACSON-D). In addition, we demonstrated the choice of thresholds at which the probability maps were binarized. Evaluations were run over the six SBEM volumes. The comparisons showed that the standard DeepACSON performed better than a deeper network, which was prone to over-fitting. Denoising the train/test datasets as a pre-processing step improved our results as did adjusting the resolution between the training and test sets. We binarized the probability maps at thresholds, which generated the smallest VOI split/merge values.

Materials and Methods, page 18-19, line 412-431:

For the semantic segmentation of myelinated axons, we also tested a U-Net¹⁹ with residual modules, implemented in PyTorch⁵³. For that, we used a ResNet-34⁵⁴ with about 21 million parameters, pre-trained on the ImageNet dataset⁵⁵, as the encoder of the U-Net (Supplementary Fig.S5). In the encoding path of the U-Net, the height and width of the feature maps were halved, and the depth of the feature maps was doubled. In the decoding path of the U-Net, the height and width of the feature maps were doubled, and the depth of the feature maps was halved. The basic residual blocks of ResNet-34 were constructed on 3×3 convolutional layers, rectified linear unit (ReLU) as the activation function, and batch normalization⁵⁶. The basic decoding blocks applied nearest-neighbor interpolation, up-sampling the feature maps to recover the spatial resolution of input images. Feature maps generated in the encoding path were concatenated to the corresponding feature maps in the decoding path by the skip connections. The model was trained by the softmax function and cross-entropy loss function. We set the batch size equal to 8 and feed every three SBEM images with stride 1, in z-direction, as input. The training set was normalized to have the mean of (0.485, 0.456, 0.406) and the standard deviation of (0.229, 0.224, 0.225) in (x, y, z) directions. Also, half of the training set was augmented by geometrical transformation, such as horizontal and vertical flips, scaling, shifting, and rotation. For the optimization, we used Adam optimizer⁵² and set its initial learning rate $\alpha = 1 \times 10^{-4}$, the exponential decay rate for the first moment $\beta_1 = 0.9$, the exponential decay rate for the second-moment $\beta_2 = 0.999$, and the weight decay = 1×10^{-5} (we use the same notation as in ⁵²). We set the class weights to 0.3, 0.3, 0.3, and 1 for the background, myelin, myelinated axons, and mitochondria, respectively. We trained this U-Net architecture with the same datasets as mentioned for the FCN network for case scenarios: 1) the training set was BM4D denoised and down-sampled, 2) the training set was not BM4D denoised but down-sampled, 3) the training set was BM4D denoised but not down-sampled. In the first case, we evaluated the effect of using a deeper network than our FCN design, and in the second and third cases, we evaluated the effect of BM4D denoising and down-sampling the high-resolution datasets of the training set.

Materials and Methods, page 21-22, line 512-534

Segmentation evaluation metrics. We used precision P , recall R , and $F1$ scores to compare an automated segmentation to the ground truth, as there is a one-to-one match between voxels constituting the segmentations. To define these measures, let A and B be the sets of voxels of a particular ultrastructure (myelin, myelinated axon, mitochondrion) in an automated segmentation and ground-truth, respectively. We defined $P = \frac{|A \cap B|}{|B|}$, and $R = \frac{|A \cap B|}{|A|}$, and $F1$ score as $F1 = 2 \times \frac{P \times R}{P + R}$. The maximum for the precision, recall, and $F1$ score is equal to one when the test segmentation perfectly matches the ground-truth. However, these metrics do not describe topological differences, and they are sensitive to small changes in the region boundary. Therefore, we evaluated automated segmentation with metrics less sensitive to small variations in boundary, but sensitive to topological differences. For that, we measured variance of information⁶² (VOI, split and merge contributions) and Wallace indices⁶³ to account for merge and split errors of a segmentation separately. We also computed adapted Rand error (ARE) as defined by the SNEMI3D contest (<http://brainiac2.mit.edu/SNEMI3D/evaluation>) as 1 minus the maximal F -score of the Rand index⁶⁴ (excluding the zero component of the original labels). We performed these evaluations using the Gala library described by Nunez-Iglesias et al⁶⁵. The VOI metric is defined as the sum of the conditional entropies between two segmentations $VOI(A,B) = H(A|B) + H(B|A)$, where A is the automated segmentation, and B is the ground-truth. The VOI metric is decomposed into VOI split $H(A|B)$ and VOI merge $H(B|A)$ ⁶⁵. A lower VOI value indicates a better segmentation; for a perfect match between an automated segmentation and ground truth, we have VOI split = VOI merge = 0. Also, the Wallace splitting index is defined as $\frac{a}{a+b}$ and the Wallace merging index is defined as $\frac{a}{a+c}$, where a is the number of pairs of voxels in the input image that have the same label in A and the same label in B , b is the number of pairs of voxels in the input image that have the same label in A but a different label in B , c is the number of pairs of voxels in the input image that have a different label in A but the same label in B , and d is the number of pairs of voxels in the input image that have different labels in A and different labels in B . The Wallace indices are between 0 and 1, where a higher value indicates a better segmentation, i.e., less split and merge compared to the ground truth. The Rand index is defined as $\frac{a+d}{a+b+c+d}$, where a , b , c , and d are defined the same as in the case of Wallace indices. Note that we applied the adapted Rand error, where a lower value indicates a better segmentation.

Supplementary Information, page v:

Figure S5. A U-Net¹⁹ architecture with residual modules. We used a ResNet-34⁵⁴, pre-trained on the ImageNet dataset⁵⁵, as the encoder of the U-Net. Where the size of the feature maps was halved, the depth was doubled, and where the size of the feature maps was doubled, the depth was halved. The basic residual blocks of ResNet-34 were constructed on 3×3 convolutional layers, rectified linear unit (ReLU) as the activation function, and batch normalization⁵⁶ (BN). The basic decoding blocks applied nearest-neighbor interpolation, up-sampling the feature maps to recover the spatial resolution of input images. Feature maps generated in the encoding path were concatenated to the corresponding feature maps in the decoding path by the skip connections. The model was trained by the softmax function and cross-entropy loss function.

References:

5. Abdollahzadeh, A., Belevich, I., Jokitalo, E., Tohka, J. & Sierra, A. Automated 3D Axonal Morphometry of White Matter. *Sci. Reports* **9**, 6084 (2019).
15. Maggioni, M., Katkovnik, V., Egiazarian, K. & Foi, A. Nonlocal Transform-Domain Filter for Volumetric Data Denoising and Reconstruction. *IEEE Transactions on Image Process.* **22**, 119–133 (2013).
18. Long, J., Shelhamer, E. & Darrell, T. Fully Convolutional Networks for Semantic Segmentation. *2015 IEEE Conf. on Comput. Vis. Pattern Recognit. (CVPR)*(2015).
19. Ronneberger, O., Fischer, P. & Brox, T. U-Net: Convolutional Networks for Biomedical Image Segmentation. In Navab, N., Hornegger, J., Wells, W. & Frangi, A. (eds.) *MICCAI 2015*, 234–241 (Springer International Publishing, Cham, 2015).
52. Kingma, R. & Prodanova, K. ADAM: A Method for Stochastic Optimization. *AIP Conf. Proc.* **1631**, 58–62597(2014).
53. Paszke, A. et al. PyTorch: An Imperative Style, High-Performance Deep Learning Library. In Wallach, H. et al. (eds.) *Advances in Neural Information Processing Systems* **32**, 8024–8035 (Curran Associates, Inc., 2019).

54. He, K., Zhang, X., Ren, S. & Sun, J. Deep residual learning for image recognition. *Proc. IEEE Comput. Soc. Conf. on Comput. Vis. Pattern Recognit.* 2016-Decem, 770–778 (2016).

55. Jia Deng et al. ImageNet: A large-scale hierarchical image database. In 2009 IEEE Conference on Computer Vision and Pattern Recognition, 248–255 (IEEE, 2009).

56. Ioffe, S. & Szegedy, C. Batch normalization: Accelerating deep network training by reducing internal covariate shift. *32nd Int. Conf. on Mach. Learn. ICML 2015*, 448–456 (2015).

62. Meilă, M. Comparing Clusterings by the Variation of Information. In Schölkopf, B. & Warmuth, M. (eds.) *Learning Theory and Kernel Machines. Lecture Notes in Computer Science*, 173–187 (Springer, Berlin, Heidelberg, 2003).

63. Wallace, D. L. A Method for Comparing Two Hierarchical Clusterings: Comment. *J. Am. Stat. Assoc.* **78**, 569 (1983).

64. Rand, W. M. Objective Criteria for the Evaluation of Clustering Methods. *J. Am. Stat. Assoc.* **66**, 846 (1971).

65. Nunez-Iglesias, J., Kennedy, R., Parag, T., Shi, J. & Chklovskii, D. B. Machine learning of hierarchical clustering to segment 2D and 3D images. *PLoS ONE* **8** (2013).

Comment 1.3: The authors have stressed out deep learning in the proposed study; however, the results primarily rely on post-processing and demonstrates less dependency on deep learning as the proposed network is vulnerable to underfitting.

Answer 1.3: We agree with the reviewer that we addressed the under-segmentation error, which is a hard-to-correct topological error. As shown in **Answer 1.2**, we now tested a deeper network, a U-Net with residual modules. We used a ResNet-34, pre-trained on the ImageNet dataset, as the encoder of the U-Net (**Supplementary Fig. S5**). Our results showed that using a deeper network was prone to overfitting in this application and deteriorated the evaluation measures.

Comment 1.4: The evaluation metrics are limited, and the choice of thresholding values for binarising the images remains vague. Potential drawbacks of downsampling the high-resolution images also remain uninvestigated.

Answer 1.4: We now expanded the evaluation section of the manuscript and applied the variation of information (split and merge contribution), Wallace indices (split and merge contribution), and adapted Rand error to compare DeepACSON with state-of-the-art segmentation techniques and perform an ablation study on the DeepACSON pipeline. In our ablation study, we evaluated the effect of down-sampling the high-resolution datasets to the resolution of the test volumes and the choice of binarization thresholds. We trained

DeepACSON once with down-sampled EM volumes and once with EM volumes at their original resolution. We showed that dropping this resolution adjustment step generated less accurate segmentation. Please, see **Answer 1.2** for details about DeepACSON ablation studies.

Comment 1.5: The authors have split the training and test data by 0.4 and 0.6 ratios, respectively. However, the random sampling for testing purposes involves very few patches extracted randomly, and this could lead to highly biased inference results (please check the sample sizes).

Answer 1.5: We differentiate between two test sets used to evaluate the DeepACON pipeline: 1) a test set that comprised six high-resolution EM volumes down-sampled to the resolution of low-resolution images. We applied this test set to compare DeepACSON against state-of-the-art automated segmentation methods using VOI, Wallace indices, and adapted Rand error. Note that each EM volume included approximately 300 axons, and thus we evaluated DeepACSON on approximately $6 \times 300 = 1800$ myelinated axons; 2) a test set, which comprised 50 patches of size 300×300 voxels. The patches were acquired from the ten low-resolution datasets (large field-of-view datasets), five patches per dataset. This test set was used only for expert evaluations. Note that each patch, on average, included approximately 130 axonal cross-sections and 30 mitochondria. Therefore, the expert has evaluated about 6500 axonal cross-sections and 1500 mitochondria in total, providing robust insight into the segmentation accuracy. The expertise of A.S. (from the authors' list) in microscopic tissue information and being a volunteer in contributing her time for the evaluation were the reasons that convinced us to finalize the evaluation section with an expert subjective point-of-view. We added this information in the Evaluation section on **page 10, line 165-176**.

Evaluations

We used two test sets to evaluate the DeepACON pipeline: 1) a test set that comprised six high-resolution SBEM volumes down-sampled to the resolution of low-resolution images. We applied this test set to compare DeepACSON against state-of-the-art automated segmentation methods and perform an ablation study on the DeepACSON pipeline. Labels for this test set was provided automatically using ACSON⁵ pipeline and proofread by A.S. In this test set, each SBEM volume included approximately 300 axons, and thus we evaluated DeepACSON on approximately $6 \times 300 = 1800$ myelinated axons; 2) a test set, which comprised 50 patches of size 300×300 voxels only for the expert evaluations. We randomly sampled every low-resolution (large field-of-view) dataset for five non-overlapping windows of size 300×300 voxels (10 datasets, 50 samples). Each patch, on average, included approximately 130 axonal cross-sections and 30 mitochondria. Therefore, the expert has evaluated about 6500 axonal cross-sections and 1500 mitochondria in total. The expert had no access to the dataset ID nor the sampling location. The expert evaluated the sampled images of the final segmentation by counting the number of true-positives (TP), false-positives (FP), and false-negatives (FN).

Reference:

5. Abdollahzadeh, A., Belevich, I., Jokitalo, E., Tohka, J. & Sierra, A. Automated 3D Axonal Morphometry of White Matter. *Sci. Reports* 9, 6084 (2019).

Comment 1.6: The authors have not compared the computational complexity of the proposed method to that of the state-of-the-art methods and have just provided the timing of the proposed pipeline.

Answer 1.6: As suggested by the reviewer, we now included the computation complexity of DeepACSON, and compared the computation time of our method to state-of-the-art automated segmentation techniques, as discussed in **Answer 1.1**.

Comment 1.7: The authors have referenced some of the state-of-the-art methods in the discussion section. However, the comparisons made in that section seems to be subjective rather than objective as the discussion section lacks justification of the claims that the authors have made. In summary, I think this paper provides very limited novelty to the field and needs much comprehensive analysis and justification of the claims made in the paper.

Answer 1.7: As suggested by the reviewer, we now expanded the Evaluation section in the Results section, and compared our method to state-of-the-art automated segmentation techniques, DeepEM2D, DeepEM3D, and FFN. We scored these techniques over the variation of information (split and merge), Wallace indices (split and merge), and adapted Rand error. We showed that DeepACSON outperforms state-of-the-art automated techniques in the segmentation of ultrastructures in low-resolution datasets of white matter. This information is now included in Results and Materials and Methods.

Please, see Fig.6a-c in Answer 1.1 for the comparison results.

Please, see “segmentation evaluation metrics” in Answer 1.2.

Results, page 10, line 177-188:

We compared DeepACSON with state-of-the-art segmentation techniques; DeepEM2D⁸ and DeepEM3D⁸, which rely on a precise semantic segmentation, and FFN¹⁰, which accounts for the shape of neural processes during the instance segmentation. We trained DeepEM3D⁸ using the same training set as DeepACSON but with two labels, the intra-axonal space of myelinated axons versus the complement. To train FFN¹⁰, we used the same training set as DeepACSON but preserving the label of each myelinated axon. We first trained FFN, including the myelin and mitochondria labels, where the network generated very poor results. Therefore, we excluded the myelin label and included mitochondria to the intra-axonal space of myelinated axons. We trained DeepACSON and DeepEM2D/3D for one day and FFN for one week on a single NVIDIA Tesla V100-32 GB graphics processing unit (GPU). As shown in Fig. 6a-c, we quantitatively evaluated the segmentation on a test set comprising six SBEM volumes. We compared these techniques by the variation of information (VOI, split and merge contribution, lower is better), Wallace indices (split and merge contribution, higher is better), and adapted Rand error (ARE,

lower is better), defined in Materials and Methods, on the segmentation of the intra-axonal space. DeepACSON outperformed these current state-of-the-art techniques as it generated the smallest VOI measures and ARE and the biggest Wallace measures.

References:

5. Abdollahzadeh, A., Belevich, I., Jokitalo, E., Tohka, J. & Sierra, A. Automated 3D Axonal Morphometry of White Matter. *Sci. Reports* **9**, 6084 (2019).

8. Zeng, T., Wu, B. & Ji, S. DeepEM3D: approaching human-level performance on 3D anisotropic EM image segmentation. *Bioinforma. (Oxford, England)* **33**, 2555–2562 (2017).

10. Januszewski, M. et al. High-precision automated reconstruction of neurons with flood-filling networks. *Nat. Methods* **15**, 605–610 (2018).

Reviewer #2:

Reviewer summary: DeepACSON is a segmentation method that utilizes segmented data from a previous method, ACSON, to train a network for segmenting lower resolution data. By reducing the size of the original data along with the segmented data from ACSON, DeepACSON can be trained with this “ground truth” data. The authors claim two major advantages of this technique:

1. Less human interaction, “providing human-annotation-free training sets.”
2. Being able to use lower resolution and thus larger field of view datasets.

Community Impact

This method can be useful to the neuronal community that uses EM data for their studies. The ability to take a single high-resolution sample to train from and then use low-resolution there after could allow for higher throughput. A much larger section of brain matter could be imaged quickly and more reliably at this lower resolution.

Originality

This method is very close to previous methods mentioned in the Introduction. However, DeepACSON differs enough to be called an improvement or at least a hybrid of prior work.

Answer: We thank the Reviewer for the accurate summary of the study. We have highlighted and clarified the differences to previous methods as well as provided comparisons to them in the revised version.

Comment 2.1: It would be of interest to the reader to have timing comparison of this method to some of the others. Five days to segment a single image seems to be a lot. But maybe, this needs to be in the context of volume per hour or something comparable.

Answer 2.1: We now compared the computation time of DeepACSON against DeepEM2D⁸, DeepEM3D⁸, and FFN¹⁰ over six datasets of size 290×290×285 voxel³. On average, DeepACSON spends four minutes to segment volumes of such size, compared to FFN, which on average, spends about 28 minutes. DeepEM2D and DeepEM3D, on average, require about one minute to segment a dataset of size 290×290×285 voxel³. However, DeepEM2D/3D apply no mechanism to address potential topological errors. The computation time from the shortest to the longest was as follows: DeepEM2D, DeepEM3D, DeepACSON, and FFN. We also included the computation complexity of deepACSON for the BM4D¹⁵ denoising step, cylindrical shape decomposition algorithm¹⁴, and convolutional neural networks (CNNs) in generating semantic segmentations. We added this information to the Results section on **page 11-13, line 217-235**, and also in the panel **g of the new Figure 6**.

Computation time

The pipeline required about five days to segment a raw, low-resolution SBEM dataset of 4000×2000×1200 voxels into its final segmentation. The details are presented in Supplementary Table S4. Approximately 40% of the DeepACSON computation time was spent during BM4D denoising¹⁵. We run BM4D filtering on non-overlapping patches of the SBEM volumes to enable parallel processing. We remark that BM4D filtering can be dropped from the pre-processing steps to improve the computation time, but as shown in Fig. 6 d-f, dropping BM4D filtering made segmentations less accurate. Approximately 30% of the DeepACSON computation time was spent on the CSD algorithm. In more detail, the time complexity of the sub-voxel precise skeletonization is $O(n N_{\Omega} \log N_{\Omega})$, where n is the number of skeleton branches, and N_{Ω} is the number of voxels of a discrete object, i.e., a myelinated axon. The $N_{\Omega} \log N_{\Omega}$ factor is from the fast marching algorithm²¹. The time complexity to determine a critical point is $O(N_p)$, where N_p is the number of inquiry points to check for the cross-sectional changes in a decomposition interval. Therefore, the overall time complexity of the CSD algorithm is $O(n N_{\Omega} \log N_{\Omega}) + O(N_p)$. Approximately 10% of the DeepACSON computation time was spent on the semantic segmentation, with time complexity of $O(N)$, where N is the number of voxels in an SBEM dataset.

We also compared the computation time of DeepACSON, DeepEM2D, DeepEM3D, and FFN techniques (see Fig.6g for results). These techniques were compared over the six test datasets on a computer with an NVIDIA Tesla V100-32 GB GPU, 2 × Intel Xeon E5 2630 CPU 2.4 GHz, and 512 GB RAM. DeepEM2D and DeepEM3D had the shortest computation time (about 1 minute) as the segmentation mainly relies on an Inception-ResNet-v2 network²² and watershed segmentation. FFN required the longest computation time for an end-to-end segmentation (about 28 minutes). DeepACSON required about 4 minutes (using 15 CPU cores) to segment the test datasets, which was longer than DeepEM2D/3D and shorter than FFN.

Figure 6. DeepACSON evaluations. Comparison of DeepACSON against state-of-the-art segmentation, DeepEM2D, DeepEM3D, and FFN, by (a) variation of information (VOI, split and merge contribution, lower is better), (b) Wallace indices (split and merge contribution, higher is better), and (c) adapted Rand error (ARE, lower is better) and the sum of VOI split and VOI merge (VOI sum, lower is better). DeepACSON outperformed these techniques as it produced the smallest VOI split, VOI merge, VOI sum, and ARE and the biggest Wallace split and merge values. Comparison of the design parameters of DeepACSON: standard DeepACSON (DeepACSON-A), a U-Net with residual modules (DeepACSON-B), the effect of BM4D denoising (DeepACSON-C), and adjusting the resolution between the training and test sets (DeepACSON-D) over (d) VOI (split and merge contribution) (e) Wallace indices (split and merge contribution), and (f) ARE and VOI sum. The filled circles and error bars show the mean and standard deviation of the evaluations, respectively. The dash-dotted lines show the choice of binarization threshold. The comparisons were run over the best threshold, i.e., smallest VOI merge and VOI split. (g) Comparing the computation time of DeepACSON against DeepEM2D/3D and FFN (mean ± standard deviation). All comparisons were run over six test SBEM datasets of size $290 \times 290 \times 285$ voxel³ automatically segmented using the ACSON⁵ pipeline.

References:

5. Abdollahzadeh, A., Belevich, I., Jokitalo, E., Tohka, J. & Sierra, A. Automated 3D Axonal Morphometry of White Matter. *Sci. Reports* **9**, 6084 (2019).
8. Zeng, T., Wu, B. & Ji, S. DeepEM3D: approaching human-level performance on 3D anisotropic EM image segmentation. *Bioinforma. (Oxford, England)* **33**, 2555–2562 (2017).
10. Januszewski, M. et al. High-precision automated reconstruction of neurons with flood-filling networks. *Nat. Methods* **15**, 605–610 (2018).
14. Abdollahzadeh, A., Sierra, A. & Tohka, J. Cylindrical shape decomposition for 3D segmentation of tubular objects. *arXiv:1911.00571v2 [cs.CV](2019)*.

15. Maggioni, M., Katkovnik, V., Egiazarian, K. & Foi, A. Nonlocal Transform-Domain Filter for Volumetric Data Denoising and Reconstruction. *IEEE Transactions on Image Process.* **22**, 119–133 (2013).

21. Sethian, J. A. A fast marching level set method for monotonically advancing fronts. *Proc. Natl. Acad. Sci.* **93**, 1591–1595 (1996).

22. Längkvist, M., Karlsson, L. & Loutfi, A. A review of unsupervised feature learning and deep learning for time-series modeling. *Pattern Recognit. Lett.* **42**, 11–24 (2014).

Comment 2.2: This manuscript emphasizes imaging speed. However, it is not clear how much is gained without knowing if this method is just as fast computationally as others? In the discussion section it is mentioned that it takes 24 hours to train on the high-resolution data using ACSON. It would be beneficial for the authors to comment on where most of the computational time is spent, clarify the timing from start to finish. Meaning, break out the training phase from the high-resolution, training on low-resolution, and running final segmentation.

Answer 2.2: As described in the **Answer 2.1**, we now expanded the Computation time section of the manuscript. We included the computation complexity of the major time-consuming steps of the DeepACSON pipeline. The percentage of the computation time of the major time-consuming steps over the duration of the whole pipeline was stated. We also compared the computation time of DeepACSON to DeepEM2D, DeepEM3D, and FFN techniques. DeepACSON performed faster than FFN but slower than DeepEM2D/3D. Note that DeepEM2D and DeepEM3D do not apply a mechanism to address possible topological errors of the segmentation, as in FFN or DeepACSON. Please, see **Answer 2.1** for a comprehensive response regarding the computation time.

Comment 2.3: It is mentioned that training sets were “augmented using randomized histogram distortions...” to make this method more robust to changes in contrast and brightness. How does this perform in practice with different sample preparation? Were there samples made by more than one person?

Answer 2.3: Our samples were prepared for SBEM by a single person following a pedantic procedure. We speculate that with proper data augmentation, CNNs can generalize their solution when the intensity and contrast vary because of the sample preparation by more than one person or due to different staining protocols. Note that the intensity of an ultrastructure, e.g., myelin, or the contrast between different ultrastructures, e.g., between myelin and the intra-axonal space, varies even within a dataset or among datasets prepared by one person under the same staining protocol. DeepACSON segmentation was equally robust against the intensity and contrast variations within and among different datasets.

Comment 2.4: It is not clear why a new GUI was necessary to write for validation. How do the authors ensure the GUI is not biasing the results in any way?

Answer 2.4: The reason for developing the GUI called gACSON was to facilitate the visualization and validation procedures for the expert. gACSON enables the expert to manually mouse-click on the segmented image overlaid on the original EM image, and express if a segmentation component was a true-positive (TP), false-positive (FP), or false-negative (FN) as shown in the **Supplementary Figure S1, page i**. The source code for gACSON is available at <https://github.com/AndreaBehan/g-ACSON>. A GUI, in itself, cannot ensure an unbiased evaluation of the segmentation because the GUI just counts the number of TP, FP, and FN that the expert assigns and calculates the precision, recall, and F1 scores. We minimized the biasedness of the expert by providing no access to the dataset ID nor the sampling location. The expert evaluated 50 samples of 300×300 voxels, and each sample included about 200 axonal cross-sections. The expert evaluated about 10000 axonal cross-sections providing robust insight into the segmentation accuracy. We added this information to the Evaluations section.

Results, page 10, line170-176:

2) a test set, which comprised 50 patches of size 300×300 voxels only for the expert evaluations. We randomly sampled every low-resolution (large field-of-view) dataset for five non-overlapping windows of size 300×300 voxels (10 datasets, 50 samples). Each patch, on average, included approximately 130 axonal cross-sections and 30 mitochondria. Therefore, the expert has evaluated about 6500 axonal cross-sections and 1500 mitochondria in total. The expert had no access to the dataset ID nor the sampling location. The expert evaluated the sampled images of the final segmentation by counting the number of true-positives (TP), false-positives (FP), and false-negatives (FN).

Results, page 11, line 210-214:

Finally, an expert (A.S.) evaluated the DeepACSON segmentation of myelinated axons and mitochondria in at an object-level using GUI-based visualization software, called gACSON²⁰ that we developed for this purpose (Supplementary Fig.S1). We developed gACSON to facilitate the visualization and validation procedures for the expert. gACSON enables the expert to manually mouse-click on the segmented image overlaid on the original EM image and express if a segmentation component was a TP, FP, or FN, as shown in Supplementary Fig.S1.

Comment 2.5: The simplification of mitochondria to a centroid seems ill advised to measure distribution. Take for instance two large mitochondria almost touching. These centroids can be equi-distance to that of two small mitochondria. However, the two large mitochondria would seem more dense than the two small ones... I think minimum distance between mitochondria, or even the coverage of mitochondria projected on the skeleton would be a more informative metric. Alternatively, the centroid distance can be useful if the variance of mitochondria length is small enough. In which case, reporting this variance would be sufficient..

Answer 2.5: We defined an alternative measure for the inter-mitochondrial distance, as suggested by the reviewer. Now, we measure the inter-mitochondrial distance along each myelinated axon using two definitions: we projected the centroids of mitochondria on the

axonal skeleton and measured the geodesic distance between the consecutive projected centroids, and we projected the entirety of mitochondria on the axonal skeleton and measured the shortest geodesic distance between two consecutive mitochondria. Both measurements are equivalently descriptive of the distance between consecutive mitochondria along axons, and they are highly correlated; the Pearson correlation coefficient within each dataset was 0.99, where the number of observations exceeds 10000 per dataset. We compared the inter-mitochondrial distance, both definitions, between sham-operated and TBI rats, and we did not find a significant difference between the two groups. We added this information to Results and Supplementary Information.

Results, page 7-8, line 116-121:

To quantify the spatial distribution of mitochondria, we measured the inter-mitochondrial distances along each myelinated axon. We applied two definitions to quantify the inter-mitochondrial distances: we projected the centroids of mitochondria on the axonal skeleton and measured the geodesic distance between the consecutive projected centroids, and we projected the entirety of mitochondria on the axonal skeleton and measured the shortest geodesic distance between two consecutive mitochondria as shown in Supplementary Fig. S2.

Results, page 9, line 145-152:

We did not find a difference between sham-operated and TBI rats regarding the inter-mitochondrial distances (distance between centroids) of the contralateral cingulum ($F = 0.33$, $p = 0.603$), contralateral corpus callosum ($F = 0.07$, $p = 0.812$), ipsilateral cingulum ($F = 6.26$, $p = 0.086$), and ipsilateral corpus callosum ($F = 1.04$, $p = 0.414$) (Fig. 5d), nor for the inter-mitochondrial distances when measuring the shortest distance between consecutive mitochondria in the contralateral cingulum ($F = 0.28$, $p = 0.630$), contralateral corpus callosum ($F = 0.05$, $p = 0.830$), ipsilateral cingulum ($F = 7.10$, $p = 0.073$), and ipsilateral corpus callosum ($F = 0.43$, $p = 0.577$) (Supplementary Fig. S2). Defining the inter-mitochondrial distance as the distance between centroids of mitochondria was highly correlated with defining the inter-mitochondrial distance as the shortest distance between consecutive mitochondria; the Pearson correlation coefficient was 0.99.

Supplementary Information, page ii:

Figure S2. Quantification of the inter-mitochondrial distance. (a) We defined the inter-mitochondrial distance in two manners: we projected the entirety of mitochondria on the axonal skeleton and measured the shortest geodesic distance between two consecutive mitochondria, (d_1), or we projected the centroids of mitochondria on the axonal skeleton and measured the geodesic distance between the consecutive projected centroids (d_2). (b) We compared the inter-mitochondrial distance, d_1 definition, between sham-operated and TBI rats. We did not find significant differences between the groups in any of the brain areas. (c) We compared the inter-mitochondrial distance, d_2 definition, between sham-operated and TBI groups. Comparing these two groups, we did not find significant differences in any of the brain areas. On each bean plot, the central mark indicates the median, and the left and right edges of the box indicate the 25th and 75th percentiles, respectively. The whiskers extend to the most extreme data points not considered outliers. The colors correspond with the animal ID.

Comment 2.6: It is mentioned that the volumes of the cells could not be directly compared because of variation between samples. Is there a reason for this variation? Can it not be normalized out in some manner?

Answer 2.6: We thank the reviewer for pointing this out. The single membrane of cell body/processes and single membrane of unmyelinated axons do not resolve at $50 \times 50 \times 50 \text{ nm}^3$ resolution, and we can not discern these two ultrastructures apart. Therefore, we could not measure the cell volume, but the volume of the extra-axonal space, i.e., cell volume plus the volume of unmyelinated axons. The volume of the extra-axonal space can vary among

datasets. For example, the presence of blood vessels in a dataset causes a variation in the volume of the extra-axonal space for two reasons: the volume of the vessels themselves and the number of cells surrounding the walls of the vessels (astrocytes, pericytes). Now, we modified this sentence in **Results, page 9, line 153-154** as:

We could not directly compare the volume of the myelin and myelinated axons among datasets because the volume of the extra-axonal space varies among datasets.

Comment 2.7: In the discussion section, it is not clear what the authors mean by “error-aware” in the second paragraph.

Answer 2.7: We modified this paragraph and removed the term “error-aware.” This paragraph can be read as:

Discussion, page 13, line 247-249:

The top-down design of DeepACSON instance segmentation allows for including a priori knowledge of the topology of neuronal processes that makes it different from the bottom-up design of the current automated neurite segmentation techniques^{8-11, 23-25}.

References:

- 8.** Zeng, T., Wu, B. & Ji, S. DeepEM3D: approaching human-level performance on 3D anisotropic EM image segmentation. *Bioinforma.* (Oxford, England) **33**, 2555–2562 (2017).
- 9.** Funke, J. et al. Large Scale Image Segmentation with Structured Loss Based Deep Learning for Connectome Reconstruction. *IEEE Transactions on Pattern Analysis Mach. Intell.* **41**, 1669–1680 (2019).
- 10.** Januszewski, M. et al. High-precision automated reconstruction of neurons with flood-filling networks. *Nat. Methods* **15**, 605–610 (2018).
- 11.** Meirovitch, Y. et al. Cross-Classification Clustering: An Efficient Multi-Object Tracking Technique for 3-D Instance Segmentation in Connectomics. In *The IEEE Conference on Computer Vision and Pattern Recognition (CVPR)* (2019).
- 23.** Funke, J., Andres, B., Hamprecht, F. A., Cardona, A. & Cook, M. Efficient automatic 3D-reconstruction of branching neurons from EM data. In *2012 IEEE Conference on Computer Vision and Pattern Recognition*, 1004–1011 (IEEE, 2012).
- 24.** Nunez-Iglesias, J., Kennedy, R., Plaza, S. M., Chakraborty, A. & Katz, W. T. Graph-based Active Learning of Agglomeration (GALA): A python library to segment 2D and 3D neuroimages. *Front. Neuroinformatics* **8**, 1–6 (2014).
- 25.** Maitin-Shepard, J., Jain, V., Januszewski, M., Li, P. & Abbeel, P. Combinatorial energy learning for image segmentation. *Adv. Neural Inf. Process. Syst.* 1974–1982 (2016).

Comment 2.8: My largest concern is that the source code has not been made available to the reviewers.

Answer 2.8: The code is published at <https://github.com/aAbdz/DeepACSON>

Reviewer recommendation: Over all this paper is well written. The benefits of this method if adopted should increase throughput of neuron tracing in large field of views. I would recommend that, after minor changes relating to the comments above and the release of the code for review, this manuscript be accepted.

Answer: We thank the reviewer for positive feedback.

Reviewer #3:

Comment 3.1: -Introduction and motivation: The reason for doing the image segmentation on the low-resolution images (using learning) is not well or clearly motivated.

Answer 3.1: The motivation of the low-resolution imaging is to reduce the imaging time and the size of datasets, imaging larger tissue volumes. Being able to image large tissue volumes is important, as for analyzing a longer length of axons, the entire cells and cell surroundings, and covering/imaging a representative tissue volume of two brain areas in the same sample (here, cingulum and corpus callosum). In our study, we acquired SBEM images of the white matter at low- and high-resolution simultaneously. The low-resolution datasets were acquired from big tissue volumes of $200 \times 100 \times 65 \mu\text{m}^3$ with a voxel size of $50 \times 50 \times 50 \text{ nm}^3$. The high-resolution datasets were acquired from small tissue volumes of $15 \times 15 \times 15 \mu\text{m}^3$ and imaged with a voxel size of $15 \times 15 \times 50 \text{ nm}^3$ (please, see Fig. 2a). The high-resolution datasets covered only a very small tissue volume, while the low-resolution images displayed a 400 times bigger field-of-view. Covering a large field-of-view at a high-resolution requires months of imaging and produces massive datasets, which can limit imaging the tissue volume per brain sample. Now, we clarified our motivation for low-resolution imaging in the Introduction section.

Figure 2. Low- and high-resolution SBEM imaging of the contralateral corpus callosum and cingulum of a sham dataset. **(a)** We acquired SBEM images of the white matter, corpus callosum (cc) and cingulum (cg), simultaneously at the high- and low-resolution. The field-of-view of the low-resolution dataset is $204.80 \times 102.20 \times 65.30 \mu\text{m}^3$ equivalent to $4096 \times 2044 \times 1306$ voxels in x, y, and z directions, respectively, which is about 400 times larger than the field-of-view of the high-resolution datasets. **(b)** Images of the low- and high-resolution datasets acquired from the same location (the orange-rendered volume in **(a)**). The visualization of the high- and low-resolution images shows that myelin, myelinated axons, mitochondria, and cell nuclei were resolved in both settings. In contrast, the axonal membrane at nodes of Ranvier (cyan panel, arrowheads) and unmyelinated axons (fuchsia panel, asterisks) was only resolved in the high-resolution images. The purple panel shows a cell nucleus from the low-resolution dataset **(a)**, where the membrane is resolved, but not continuously. **(c)** A 3D rendering of myelinated axons in the high-resolution SBEM dataset (contralateral sham #25) segmented by the automated ACSON pipeline.

Introduction, page 2, line 24-29:

Although these automated EM segmentation techniques have yielded accurate reconstructions of neuronal processes, they have focused on EM datasets at very high-resolution to explore synaptic connectivity. At synaptic resolutions, EM imaging of a large tissue volume generates a massive dataset. Imaging 1 mm^3 tissue at $4 \times 4 \times 40 \text{ nm}^3$ tera-voxels in size, demanding fully automated image acquisition techniques and microscopes which run for several months continuously^{2,3}. By taking coarser resolution images, we can make acquiring big tissue volumes a plausible

task: imaging 1 mm³ tissue at 50×50×50 nm³ generates a dataset of eight tera-voxels during a few days.

References:

2. Zheng, Z. et al. A Complete Electron Microscopy Volume of the Brain of Adult *Drosophila melanogaster*. *Cell* **174**, 730–743 (2018).

3. Maniates-Selvin, J. T. et al. Reconstruction of motor control circuits in adult *Drosophila* using automated transmission electron microscopy. *bioRxiv* 2020.01.10.902478 (2020). DOI 10.1101/2020.01.10.902478.

Comment 3.2: The introduction has a list of relevant studies with some trivialised details, but lacks distillation and categorisation of the related work. The introduction has not summarised so well about the existing works: what are the limitations and how this proposed new approach is motivated?

Answer 3.2: We now expanded the Introduction section on existing works and mentioned the limitations of the current techniques and the motivation of our approach.

Introduction, page 1-2, line 8-11

Semi-automated segmentation methods based on machine learning approaches^{4,6} have improved the rate of segmentation. However, these methods still require a considerable amount of manual interaction as the segmentation is driven on the manually extracted skeletons of neuronal processes, proofreading, or correction of errors.

Introduction, page 2, line 29-40

*However, imaging at low-resolution can limit the visualization of the cellular membranes, such as at nodes of Ranvier, where no distinctive image feature differentiates the intra- and extra-axonal space of a myelinated axon. Distinctive image features are required for a segmentation technique with a bottom-up design. Bottom-up design is subjected to greedy optimization, which makes the locally optimal choice at each stage while intending to find a global optimum. Therefore, the mentioned automated techniques⁷⁻¹¹ cannot be used to segment low-resolution images. Techniques such as DeepEM3D⁸ and its cloud-based implementation⁷, which only rely on a precise semantic segmentation, encounter either over- or under-segmentation errors, depending on the instance segmentation step. Or techniques such as FFN¹⁰ and its multi-object tracking counterpart¹¹, where networks learn the shape of a neural process, encounter over-segmentation errors. Merging FFN super-voxels does not necessarily generate a correct segmentation of an axon as the segmentation leaks to the extra-axonal space at nodes of Ranvier. Therefore, we developed DeepACSON, a **Deep** learning-based **AutomatiC** Segmentation of axONs, to account for severe membrane discontinuities inherited with low-resolution imaging for tens of thousands of myelinated axons.*

References:

4. Berning, M., Boergens, K. M. & Helmstaedter, M. *SegEM: Efficient Image Analysis for High-Resolution Connectomics*. *Neuron* **87**, 1193–1206 (2015).
6. Dorkenwald, S. et al. *Automated synaptic connectivity inference for volume electron microscopy*. *Nat. Methods* **14**, 435–442 (2017).
7. Haberl, M. G. et al. *CDeep3M—Plug-and-Play cloud-based deep learning for image segmentation*. *Nat. Methods* **15**, 677–680 (2018).
8. Zeng, T., Wu, B. & Ji, S. *DeepEM3D: approaching human-level performance on 3D anisotropic EM image segmentation*. *Bioinforma. (Oxford, England)* **33**, 2555–2562 (2017).
9. Funke, J. et al. *Large Scale Image Segmentation with Structured Loss Based Deep Learning for Connectome Reconstruction*. *IEEE Transactions on Pattern Analysis Mach. Intell.* **41**, 1669–1680 (2019).
10. Januszewski, M. et al. *High-precision automated reconstruction of neurons with flood-filling networks*. *Nat. Methods* **15**, 605–610 (2018).
11. Meirovitch, Y. et al. *Cross-Classification Clustering: An Efficient Multi-Object Tracking Technique for 3-D Instance Segmentation in Connectomics*. In *The IEEE Conference on Computer Vision and Pattern Recognition (CVPR)* (2019).

Comment 3.3: Compared with existing techniques, the paper's unique contribution is vague, because why going for low-resolution images is not elaborated clearly, especially the paper mentioned that the high-resolution images are actually available. (only a few sentences about the rationale of using low and high-resolution images are stated much later in the discussion session).

Answer 3.3: The high-resolution images are available, but only from a small portion of the total volume of the low-resolution images (please, see Fig. 2 in **Answer 3.1**). Our low-resolution images display 400 times bigger tissue volume than the high-resolution images. By working on bigger volumes, we have the possibility to trace longer length of myelinated axons or include representative volumes of the cingulum and corpus callosum. As stated in the **Answer 3.1**, we have now explained the motivation for low-resolution imaging in more detail in the revised manuscript. We also clarified the unique contribution of DeepACSON in **Introduction, page 2, line 38-40**:

*Therefore, we developed DeepACSON, a **Deep** learning-based **AutomatiC** Segmentation of axONs, to account for severe membrane discontinuities inherited with low-resolution imaging for tens of thousands of myelinated axons.*

We included the detail of the imaging acquisition of our SBEM images to emphasize the differences between high- and low-resolution to the reader in **Introduction, page 3, line 44-51**:

We applied DeepACSON on low-resolution, large field-of-view 3D-EM datasets acquired using serial block-face scanning electron microscopy¹³ (SBEM). The SBEM volumes were obtained from the corpus callosum and cingulum, in the same field-of-view, of five rats after sham-operation ($n = 2$) or traumatic brain injury (TBI) ($n = 3$). The images were acquired ipsi- and contralaterally, thus for five rats, we had ten samples. Each sample was SBEM imaged simultaneously at two resolutions, and in two fields-of-view: high-resolution images, $15 \times 15 \times 50 \text{ nm}^3$, were acquired in a small field-of-view, $15 \times 15 \times 15 \text{ }\mu\text{m}^3$, and low-resolution images, $50 \times 50 \times 50 \text{ nm}^3$, were acquired in a large field-of-view, $200 \times 100 \times 65 \text{ }\mu\text{m}^3$. The low-resolution images covered a field-of-view 400 times bigger than high-resolution images.

Also, we expanded the information of the SBEM datasets in **Results, page 3, line 65-73**:

Dataset

The samples were prepared for SBEM imaging by a single person following a pedantic procedure (Materials and Methods). We simultaneously acquired SBEM images of the white matter at the low- and high-resolution (Fig. 2a). The low-resolution datasets were acquired from big tissue volumes of $200 \times 100 \times 65 \text{ }\mu\text{m}^3$ with a voxel size of $50 \times 50 \times 50 \text{ nm}^3$. Two-thirds of the low-resolution images were from the corpus callosum and one-third from the cingulum (Supplementary Table S1). The high-resolution datasets were acquired from small tissue volumes of $15 \times 15 \times 15 \text{ }\mu\text{m}^3$ and imaged with a voxel size of $15 \times 15 \times 50 \text{ nm}^3$. The high-resolution images were acquired from the corpus callosum. All the images were acquired from the ipsi- and contralateral hemispheres of the sham-operated and TBI animals. Figure 2a shows the contralateral corpus callosum and cingulum of a sham-operated rat in the low- and high-resolution.

Additionally, we added the information regarding the high-resolution images in the Supplementary Table S1.

Supplementary Information, page ix

Table S1. Characteristics of the low-resolution (LR) and high-resolution (HR) SBEM datasets. For each rat, we collected the low-resolution images from the ipsi- and contralateral of the corpus callosum and cingulum. The low-resolution images from the ipsilateral of the sham 49 rat included only the cingulum. The high-resolution images were collected from the ipsi- and contralateral of the corpus callosum. The size of datasets is given in voxels (x,y,z).

Condition	Rat ID	LR-size (voxel)	LR (nm^3)	HR-size (voxel)	HR (nm^3)
Sham	#25 contra	2044 × 4096 × 1306	50 × 50 × 50	1042 × 1048 × 285	13.8 × 13.8 × 50
	#25 ipsi	4096 × 2048 × 1384	50 × 50 × 50	1049 × 1076 × 285	15.4 × 15.4 × 50
	#49 contra	4096 × 2048 × 1882	50 × 50 × 50	1081 × 1053 × 285	18.3 × 18.3 × 50
	#49 ipsi	2048 × 2048 × 1210	50 × 50 × 50	1037 × 1058 × 285	13.0 × 13.0 × 50
TBI	#2 contra	4096 × 2048 × 1086	50 × 50 × 50	1048 × 1124 × 285	15.0 × 15.0 × 50
	#2 ipsi	2154 × 4134 × 620	50 × 50 × 50	1343 × 1316 × 285	15.0 × 15.0 × 50
	#24 contra	4091 × 2028 × 1348	50 × 50 × 50	1289 × 1280 × 285	15.0 × 15.0 × 50
	#24 ipsi	2946 × 2162 × 1250	50 × 50 × 50	1290 × 1295 × 285	15.0 × 15.0 × 50
	#28 contra	4096 × 2048 × 1278	50 × 50 × 50	1076 × 1051 × 285	16.5 × 16.5 × 50
	#28 ipsi	4075 × 2000 × 1300	50 × 50 × 50	1035 × 1056 × 285	16.5 × 16.5 × 50

References:

13. Denk, W. & Horstmann, H. *Serial Block-Face Scanning Electron Microscopy to Reconstruct Three-Dimensional Tissue Nanostructure. PLoS Biol.* 2, e329 (2004).

Comment 3.4: The paper uses bottom-up and top-down design as the standard to classify different learning techniques, which is a debatable way of differentiating their own work from the previous works. In essence, the bottom-up approach uses the local features to detect or represent semantic information - this is the case no matter how CNN is used (referring to thousands of papers from the computer vision), because this is how neural network works, since NNs have no parameterized model to represent the Marco information, and this brings the following statement into a question: "This approach is made possible by the use of a-priori knowledge of the topology of myelinated axons and cell nuclei.". How the typological information is used as prior knowledge? The best it can do is the pre-training of networks given enough datasets, which is different from the "knowledge" in normal human's definition.

Answer 3.4: We agree with the reviewer that in the case of CNNs for the semantic segmentation, this division would not be possible. But for the instance segmentation, which mostly comes after the semantic segmentation, we argue that our division holds. We now clarified that by the bottom-up and top-down designs, we refer to the instance segmentation strategies in the Introduction section:

Introduction, page 2, line 15-17:

DCNNs are typically used for the semantic segmentation, whereas other, more traditional image analysis techniques are used for the instance segmentation. Moreover, the segmentation techniques generally favor a bottom-up design, i.e., over-segmentation and subsequent merge.

Introduction, page 2-3, line 42-44:

However, the instance segmentation of DeepACSON approaches the segmentation problem from a top-down perspective, i.e., under-segmentation and subsequent split, using a priori knowledge of the topology of myelinated axons and cell nuclei.

Comment 3.5: -methodology

"The simultaneous high- and low-resolution imaging enables to use the high-resolution images as the training data to segment low-resolution datasets." There is a one-to-one correspondence between high res and low-resolution images, ie both of them record the same spatial-temporal information, and the low-resolution images *are* down-sampled from the very same higher resolution ones. Of course, using the high-resolution images as the ground truth for training the low-resolution datasets down-sampled from themselves, you can always get very good fitting results with minimal loss - using simply brute force supervised learning, for example.

Answer 3.5: The high- and low-resolution images overlap only for $15 \times 15 \times 15 \mu\text{m}^3$. The high-resolution images are available only for a small fraction of the total volume of the low-

resolution images, $200 \times 100 \times 65 \mu\text{m}^3$. Our low-resolution images display 400 times bigger tissue volume than the high-resolution images. Please, see **Answer 3.1** and Fig. 2 in the same answer. We down-sampled the high-resolution images to the resolution of low-resolution images to provide a training set for the CNNs. The semantic segmentation of the CNNs received high evaluation scores, which means that the CNNs have been successful in generalizing what they learned from the small high-resolution training material to the test set.

Comment 3.6: computational advantage

There's not yet information to compare the processing time for the low-resolution images vs the processing time of the higher resolution images *directly*. Note that most of the computational time is in (1) BM4D filtering and (2) CSD SVM code, which is not related with the CNNs being the selling point of the paper.

Answer 3.6: We have clarified our setup, as explained in **Answers 3.1, 3.2, and 3.3**. Our goal is to segment low-resolution images in a large field-of-view using the information learned from the high-resolution images acquired in a small field-of-view. Therefore, the processing of high-resolution images is not relevant to this manuscript as the acquisition of large field-of-view in high-resolutions currently is not possible. The automated processing of the high-resolution images can be done by our earlier ACSON⁵ method. As explained in the current manuscript, ACSON is utilized to generate the training data for DeepACSON.

We now included the computation complexity of the deepACSON pipeline. We also compared the computation time of DeepACSON against DeepEM2D, DeepEM3D, and FFN. The computation time of DeepACSON was shorter than FFN and longer than DeepEM2D, DeepEM3D. We added this information to **page 11-13, line 217-235, and Figure 6, panel g**.

Computation time

The pipeline required about five days to segment a raw, low-resolution SBEM dataset of $4000 \times 2000 \times 1200$ voxels into its final segmentation. The details are presented in Supplementary Table S4. Approximately 40% of the DeepACSON computation time was spent during BM4D denoising¹⁵. We run BM4D filtering on non-overlapping patches of the SBEM volumes to enable parallel processing. We remark that BM4D filtering can be dropped from the pre-processing steps to improve the computation time, but as shown in Fig. 6 d-f, dropping BM4D filtering made segmentations less accurate. Approximately 30% of the DeepACSON computation time was spent on the CSD algorithm. In more detail, the time complexity of the sub-voxel precise skeletonization is $O(n N_\Omega \log N_\Omega)$, where n is the number of skeleton branches, and N_Ω is the number of voxels of a discrete object, i.e., a myelinated axon. The $N_\Omega \log N_\Omega$ factor is from the fast marching algorithm²¹. The time complexity to determine a critical point is $O(N_p)$, where N_p is the number of inquiry points to check for the cross-sectional changes in a decomposition interval. Therefore, the overall time complexity of the CSD algorithm is $O(n N_\Omega \log N_\Omega) + O(N_p)$. Approximately 10% of the DeepACSON computation time was spent on the semantic segmentation, with time complexity of $O(N)$, where N is the number of voxels in an SBEM dataset.

We also compared the computation time of DeepACSON, DeepEM2D, DeepEM3D, and FFN techniques (see Fig.6g for results). These techniques were compared over

the six test datasets on a computer with an NVIDIA Tesla V100-32 GB GPU, 2 × Intel Xeon E5 2630 CPU 2.4 GHz, and 512 GB RAM. DeepEM2D and DeepEM3D had the shortest computation time (about 1 minute) as the segmentation mainly relies on an Inception-ResNet-v2 network²² and watershed segmentation. FFN required the longest computation time for an end-to-end segmentation (about 28 minutes). DeepACSON required about 4 minutes (using 15 CPU cores) to segment the test datasets, which was longer than DeepEM2D/3D and shorter than FFN.

Figure 6. DeepACSON evaluations. Comparison of DeepACSON against state-of-the-art segmentation, DeepEM2D, DeepEM3D, and FFN, by (a) variation of information (VOI, split and merge contribution, lower is better), (b) Wallace indices (split and merge contribution, higher is better), and (c) adapted Rand error (ARE, lower is better) and the sum of VOI split and VOI merge (VOI sum, lower is better). DeepACSON outperformed these techniques as it produced the smallest VOI split, VOI merge, VOI sum, and ARE and the biggest Wallace split and merge values. Comparison of the design parameters of DeepACSON: standard DeepACSON (DeepACSON-A), a U-Net with residual modules (DeepACSON-B), the effect of BM4D denoising (DeepACSON-C), and adjusting the resolution between the training and test sets (DeepACSON-D) over (d) VOI (split and merge contribution) (e) Wallace indices (split and merge contribution), and (f) ARE and VOI sum. The filled circles and error bars show the mean and standard deviation of the evaluations, respectively. The dash-dotted lines show the choice of binarization threshold. The comparisons were run over the best threshold, i.e., smallest VOI merge and VOI split. (g) Comparing the computation time of DeepACSON against DeepEM2D/3D and FFN (mean ± standard deviation). All comparisons were run over six test SBEM datasets of size 290×290×285 voxel³ automatically segmented using the ACSON⁵ pipeline.

References:

5. Abdollahzadeh, A., Belevich, I., Jokitalo, E., Tohka, J. & Sierra, A. Automated 3D Axonal Morphometry of White Matter. *Sci. Reports* **9**, 6084 (2019).

8. Zeng, T., Wu, B. & Ji, S. *DeepEM3D: approaching human-level performance on 3D anisotropic EM image segmentation*. *Bioinforma. (Oxford, England)* **33**, 2555–2562 (2017).

10. Januszewski, M. *et al.* *High-precision automated reconstruction of neurons with flood-filling networks*. *Nat. Methods* **15**, 605–610 (2018).

14. Abdollahzadeh, A., Sierra, A. & Tohka, J. *Cylindrical shape decomposition for 3D segmentation of tubular objects*. *arXiv:1911.00571v2 [cs.CV](2019)*.

15. Maggioni, M., Katkovnik, V., Egiazarian, K. & Foi, A. *Nonlocal Transform-Domain Filter for Volumetric Data Denoising and Reconstruction*. *IEEE Transactions on Image Process.* **22**, 119–133 (2013).

21. Sethian, J. A. *A fast marching level set method for monotonically advancing fronts*. *Proc. Natl. Acad. Sci.* **93**, 1591–1595 (1996).

22. Långkvist, M., Karlsson, L. & Loutfi, A. *A review of unsupervised feature learning and deep learning for time-series modeling*. *Pattern Recognit. Lett.* **42**, 11–24 (2014).

Comment 3.7: Computation time (The pipeline required about five days to segment one raw low-resolution SBEM dataset of size 4 000×2 000×1 200 voxels into its final segmentation. The details are presented in Supplementary Table S4).

Supplementary material gives the specification of CPU, GPU and time. There are several technical questions: 1 the pipeline seems not automatically because it's a mixture of MATLAB code and Python code; 2 the hurdle of the computational time is limited primarily by the MATLAB computing which is known to be slow, and the second is the CSD Python code (despite Python is popular in machine learning but it is too slow compared to C++, and especially if the CSD Python code is not optimised). In short, if the whole codebase is all written in Python class and heavily optimised, the speed should be much faster.

Even regardless of all these technical aspects, when it comes to the central motivation of the paper, what is the indication of this computation time? What it supports? Because it seems to possess such low-resolution images is also very slow (5 days). The computational time usually can be resolved by parallel computing, change/optimize the programming code, use clusters instead of PCs. Therefore the justification of this approach is not clear.

Answer 3.7: We agree with the reviewer that the pipeline is not written on a single platform. However, we argue that the pipeline is automatic, meaning that no human interaction is required for the segmentation steps.

We agree about the superiority of C++ to speed up the computation time. We also remark that Matlab and Python have almost the same computation performance (<https://julialang.org/benchmarks/>). Indeed, unlike Python, Matlab execution engine currently uses just-in-time compilation use to compile all Matlab codes (<https://se.mathworks.com/products/matlab/matlab-execution-engine.html>). However, the main limitation of MATLAB is that it is proprietary (Matlab compiler and stand-alone

applications provide a work-around, which, however, is not perfect). Our future work is to transfer the DeepACSON pipeline, end-to-end, to Python.

We agree that parallel-computing on high-performance computing (HPC) servers reduces the computation time. Henceforth, the cylindrical shape decomposition (CSD) technique analyzes myelinated axons in-parallel on HPCs. In fact, the ability of the CSD algorithm in parallel-processing of myelinated axons enabled analyzing hundreds of thousands of myelinated axons, which traversed the low-resolution datasets (**Discussion, page 13, line 261-263**). Also, we run BM4D filtering on HPCs in-parallel, such that we divided the low-resolution images into non-overlapping patches and applied BM4D on patches in-parallel.

Discussion, page 13, line 260-263.

The CSD technique can analyze myelinated axons in-parallel on high-performance computing (HPC) servers. The parallelizability of the CSD algorithm makes the DeepACSON pipeline highly scalable. For example, we analyzed hundreds of thousands of myelinated axons traversing large field-of-view datasets on different CPU cores of different HPC servers, reducing the computation time of the segmentation.

Comment 3.8: - motivation of using low-resolution images (especially down-sampled from existing high-resolution ones)

The author also clearly states that “We segmented the high-resolution datasets using our earlier automated ACSON pipeline⁴. We used the ACSON segmentation of the high-resolution datasets to train DeepACSON, eliminating the need for manually annotated training sets.” and “We acquired SBEM images of the white matter simultaneously at low- and high-resolution”.

So indeed, high-resolution datasets are available, which means that a lot of existing methods can be used directly to analyse as well, and therefore what is the motivation of using down-sampled low-resolution images?

Answer 3.8: We have clarified our setup, as explained in **Answers 3.1, 3.2, and 3.3**. The high-resolution images are available only for a small field-of-view, and we down-sampled the high-resolution images to build a training set for CNNs to semantic-segment the low-resolution (large field-of-view) images. We now compared DeepACSON to state-of-the-art automated methods in the segmentation of low-resolution (large field-of-view) images, as the reviewer suggested. This information is added to Results, Material and Methods, and Fig. 6a-c (**Answer 3.6**).

Please, see Fig. 6a-c in Answer 1.1 for the comparison results.

Results, page 10, line 177-188:

We compared DeepACSON with state-of-the-art segmentation techniques; DeepEM2D⁸ and DeepEM3D⁸, which rely on a precise semantic segmentation, and FFN¹⁰, which accounts for the shape of neural processes during the instance segmentation. We trained DeepEM3D⁸ using the same training set as DeepACSON

but with two labels, the intra-axonal space of myelinated axons versus the complement. To train FFN¹⁰, we used the same training set as DeepACSON but preserving the label of each myelinated axon. We first trained FFN, including the myelin and mitochondria labels, where the network generated very poor results. Therefore, we excluded the myelin label and included mitochondria to the intra-axonal space of myelinated axons. We trained DeepACSON and DeepEM2D/3D for one day and FFN for one week on a single NVIDIA Tesla V100-32 GB graphics processing unit (GPU). As shown in Fig.6a-c, we quantitatively evaluated the segmentation on a test set comprising six SBEM volumes. We compared these techniques by the variation of information (VOI, split and merge contribution, lower is better), Wallace indices (split and merge contribution, higher is better), and adapted Rand error (ARE, lower is better), defined in Materials and Methods, on the segmentation of the intra-axonal space. DeepACSON outperformed these current state-of-the-art techniques as it generated the smallest VOI measures and ARE and the biggest Wallace measures.

Materials and Methods, page 21-22, line 512-534

Segmentation evaluation metrics. We used precision P , recall R , and $F1$ scores to compare an automated segmentation to the ground truth, as there is a one-to-one match between voxels constituting the segmentations. To define these measures, let A and B be the sets of voxels of a particular ultrastructure (myelin, myelinated axon, mitochondrion) in an automated segmentation and ground-truth, respectively. We defined $P = \frac{|A \cap B|}{|B|}$, and $R = \frac{|A \cap B|}{|A|}$, and $F1$ score as $F1 = 2 \times \frac{P \times R}{P + R}$. The maximum for the precision, recall, and $F1$ score is equal to one when the test segmentation perfectly matches the ground-truth. However, these metrics do not describe topological differences, and they are sensitive to small changes in the region boundary. Therefore, we evaluated automated segmentation with metrics less sensitive to small variations in boundary, but sensitive to topological differences. For that, we measured variance of information⁶² (VOI, split and merge contributions) and Wallace indices⁶³ to account for merge and split errors of a segmentation separately. We also computed adapted Rand error (ARE) as defined by the SNEMI3D contest (<http://brainiac2.mit.edu/SNEMI3D/evaluation>) as 1 minus the maximal F-score of the Rand index⁶⁴ (excluding the zero component of the original labels). We performed these evaluations using the Gala library described by Nunez-Iglesias et al⁶⁵. The VOI metric is defined as the sum of the conditional entropies between two segmentations $VOI(A,B) = H(A|B) + H(B|A)$, where A is the automated segmentation, and B is the ground-truth. The VOI metric is decomposed into VOI split $H(A|B)$ and VOI merge $H(B|A)$ ⁶⁵. A lower VOI value indicates a better segmentation; for a perfect match between an automated segmentation and ground truth, we have VOI split = VOI merge = 0. Also, the Wallace splitting index is defined as $\frac{a}{a+b}$ and the Wallace merging index is defined as $\frac{a}{a+c}$, where a is the number of pairs of voxels in the input image that have the same label in A and the same label in B , b is the number of pairs of voxels in the input image that have the same label in A but a different label in B , c is the number of pairs of voxels in the input image that have a different label in A but the same label in B , and d is the number of pairs of voxels in the input image that have different labels in A and different labels in B . The Wallace indices are between 0 and 1, where a higher value indicates a better segmentation, i.e., less split and merge

compared to the ground truth. The Rand index is defined as $\frac{a+d}{a+b+c+d}$, where a , b , c , and d are defined the same as in the case of Wallace indices. Note that we applied the adapted Rand error, where a lower value indicates a better segmentation.

References

5. Abdollahzadeh, A., Belevich, I., Jokitalo, E., Tohka, J. & Sierra, A. Automated 3D Axonal Morphometry of White Matter. *Sci. Reports* **9**, 6084 (2019).
8. Zeng, T., Wu, B. & Ji, S. DeepEM3D: approaching human-level performance on 3D anisotropic EM image segmentation. *Bioinforma. (Oxford, England)* **33**, 2555–2562 (2017).
10. Januszewski, M. et al. High-precision automated reconstruction of neurons with flood-filling networks. *Nat. Methods* **15**, 605–610 (2018).
62. Meilã, M. Comparing Clusterings by the Variation of Information. In Schölkopf, B. & Warmuth, M. (eds.) *Learning Theory and Kernel Machines. Lecture Notes in Computer Science*, 173–187 (Springer, Berlin, Heidelberg, 2003).
63. Wallace, D. L. A Method for Comparing Two Hierarchical Clusterings: Comment. *J. Am. Stat. Assoc.* **78**, 569 (1983).
64. Rand, W. M. Objective Criteria for the Evaluation of Clustering Methods. *J. Am. Stat. Assoc.* **66**, 846 (1971).
65. Nunez-Iglesias, J., Kennedy, R., Parag, T., Shi, J. & Chklovskii, D. B. Machine learning of hierarchical clustering to segment 2D and 3D images. *PLoS ONE* **8** (2013).

Comment 3.9: Also, regarding “We used the ACSON segmentation of the high-resolution datasets to train DeepACSON”, we know that the training of deep neural network always have artifacts, where the tests are different from training sets. So how the authors take that into account? (also see a separate comment regarding support vector machine later).

Answer 3.9: We now performed an ablation study on the DeepACSON pipeline and evaluated the effect of down-sampling the training set to the resolution of low-resolution images. We found that the semantic segmentation of CNNs receives better evaluation scores including the down-sampling step compared to using the high-resolution images directly as the training set. We added this information in Results as follows:

Please, see Fig. 6d-f presented in Answer 3.6 for the results of the ablation study, and “segmentation evaluation metrics” in Answer 3.8.

Results, page 10-11, line 189-199:

We also evaluated the DeepACSON pipeline to understand its design parameters, such as denoising, resolution adjustment, and an alternative, deeper architecture. In

Fig.6d-f, we denoted the standard DeepACSON design as DeepACSON-A, which used a light fully convolutional network¹⁸ (FCN) for the semantic segmentation. The standard DeepACSON was trained using down-sampled and BM4D filtered volumes. We replaced the FCN design of the standard DeepACSON with a U-Net¹⁹ with residual modules, denoted as DeepACSON-B in Fig.6d-f. In this figure, we also show the effect of omitting BM4D¹⁵ denoising as a pre-processing step (DeepACSON-C) and down-sampling the high-resolution images to generate the training set (DeepACSON-D). In addition, we demonstrated the choice of thresholds at which the probability maps were binarized. Evaluations were run over the six SBEM volumes. The comparisons showed that the standard DeepACSON performed better than a deeper network, which was prone to over-fitting. Denoising the train/test datasets as a pre-processing step improved our results as did adjusting the resolution between the training and test sets. We binarized the probability maps at thresholds, which generated the smallest VOI split/merge values.

References

15. Maggioni, M., Katkovnik, V., Egiazarian, K. & Foi, A. Nonlocal Transform-Domain Filter for Volumetric Data Denoising and Reconstruction. *IEEE Transactions on Image Process.* **22**, 119–133 (2013).
18. Long, J., Shelhamer, E. & Darrell, T. Fully Convolutional Networks for Semantic Segmentation. *2015 IEEE Conf. on Comput. Vis. Pattern Recognit. (CVPR)(2015)*.
19. Ronneberger, O., Fischer, P. & Brox, T. U-Net: Convolutional Networks for Biomedical Image Segmentation. In Navab, N., Hornegger, J., Wells, W. & Frangi, A. (eds.) *MICCAI 2015*, 234–241 (Springer International Publishing, Cham, 2015).

Comment 3.10: In the Essence, the key scientific fundamental is that a low-resolution image, no matter original or down-sampled, loses important features and information, therefore to acquire truthful high-resolution data at the first place is the reason for scientists to improve high-resolution imaging techniques - that is a whole purpose, because no matter humans or trained CNNs, all the extracted information and understanding are all *probabilistic* based on limited information because of lack of certainty.

Answer 3.10: We agree with the reviewer that low-resolution imaging loses information. We reduced the imaging resolution to the point that we were still able to trace and quantify the cellular components of white matter. Therefore, we were able to image a larger volume of tissue in a shorter imaging time; please see **Answer 3.1 and 3.3**. As suggested by the reviewer, we clarify our motivation for the low-resolution imaging and the advantages of acquiring large field-of-view datasets. Please, see **Answer 3.1 and Discussion, page 14, line 295-297:**

Large field-of-view imaging enables quantifying parameters whose measurement in a small field-of-view is not reliable as the measurement in the small field-of-view may reflect a very local characteristic of the underlying ultrastructure. Particular examples of such parameters are the tortuosity of myelinated axons, inter-mitochondrial distance, and cell density.

Comment 3.11: Datasets

In addition to the description, Please provide a table to clearly show what are the datasets resolved in both low and high-resolution, which are only resolved only in high-resolution. A table is much clearer to read.

Answer 3.11: As suggested by the reviewer, we included the information regarding the high-resolution datasets in the Supplementary Table S1. Please, see Supplementary Table S1 in **Answer 3.3**.

Comment 3.12: “the remaining six datasets were reserved for testing”, and “six unseen high-resolution SBEM datasets labeled by the automated ACSON pipeline”

It is a small number of datasets, which are not sufficient to have valid statistical conclusions. Is there a more indicated number, for example, the number of images?

Answer 3.12: The evaluation metrics that we assessed the DeepACSON pipeline is now expanded. Therefore, as suggested by the reviewer, we now included the average number of myelinated axons, over which the evaluations were run. We added this information in the Evaluation section on **page 10, line 165-176**.

Evaluations

We used two test sets to evaluate the DeepACON pipeline: 1) a test set that comprised six high-resolution SBEM volumes down-sampled to the resolution of low-resolution images. We applied this test set to compare DeepACSON against state-of-the-art automated segmentation methods and perform an ablation study on the DeepACSON pipeline. Labels for this test set was provided automatically using ACSON⁵ pipeline and proofread by A.S. In this test set, each SBEM volume included approximately 300 axons, and thus we evaluated DeepACSON on approximately $6 \times 300 = 1800$ myelinated axons; 2) a test set, which comprised 50 patches of size 300×300 voxels only for the expert evaluations. We randomly sampled every low-resolution (large field-of-view) dataset for five non-overlapping windows of size 300×300 voxels (10 datasets, 50 samples). Each patch, on average, included approximately 130 axonal cross-sections and 30 mitochondria. Therefore, the expert has evaluated about 6500 axonal cross-sections and 1500 mitochondria in total. The expert had no access to the dataset ID nor the sampling location. The expert evaluated the sampled images of the final segmentation by counting the number of true-positives (TP), false-positives (FP), and false-negatives (FN).

Comment 3.13: -Evaluation

In that section “White matter 3D morphology analysis”, there are descriptions of how these measurements are quantified, but there’s no information about how this process of measurement is done. The question is that: is this procedure done manually, for

automatically? This is important because a manual procedure provides limited data and therefore the statistical indication is compromised. Instead, an automated procedure or benchmark can process a large quantity of data, yielding much better statistics. (Computer vision algorithms of object recognition have a well-established image bank benchmark to quantify.)

Answer 3.13: The white matter 3D morphology analysis was accomplished automatically; a moving reference frame, i.e., a perpendicular plane to the axonal skeleton, extracts cross-sections of a myelinated axon at each skeleton point. We added the term “automatically” to the text as:

Results, page 7, line 112-113:

For every myelinated axon, we automatically extracted cross-sections along its axonal skeleton, i.e., central axis, with a plane perpendicular to the skeleton.

Also, benchmarking DeepACSON requires white matter electron microscopy datasets appended with the ground truth segmentation, which to our knowledge, currently, there are no such datasets available. Therefore, we compared DeepACSON segmentation with state-of-the-art automated segmentation techniques on six SBEM volumes that we had available. Each of these six SBEM volumes, on average, included approximately 300 axons. In total, about 1800 myelinated axons, were used to perform the evaluations. We scored these techniques over the variation of information (split and merge), Wallace indices (split and merge), and adapted Rand error. This added comparison to state-of-the-art techniques was also run automatically. We included our evaluation in detail in **Answer 3.8**.

Comment 3.14: Also, it is not convincing that the evaluation using 1 human expert is conclusive, and we don't know how much the human expert is biased due to other factors (conflict of interest, benefits), despite an expert has no access to the info.

Answer 3.14: We agree with the reviewer that the expert's evaluation can be biased. However, the expertise of A.S. (from the authors' list) in microscopic tissue information and being a volunteer in contributing her time for the evaluation were the reasons that convinced us to finalize the evaluation section with an expert subjective point-of-view. We added new information regarding the test sets in **Answer 3.12**. The test set that the expert evaluated contained 50 patches, that each patch, on average, included approximately 130 axonal cross-sections and 30 mitochondria. Therefore, the expert has evaluated about 6500 axonal cross-sections and 1500 mitochondria in total. We minimized the biasedness of the expert by providing no access to the dataset ID nor the sampling location. In addition, we expanded the Evaluation section of the manuscript thoroughly, **Answer 3.8**, and the expert's evaluation now can be read as an added qualitative measure over the whole pipeline.

Comment 3.15: - CSD SVM

There are improvements that can be made regarding the relationship between the artefacts from the CSD composition algorithm using support vector machine and the overall

contribution of DeepACSON. There should be more discussion about ambiguities and artefacts processed from CSD the algorithm.

The trained SVM is manually designed and tuned (specifically adapted parameters), which do not generalise to corner cases. This makes step 4 “eliminating false positives” less robust for processing different new datasets.

Answer 3.15: As suggested by the reviewer, we discussed the CSD algorithm and its artifacts in **Discussion, page 13-14, line 259-267:**

We have compared the CSD algorithm to state-of-the-art shape decomposition techniques^{29,30}, and we have shown that it outperforms these methods in the segmentation applications¹⁴. The CSD technique can analyze myelinated axons in-parallel on high-performance computing (HPC) servers. The parallelizability of the CSD algorithm makes the DeepACSON pipeline highly scalable. For example, we analyzed hundreds of thousands of myelinated axons traversing large field-of-view datasets on different CPU cores of different HPC servers, reducing the computation time of the segmentation. We also remark that the CSD algorithm evaluates the cylindricity of an object using the object curve skeleton. In cases where the surface protrusion of the object is very irregular the skeletonization may over-estimate the number of skeleton branches. The CSD algorithm detects maximal-length straight sub-skeletons following the skeletonization, but yet over-segment the surface protrusion, yielding false-positives. We eliminated false-positives after the CSD algorithm using support vector machines.

References

- 14.** Abdollahzadeh, A., Sierra, A. & Tohka, J. Cylindrical shape decomposition for 3D segmentation of tubular objects. *arXiv: 1911.00571v2 [cs.CV]* (2019). URL <http://arxiv.org/abs/1911.00571>.
- 29.** Kaick, O. V., Fish, N., Kleiman, Y., Asafi, S. & Cohen-OR, D. Shape Segmentation by Approximate Convexity Analysis. *ACM Transactions on Graph.* **34**, 1–11 (2015).
- 30.** Reniers, D., van Wijk, J. J. & Telea, A. Computing Multiscale Curve and Surface Skeletons of Genus 0 Shapes Using a Global Importance Measure. *IEEE Transactions on Vis. Comput. Graph.* **14**, 355–368 (2008).

In the manuscript, we also clarified that the SVM hyperparameter selection is automatic. We modified the text mentioned in **Materials and Methods, page 21, line 502-505** as follows:

To determine the optimal SVM hyperparameters, the regularization parameter C and kernel parameter σ , we selected the pair that minimized 5-fold cross-validation error on the training set using Bayesian optimization algorithm^{60,61}. C and σ were constrained in the range $[10^{-6}, 10^6]$. The optimal parameters were $C = 1.12$ and $\sigma = 8.48$.

References

60. Mockus, J., Tiesis, V. & Zilinskas, A. *The application of Bayesian methods for seeking the extremum. Towards Glob. Optimisation 2*, 117–129 (1978).

61. Snoek, J., Larochelle, H. & Adams, R. P. *Practical Bayesian Optimization of Machine Learning Algorithms. In Proceedings of the 25th International Conference on Neural Information Processing Systems - Volume 2, NIPS'12*, 2951–2959 (Curran Associates Inc., USA, 2012).

Comment 3.16: - Discussion section

Many part of the discussion section actually is doing the job of the introduction section. Instead, the discussion should concentrate more on the conclusive findings and further implications.

Answer 3.16: We thank the reviewer for noticing this. We now removed repetitions from the Discussion section. By performing the ablation studies on the DeepACSON pipeline and comparing our method to state-of-the-art automated segmentation techniques, now we have a more conclusive discussion.

Comment 3.17: - English Writing and language usage

Some examples of imprecise wording: “The purple panel shows a cell nucleus from the low-resolution dataset (a), which membrane is resolved, but not continuously.”; “The segmentation of myelinated axons and cell nuclei was finalized by eliminating non-axonal and non-nucleus structures with support vector machines (SVMs).”; “We evaluated the performance of the two SVMs by a leave-one-group-out (LOGO) cross-validation, where the classifier was trained excluding the data from one group of animals (sham-operated or TBI) from training and evaluated against it (Supplementary Table S2).

Word usage: use “where” instead of “which”; use ‘using’ instead of ‘with’; avoid using several clauses and put ‘it’ at the end of a long sentence (what exactly this “it” is pointing to?).

Answer 3.17: Thank you for noticing these issues. We corrected the manuscript as suggested by the reviewer.

Comment 3.18: The paper overall needs significant re-writing and requires proofreading from native speakers and senior academics.

Answer 3.18: Senior academics have proofread the paper.

Reviewers' comments:

Reviewer #1 (Remarks to the Author):

The authors have addressed some of the concerns raised in the previous round, partially. The significance of the contribution remains weak as the improvements are limited/negligible to the previous studies.

My concerns are:

- Comments to the answer 1.1: The authors have stated that their proposed method outperforms the SOTA in terms of accuracy. As highlighted in previous review round, accuracy is not considered as a primary judgment on deep neural networks. This metric could lead to biased comparisons, and the evaluation could become prone to ill-posed deductions. The authors have stated that about 40% of the time has spent on denoising. It seems that DeepASCON has a minor contribution to the segmentation task. The authors have also stated that skipping the denoising step leads to poor results. The role of deep learning is to address and automate such limitations, and I am confident that with deep learning, we can fit the underlying to any noise pattern within data. In other words, it is reliable to automate fitting deep neural networks to data distribution and minimise data manipulation such as sophisticated pre-processing steps. Otherwise, the role of deep learning will remain insignificant as I find in this study.

The improvement of ARA remains very insignificant compared to the DeepEM2D/3D, as shown in Figure 6.

- Comments to the answer 1.2 and 1.3: the authors have proposed to perform ablation studies by using ResNet-34 as an encoder (pre-trained). I am not sure if this means ablation study. I cannot find any comprehensive statistical analysis over tweaking the network architecture "DeepASCON" and why they have opted ResNet-34, remains unclear. The ablation study is not satisfactory at this stage.

- Comments to the answer 1.6: I am afraid if the computational complexity means only a comparison of training/inference timings. The authors have failed to convince the reader why DeepASCON is computationally efficient over SOTA as they have highlighted one of the important features of DeepASCON as "faster" method.

In summary, my evaluation is that DeepASCON contributes to insignificant improvements over SOTA in terms of deep learning context.

Reviewer #2 (Remarks to the Author):

All of my concerns have been addressed. Recommend publish.

Reviewer #3 (Remarks to the Author):

Brief summary

The research work develops learning-based segmentation to solve membrane discontinuity and segment large field of view images using low-resolution images.

Overall: The manuscript has necessary revisions to reflect review comments. The revised paper contains more technical details, which improves the reproducibility of the paper.

- Specific comments

1. In the abstract, it is worth mentioning by one sentence what is algorithm, method or contribution that results in DeepACSON's ability to resolve severe membrane discontinuities.

2. To help readers to understand better the contributions of the scientific work, a dedicated explanation to state its novelty is needed. For example, is that an original invention of some image processing algorithms? Integration of existing technologies/techniques? Or a capability or functionality that has not been achieved before? (eg, specifically for solving the limitation of low-resolution imaging, as suggested in "Therefore, we developed DeepACSON, a Deep learning-based Automatic Segmentation of axONS, to account for severe membrane discontinuities inherited with low-resolution imaging for tens of thousands of myelinated axons.") At the moment, the Introduction has covered these related aspects very broadly, but it should be made clear for readers "what exactly the scientific contribution is".

3. The question still exists in the revision, because it is vague to use "knowledge" in the context of the a computer algorithm, unless human is manually involved in the process using human knowledge. In "i.e., under-segmentation and subsequent split, using a priori knowledge of the topology of myelinated axons and cell nuclei". Instead of "a priori knowledge of the topology of ..", does it intend to say "Distinct features" or "Distinct differences" of the topology of myelinated axons and cell nuclei?

4. In "We down-sampled the high-resolution images to the resolution of low-resolution images to provide a training set for the CNNs. The semantic segmentation of the CNNs received high evaluation scores, which means that the CNNs have been successful in generalizing what they learned from the small high-resolution training material to the test set." This has been made more clearer now regarding the training and test sets. Though the paper has info "Only four of the ten datasets were used as training data, while the remaining six datasets were reserved for testing.", this text is quite hidden. So, in addition to Table S1-S4, please provide a separate Table to provide and list the number of training and test sets, respectively, and include the training loss and score values.

5. Related to comment 4, since it is quite small training sets in order to generalize, please include some discussion of the "overfitting" problem in the paper as well.

6. For better clarity, some key statements shall be presented in abstract and introduction earlier on. There are good, clear statements in the response letter as such: “we down-sampled the high-resolution images to build a training set for CNNs to semantic-segment the low-resolution (large field-of-view) images”, and “Our goal is to segment low-resolution images in a large field-of-view using the information learned from the high-resolution images acquired in a small field-of-view.”. These are more straightforward, easily understandable compared to many descriptions in the paper. However, this simple goal and clear statement should be made clear and added in the Introduction. The Introduction now stacks quite some technical details/numbers, and such a one-sentence summary of goal can very much improve the clarity.

7. Expert evaluation. As in Answer 3.14 “and the expert’s evaluation now can be read as an added qualitative measure over the whole pipeline.”, it is worth adding a clear sentence to explicitly say: Expert evaluation is an “added qualitative measure”, or “an additional qualitative measure”.

- Consistency, accuracy, grammar/brevity of the writing

Newly revised text is not well integrated and rather a stack of separate sentences. At times, readers have to read back and forth of several sentences to figure out what a noun or pronoun refers to.

The newly added text is less polished and needs more rework to improve in terms of basic scientific writing, and to match the rest of the paper, eg redundant repetition of words and minor grammar mistakes (wrong use of preposition). Some examples, suggestions below.

The low-resolution images covered a field-of-view 400 times bigger than high-resolution images.

 The low-resolution images covered a field-of-view 400 times bigger than *the* high-resolution images.

Semi-automated segmentation methods based on machine learning approaches^{4,6} have improved the rate of segmentation. However, these methods still require a considerable amount of manual interaction as the segmentation is driven on the manually extracted skeletons of neuronal processes, proofreading, or correction of errors.

 Semi-automated segmentation methods based on machine learning approaches^{4,6} have improved the rate of segmentation, but still require a considerable amount of manual interaction because of the manually extracted skeletons of neuronal processes, proofreading, or correction of errors.

as the segmentation is driven on the manually extracted skeletons of neuronal processes, proofreading, or correction of errors.

 is driven on  is based on

Distinctive image features are required for a segmentation technique with a bottom-up design. Bottom-up design is subjected to greedy optimization, which makes the locally optimal choice at each stage while intending to find a global optimum.

 Distinctive image features are required for a segmentation technique with a bottom-up design that is subjected to greedy optimization, making the locally optimal choice at each stage while finding a global optimum.

Therefore, the mentioned automated techniques 7-11 cannot be used to segment low-resolution images. Techniques such as DeepEM3D8 and its cloud-based implementation 7,

 Therefore, the mentioned automated techniques 7-11 cannot be used to segment low-resolution images, such as DeepEM3D8 and its cloud-based implementation 7, which

Reviewers' comments:

Reviewer #1:

The authors have addressed some of the concerns raised in the previous round, partially. The significance of the contribution remains weak as the improvements are limited/negligible to the previous studies.

Comment 1.1: Comments to the answer 1.1: The authors have stated that their proposed method outperforms the SOTA in terms of accuracy. As highlighted in previous review round, accuracy is not considered as a primary judgment on deep neural networks. This metric could lead to biased comparisons, and the evaluation could become prone to ill-posed deductions.

Answer 1.1: We think that this comment is partly due to our previous answer 1.1, where we used the term 'accuracy' in an imprecise way, referring to the performance metrics in general rather than to the particular performance metric 'accuracy.' We fully agree with the Reviewer that the accuracy should not be used as a primary metric on deep learning-based segmentation tasks, and it was not our intention to argue for this. In the previous revision, we have evaluated DeepACSON and different deep learning-based segmentation methods using various metrics (F1-score, precision, recall, the variance of information, adapted Rand index, Wallace index) defined in the paragraph "Segmentation evaluation metrics" at the end of the methods section⁸⁻¹¹. We have

specified that we used multiple metrics to evaluate the segmentation in the Results section:

Results, pages 10-11, lines 191-195

We compared these techniques on the segmentation of the intra-axonal space using three metrics: the variation of information (VOI, split and merge contribution, lower value is better), Wallace indices (split and merge contribution, higher value is better), and adapted Rand error (ARE, lower value is better). These metrics are defined in Materials and Methods. DeepACSON outperformed these current state-of-the-art techniques as it generated the smallest VOI measures and ARE and the biggest Wallace measures.

Results, page 11, lines 209-211

In addition, we evaluated the semantic segmentation of the standard DeepACSON on an ultrastructural level, i.e., myelin and myelinated axons (including mitochondria), on the six SBEM volumes. For this evaluation, we reported precision (positive predictive value), recall (sensitivity), and F1 scores (harmonic mean of precision and recall) in Supplementary Fig. S1.

Moreover, DeepACSON is a pipeline, and neural networks are only one component of the pipeline, responsible for semantic segmentation. We now clarified the goal of developing the DeepACSON pipeline and its contributions in the **Introduction, page 2-3, lines 39-45**:

Our goal is to segment low-resolution images in a large field-of-view using the information learned from the high-resolution images acquired in a small field-of-view. To achieve this goal, we developed a pipeline called DeepACSON, a Deep learning-based AutomatiC Segmentation of axONs, to account for severe membrane discontinuities inescapable with low-resolution imaging of tens of thousands of myelinated axons. The proposed pipeline utilizes an innovative combination of the existing deep learning-based methods for semantic segmentation and a novel shape decomposition technique for instance segmentation that uses the information about the geometry of myelinated axons and cell nuclei. Applying DeepACSON, we were able to segment low-resolution large field-of-view datasets of white matter automatically.

References

- 8.** Zeng, T., Wu, B. & Ji, S. DeepEM3D: approaching human-level performance on 3D anisotropic EM image segmentation. *Bioinforma.* (Oxford, England) **33**, 2555–2562 (2017).
- 9.** Funke, J. et al. Large Scale Image Segmentation with Structured Loss Based Deep Learning for Connectome Reconstruction. *IEEE Transactions on Pattern Analysis Mach. Intell.* **41**, 1669–1680 (2019).
- 10.** Januszewski, M. et al. High-precision automated reconstruction of neurons with flood-filling networks. *Nat. Methods* **15**, 605–610 (2018).

11. Meirovitch, Y. et al. Cross-Classification Clustering: An Efficient Multi-Object Tracking Technique for 3-D Instance Segmentation in Connectomics. In 2019 IEEE/CVF Conference on Computer Vision and Pattern Recognition (CVPR), 8417–8427 (IEEE, 2019).

Comment 1.2: The authors have stated that about 40% of the time has spent on denoising. It seems that DeepASCON has a minor contribution to the segmentation task. The authors have also stated that skipping the denoising step leads to poor results. The role of deep learning is to address an automate such limitations, and I am confident that with deep learning, we can fit the underlying to any noise pattern within data. In other words, it is reliable to automate fitting deep neural networks to data distribution and minimise data manipulation such as sophisticated pre-processing steps. Otherwise, the role of deep learning will remain insignificant as I find in this study.

Answer 1.2: We agree with the reviewer that spending 40% of the pipeline on denoising is computationally expensive. However, BM4D is an advanced non-local denoising method, whose application as a pre-processing step improved the segmentation results. As the time consumption of the segmentation pipeline is not as important as the segmentation quality, we argue that it is better to use advanced pre-processing steps if these can improve the quality of the final segmentation. We emphasized this information in **Results, page 12, lines 227-231** as follows:

Approximately 40 % of the DeepACSON computation time was spent on BM4D denoising¹⁵. We run BM4D filtering on non-overlapping patches of the SBEM volumes to enable parallel processing. BM4D is computationally expensive for denoising large EM volumes; however, as shown in Fig.6 d-f, the application of BM4D improved the segmentation results. The number of floating point operations required by BM4D is $O(N)$ with large constants, where N is the number of voxels²¹.

References

15. Maggioni, M., Katkovnik, V., Egiazarian, K. & Foi, A. Nonlocal Transform-Domain Filter for Volumetric Data Denoising and Reconstruction. *IEEE Transactions on Image Process.* **22**, 119–133 (2013).

21. Dabov, K., Foi, A., Katkovnik, V. & Egiazarian, K. Image denoising with block-matching and 3D filtering. In Dougherty, E. R., Astola, J. T., Egiazarian, K. O., Nasrabadi, N. M. & Rizvi, S. A. (eds.) *Image Processing: Algorithms and Systems, Neural Networks, and Machine Learning*, vol. 6064, 606414 (2006).

Comment 1.3: The improvement of ARA remains very insignificant compared to the DeepEM2D/3D, as shown in Figure 6.

Answer 1.3: The comparison of DeepACSON to DeepEM2D/3D and FFN showed that while improvements in adapted Rand error (ARE) were modest, the variation of information (VOI) and Wallace metrics substantially improved as compared to SOTA. The reference [65] argues that the VOI metric has several advantages over the Rand Index (RI) and is a better metric for comparing segmentation results. For example, errors in the VOI scale linearly with the error size, whereas the Rand Index scales quadratically. This makes VOI more directly comparable between volumes. Also, because RI is based on point pairs and the vast majority of pairs are in disjoint regions, RI has a limited useful range near zero, and that range is different for each dataset. In contrast, VOI ranges between zero and $\log(K)$, where K is the number of objects in the image. This information is added to the **Materials and Methods, page 22, lines 548-553:**

As Nunez-Iglesias et al.⁶⁵ argued, the VOI metric has several advantages over the Rand index and is a better metric for comparing EM segmentation results. For example, errors in the VOI scale linearly with the error size, whereas the Rand index scales quadratically, making VOI more directly comparable between volumes than the Rand index. Also, the Rand index has a limited useful range near one, and that range is different for each image. In contrast, VOI ranges between zero and $\log(K)$, where K is the number of objects in the image.

Furthermore, DeepEM2D/3D does not apply a mechanism to address potential topological errors, neither for over-segmentation nor under-segmentation. DeepACSON addresses the under-segmentation error in myelinated axons, as depicted in Supplementary Fig. S3. We cannot ignore topological errors because one of our goals is to extract morphological information, such as axonal tortuosity and diameter, to study ultrastructures. In the **Introduction, page 2, lines 33-35**, we now clarified that DeepEM2D/3D does not address the topological errors after semantic segmentation.

For example, techniques such as DeepEM3D⁸ and its cloud-based implementation⁷ essentially rely on a precise semantic segmentation and apply no mechanism to correct potential topological errors during instance segmentation. Therefore, semantic segmentation errors propagate into instance segmentation as either over- or under-segmentation.

Figure S3. Decomposition of under-segmented myelinated axons into their semantic axonal components using the CSD algorithm.

References

7. Haberl, M. G. et al. CDeep3M—Plug-and-Play cloud-based deep learning for image segmentation. *Nat. Methods* **15**, 677–680 (2018).
8. Zeng, T., Wu, B. & Ji, S. DeepEM3D: approaching human-level performance on 3D anisotropic EM image segmentation. *Bioinforma. (Oxford, England)* **33**, 2555–2562 (2017).
65. Nunez-Iglesias, J., Kennedy, R., Parag, T., Shi, J. & Chklovskii, D. B. Machine learning of hierarchical clustering to segment 2D and 3D images. *PLoS ONE* **8** (2013).

Comment 1.4. Comments to the answer 1.2 and 1.3: the authors have proposed to perform ablation studies by using ResNet-34 as an encoder (pre-trained). I am not sure if this means ablation study. I cannot find any comprehensive statistical analysis over

tweaking the network architecture “DeepACSON” and why they have opted ResNet-34, remains unclear. The ablation study is not satisfactory at this stage.

Answer 1.4: We have studied the main components of the DeepACSON pipeline, replacing the original FCN network with a deeper network (U-Net with ResNet encoder), the effect of BM4D denoising as a pre-processing step, and the resolution adjustment between training and test sets. Considering that we used a standard FCN architecture in the DeepACSON pipeline, we consider this high-level ablation study more important than the traditional ablation study of the FCN architecture details. This information is now added to the **Results, page 11, lines 196-199:**

We evaluated the DeepACSON pipeline to understand the behavior of its main components better. We replaced the original fully convolutional network¹⁸ (FCN) with a U-Net¹⁹ and omitted the BM4D denoising and resolution adjustment steps from the pipeline. We considered this high-level ablation study more informative than the traditional ablation study of the details of the standard FCN architecture.

The original design of DeepACSON neural networks had ten layers and did not include residual blocks. Therefore, we used a U-Net with ResNet encoder (a deeper network which includes residual blocks) to compare to the original DeepACSON. We selected the U-Net architecture because it is widely used for semantic segmentation of biomedical image volumes, resulting in precise segmentation and not requiring many annotated training images²⁹. Also, ResNet is the most common network for image feature extraction, as we used in the encoding path of U-Net; the residual blocks of ResNet are easier to optimize and can gain accuracy by increasing the network depth⁵⁴. We now added this information to **Materials and Methods, page 18, lines 425-429:**

We selected the U-Net architecture because it is widely used for the semantic segmentation of biomedical image volumes, resulting in precise segmentation and not requiring many annotated training images²⁹. Also, ResNet, which we used in the encoding path of U-Net, is the most widely used network for image feature extraction. The residual blocks of ResNet are easy to optimize and can gain accuracy from increased network depth⁵⁴.

We also demonstrated that a deep architecture could experience overfitting and produce worse results than the original DeepACSON design. We now discussed this in the **Discussion, page 14, lines 275-281:**

We trained the FCN of DeepACSON with four SBEM volumes to segment myelinated axons and six volumes to segment cell nuclei. These volumes included about 300 axons, one or two cell nuclei, and approximately 300x300x300 voxels that is sufficient to train a semantic segmentation network according to our experiments. We note that training a network for semantic segmentation does not necessarily need many annotated training images²⁹. To avoid overfitting, DeepACSON utilized a ten-layers fully convolutional network in its original design. We compared this design to a deeper network (U-net with ResNet encoder), demonstrating that the deeper network can experience

overfitting and produce worse VOI, ARE, and Wallace indices than the original design.

References

18. Long, J., Shelhamer, E. & Darrell, T. Fully Convolutional Networks for Semantic Segmentation. 2015 IEEE Conf. on Comput. Vis. Pattern Recognit. (CVPR) (2015).
19. Ronneberger, O., Fischer, P. & Brox, T. U-Net: Convolutional Networks for Biomedical Image Segmentation. In Navab, N., Hornegger, J., Wells, W. & Frangi, A. (eds.) MICCAI 2015, 234–241 (Springer International Publishing, Cham, 2015).
29. Falk, T. et al. U-Net: deep learning for cell counting, detection, and morphometry. Nat. Methods **16**, 67–70 (2019).
54. He, K., Zhang, X., Ren, S. & Sun, J. Deep residual learning for image recognition. Proc. IEEE Comput. Soc. Conf. on Comput. Vis. Pattern Recognit. 2016-Decem, 770–778 (2016).

Comment 1.5: Comments to the answer 1.6: I am afraid if the computational complexity means only a comparison of training/inference timings. The authors have failed to convince the reader why DeepASCON is computationally efficient over SOTA as they have highlighted one of the important features of DeepASCON as “faster” method.

Answer 1.5: We agree with the reviewer that computational complexity may not only mean the training/inference timing. Therefore, we have included the complexity of the method measured through the number of basic arithmetic operations performed. This information is described in the **Results, pages 12-13, lines 227-237** as follows:

Approximately 40 % of the DeepACSON computation time was spent on BM4D denoising¹⁵. We run BM4D filtering on non-overlapping patches of the SBEM volumes to enable parallel processing. BM4D is computationally expensive for denoising large EM volumes; however, as shown in Fig.6 d-f, the application of BM4D improved the segmentation results. The number of floating point operations required by BM4D is $O(N)$ with large constants, where N is the number of voxels²¹. Approximately 30% of the DeepACSON computation time was spent on the CSD algorithm. In more detail, the time complexity of the sub-voxel precise skeletonization is $O(n N_{\Omega} \log N_{\Omega})$, where n is the number of skeleton branches, and N_{Ω} is the number of voxels of a discrete object, i.e., a myelinated axon. The $N_{\Omega} \log N_{\Omega}$ factor is from the fast marching algorithm²¹. The time complexity to determine a critical point is $O(N_p)$, where N_p is the number of inquiry points to check for the cross-sectional changes in a decomposition interval. Therefore, the overall time complexity of the CSD algorithm is $O(n N_{\Omega} \log N_{\Omega}) + O(N_p)$ for one axon. The inference time of the FCN corresponded to approximately 10% of the DeepACSON computation time. For the general analysis of the time complexity of FCNs, we refer to²³.

References

15. Maggioni, M., Katkovnik, V., Egiazarian, K. & Foi, A. Nonlocal Transform-Domain Filter for Volumetric Data Denoising and Reconstruction. *IEEE Transactions on Image Process.* **22**, 119–133 (2013).
21. Dabov, K., Foi, A., Katkovnik, V. & Egiazarian, K. Image denoising with block-matching and 3D filtering. In Dougherty, E. R., Astola, J. T., Egiazarian, K. O., Nasrabadi, N. M. & Rizvi, S. A. (eds.) *Image Processing: Algorithms and Systems, Neural Networks, and Machine Learning*, vol. 6064, 606414 (2006).
23. He, K. & Sun, J. Convolutional neural networks at constrained time cost. In *2015 IEEE Conference on Computer Vision and Pattern Recognition (CVPR)*, vol. 15, 5353–5360 (IEEE, 2015).

Reviewer summary: In summary, my evaluation is that DeepASCON contributes to insignificant improvements over SOTA in terms of deep learning context.

We thank the reviewer for the suggestions and comments, through which we improved our manuscript. As mentioned in **Answer 1.1**, we now emphasized that the contribution of DeepACSON is not about the neural networks but addressing the fundamental problem of under-segmentation in image segmentation of EM datasets. Furthermore, DeepACSON segments white matter in its components, offering quantitative biological information in 3D, and a tissue model with various applications in different research disciplines.

Reviewer #2:

Reviewer recommendation: All of my concerns have been addressed. Recommend publish.

We thank the reviewer for the recommendation.

Reviewer #3:

Reviewer summary: The research work develops learning-based segmentation to solve membrane discontinuity and segment large field of view images using low-resolution images.

Overall: The manuscript has necessary revisions to reflect review comments. The revised paper contains more technical details, which improves the reproducibility of the paper.

We thank the reviewer for the positive feedback.

Comment 3.1: In the abstract, it is worth mentioning by one sentence what is algorithm, method or contribution that results in DeepACSON's ability to resolve severe membrane discontinuities.

Answer 3.1: As suggested by the reviewer, we now added a sentence into the Abstract about the contribution of the DeepACSON pipeline to resolve severe membrane discontinuities of myelinated axons:

Abstract, page 1:

With its top-down design, DeepACSON manages to account for severe membrane discontinuities inescapable with the low-resolution imaging. In particular, the instance segmentation of DeepACSON uses the tubularity of myelinated axons, decomposing an under-segmented myelinated axon into its constituent axons.

Comment 3.2: To help readers to understand better the contributions of the scientific of work, a dedicated explanation to state its novelty is needed. For example, is that an original invention of some image processing algorithms? Integration of existing technologies/techniques? Or a capability or functionality that has not been achieved before? (eg, specifically for solving the limitation of low-resolution imaging, as suggested in "Therefore, we developed DeepACSON, a Deep learning-based AutomatiC Segmentation of axONs, to account for severe membrane discontinuities inherited with low-resolution imaging for tens of thousands of myelinated axons.") At the moment, the Introduction has covered these related aspects very broadly, but it should be made clear for readers "what exactly the scientific contribution is".

Answer 3.2: We added the scientific contribution of DeepACSON to the **Introduction, pages 2-3, lines 39-45**, as follows:

*Our goal is to segment low-resolution images in a large field-of-view using the information learned from the high-resolution images acquired in a small field-of-view. To achieve this goal, we developed a pipeline called DeepACSON, a **Deep** learning-based **AutomatiC Segmentation of axONs**, to account for severe membrane discontinuities inescapable with low-resolution imaging of tens of thousands of myelinated axons. The proposed pipeline utilizes an innovative combination of the existing deep learning-based methods for semantic segmentation and a novel shape decomposition technique for instance segmentation that uses the information about the geometry of myelinated axons and cell nuclei. Applying DeepACSON, we were able to segment low-resolution large field-of-view datasets of white matter automatically.*

Comment 3.3: The question still exists in the revision, because it is vague to use "knowledge" in the context of the a computer algorithm, unless human is manually involved in the process using human knowledge. In "i.e., under-segmentation and subsequent split, using a priori knowledge of the topology of myelinated axons and cell nuclei". Instead of "a priori knowledge of the topology of ..", does it intend to say

“Distinct features” or “Distinct differences” of the topology of myelinated axons and cell nuclei?

Answer 3.3: We removed the term “a prior knowledge” and replaced it by a more specific description, referring to distinct shape features in the **Introduction, page 3, lines 47-49:**

However, the instance segmentation of DeepACSON approaches the segmentation problem from a top-down perspective, i.e., under-segmentation and subsequent split, using the tubularity of the shape of myelinated axons and the sphericity of the shape of cell nuclei.

Discussion, page 13, lines 255-257:

The top-down design of DeepACSON instance segmentation allows for including the tubularity of the shape of myelinated axons and the sphericity of the shape of cell nuclei that make it different from the bottom-up design of the current automated neurite segmentation techniques^{8-11, 23-25}.

Comment 3.4: In “We down-sampled the high-resolution images to the resolution of low-resolution images to provide a training set for the CNNs. The semantic segmentation of the CNNs received high evaluation scores, which means that the CNNs have been successful in generalizing what they learned from the small high-resolution training material to the test set.” This has been made more clearer now regarding the training and test sets. Though the paper has info “Only four of the ten datasets were used as training data, while the remaining six datasets were reserved for testing.”, this text is quite hidden. So, in addition to Table S1-S4, please provide a separate Table to provide and list the number of training and test sets, respectively, and include the training loss and score values.

Answer 3.4: As suggested by the reviewer, we now added the information regarding the training and test sets and the training/validation loss to **Supplementary Information, page i, Figure S1**. We evaluated DeepACSON neural networks on test sets using Precision, Recall, and F1 score metrics. The same metrics over the training set are provided as a reference. We also merged supplementary Table S2 of the previous revision into supplementary Figure S1a.

Supplementary Information, page i, Figure S1

Figure S1. DeepACSON evaluation scores. **(a)** We evaluated DeepACSON neural networks on test sets using Precision, Recall, and F1 score metrics. The same metrics over the training set are provided as a reference. No large differences between the training and test metrics exist, demonstrating in part that the network did not overfit. In the DCNN-mAx section, red rows show evaluations of myelin semantic segmentation, and gray rows show evaluations of the semantic segmentation of intra-axonal spaces. The DCNN-cN training set included only ten cell nuclei, and we used all the volumes for training. The performance of SVMs was evaluated using leave-one-group-out (LOGO) cross-validation (CV). An expert evaluated the final segmentation of myelinated axons and mitochondria as an added qualitative measure over the entire pipeline. The maximum value of all scores is one. **(b)** The training and validation losses of DCNN-mAx. **(c)** The training and validation losses of DCNN-cN. We trained the networks on an NVIDIA Tesla P100-16 GB GPU for one day.

Comment 3.5: Related to comment 4, since it is quite small training sets in order to generalize, please include some discussion of the “overfitting” problem in the paper as well.

Answer 3.5: As suggested, we now discussed the overfitting problem in the **Discussion, page 14, lines 275-281:**

We trained the FCN of DeepACSON with four SBEM volumes to segment myelinated axons and six volumes to segment cell nuclei. These volumes included about 300 axons, one or two cell nuclei, and approximately 300×300×300 voxels that is sufficient to train a semantic segmentation network according to our experiments. We note that training a network for semantic segmentation does not necessarily need many annotated training images²⁹. To avoid overfitting, DeepACSON utilized a ten-layers fully convolutional network in its original design. We compared this design to a deeper network (U-net with ResNet encoder), demonstrating that the deeper network can experience overfitting and produce worse VOI, ARE, and Wallace indices than the original design.

References

29. Falk, T. et al. U-Net: deep learning for cell counting, detection, and morphometry. *Nat. Methods* **16**, 67–70 (2019).

Comment 3.6: For better clarity, some key statements shall be presented in abstract and introduction earlier on. There are good, clear statements in the response letter as such: “we down-sampled the high-resolution images to build a training set for CNNs to semantic-segment the low-resolution (large field-of-view) images”, and “Our goal is to segment low-resolution images in a large field-of-view using the information learned from the high-resolution images acquired in a small field-of-view.”. These are more straightforward, easily understandable compared to many descriptions in the paper. However, this simple goal and clear statement should be made clear and added in the Introduction. The Introduction now stacks quite some technical details/numbers, and such a one-sentence summary of goal can very much improve the clarity.

Answer 3.6: We thank the Reviewer for the suggestions. We added these statements to the **Abstract** and **Introduction** sections, clarifying our goals and contributions.

Abstract, page 1:

With its top-down design, DeepACSON manages to account for severe membrane discontinuities inescapable with the low-resolution imaging. In particular, the instance segmentation of DeepACSON uses the tubularity of myelinated axons, decomposing an under-segmented myelinated axon into its constituent axons.

Introduction, pages 2-3, lines 39-42:

Our goal is to segment low-resolution images in a large field-of-view using the information learned from the high-resolution images acquired in a small field-of-view. To achieve this goal, we developed a pipeline called DeepACSON, a Deep learning-based AutomatiC Segmentation of axONs, to account for severe

membrane discontinuities inescapable with low-resolution imaging of tens of thousands of myelinated axons.

Introduction, page 3, lines 56-59:

We down-sampled the high-resolution images to build a training set for DCNNs for the semantic segmentation of the low-resolution (large field-of-view) images, eliminating the need for manually annotated training sets. Using the DeepACSON pipeline, we segmented the low-resolution datasets, which sum up to $1.09 \times 10^7 \mu\text{m}^3$ of white matter tissue, into myelin, myelinated axons, mitochondria, and cell nuclei.

Comment 3.7: Expert evaluation. As in Answer 3.14 “and the expert’s evaluation now can be read as an added qualitative measure over the whole pipeline.”, it is worth adding a clear sentence to explicitly say: Expert evaluation is an “added qualitative measure”, or “an additional qualitative measure”.

Answer 3.7: As suggested by the Reviewer, we explicitly mentioned that the expert’s evaluation is an added qualitative measure.

Results, page 11, line 222-224:

The expert's evaluation is an added qualitative measure over the entire pipeline, which resulted in the following scores: myelinated axons (precision: 0.965 ± 0.027 , recall: 0.877 ± 0.061 , and F1 score: 0.918 ± 0.038) and mitochondria (precision: 0.856 ± 0.100 , recall: 0.804 ± 0.091 , and F1 score: 0.823 ± 0.067).

Comment 3.8: Consistency, accuracy, grammar/brevity of the writing

Newly revised text is not well integrated and rather a stack of separate sentences. At times, readers have to read back and forth of several sentences to figure out what a noun or pronoun refers to. The newly added text is less polished and needs more rework to improve in terms of basic scientific writing, and to match the rest of the paper, eg redundant repetition of words and minor grammar mistakes (wrong use of preposition). Some examples, suggestions below.

Comment 3.8.1: The low-resolution images covered a field-of-view 400 times bigger than high-resolution images.

→ The low-resolution images covered a field-of-view 400 times bigger than *the* high-resolution images.

Comment 3.8.2: Semi-automated segmentation methods based on machine learning approaches 4,6 have improved the rate of segmentation. However, these methods still require a considerable amount of manual interaction as the segmentation is driven on the manually extracted skeletons of neuronal processes, proofreading, or correction of errors.

→ Semi-automated segmentation methods based on machine learning approaches 4,6 have improved the rate of segmentation, but still require a considerable amount of manual interaction because of the manually extracted skeletons of neuronal processes, proofreading, or correction of errors.

Comment 3.8.3: as the segmentation is driven on the manually extracted skeletons of neuronal processes, proofreading, or correction of errors.

→ is driven on  is based on

Comment 3.8.4: Distinctive image features are required for a segmentation technique with a bottom-up design. Bottom-up design is subjected to greedy optimization, which makes the locally optimal choice at each stage while intending to find a global optimum.

→ Distinctive image features are required for a segmentation technique with a bottom-up design that is subjected to greedy optimization, making the locally optimal choice at each stage while finding a global optimum.

Comment 3.8.5: Therefore, the mentioned automated techniques 7-11 cannot be used to segment low-resolution images. Techniques such as DeepEM3D 8 and its cloud-based implementation 7,

→ Therefore, the mentioned automated techniques7-11 cannot be used to segment low-resolution images, such as DeepEM3D8 and its cloud-based implementation7, which

Answer 3.8: We thank the Reviewer for these excellent suggestions. We corrected the manuscript as suggested in comments 3.8.1-5, and we have carefully proofread the manuscript to improve its clarity.

REVIEWERS' COMMENTS:

Reviewer #1 (Remarks to the Author):

All my comments/concerns have been addressed. Recommend publication.